# ATM signaling modulates cohesin behavior in meiotic prophase and proliferating cells

Zhouliang Yu [1,2,3,4], Hyung Jun Kim[1] & Abby F. Dernburg [1,2,3,4] ✉

Cohesins are ancient and ubiquitous regulators of chromosome architecture and function, but their diverse roles and regulation remain poorly understood. During meiosis, chromosomes are reorganized as linear arrays of chromatin loops around a cohesin axis. This unique organization underlies homolog pairing, synapsis, double-stranded break induction, and recombination. We report that axis assembly in *Caenorhabditis elegans* is promoted by DNA-damage response (DDR) kinases that are activated at meiotic entry, even in the absence of DNA breaks. Downregulation of the cohesin-destabilizing factor WAPL-1 by ATM-1 promotes axis association of cohesins containing the meiotic kleisins COH-3 and COH-4. ECO-1 and PDS-5 also contribute to stabilizing axis-associated meiotic cohesins. Further, our data suggest that cohesin-enriched domains that promote DNA repair in mammalian cells also depend on WAPL inhibition by ATM. Thus, DDR and Wapl seem to play conserved roles in cohesin regulation in meiotic prophase and proliferating cells.

Three-dimensional chromosome architecture is strongly influenced by cohesins. The core cohesin complex comprises four proteins: a heterodimer of two large ATPases, the structural maintenance of chromosomes (SMC) proteins Smc1 and Smc3; a HEAT repeat protein known as sister chromatid cohesion 3/stromal antigen (Scc3/SA/Stag); and an alpha-kleisin protein. Cohesins are related to condensins and the SMC5/6 complex, and homologous SMC complexes are found across all domains of life[1,2].

Cohesin binding during DNA replication establishes sister chromatid cohesion, which is required for accurate chromosome segregation during mitosis and meiosis. Cohesins are also molecular motors that move along chromatin, forming loops that govern chromosome topology. They contribute to transcriptional regulation, chromosome condensation, and repair of DNA damage. The diverse chromosome architectures observed in different cell types are presumed to be modulated by cohesin subunits and additional regulatory factors, but how these factors collaborate to shape chromosomes in distinct contexts remains mysterious.

Sexually reproducing organisms produce haploid gametes through the specialized cell-division process of meiosis. During meiotic prophase, homologous chromosomes pair and undergo synapsis to enable recombination and crossover formation, which underlie reductional chromosome segregation and genetic variation[3–5]. In early meiosis, replicated chromosomes become highly elongated as cohesins reorganize to form a linear chromosome 'axis.'[6] Axis morphogenesis is a prerequisite for the induction of meiotic double-strand breaks (DSBs), homologous chromosome pairing, synapsis, and DSB repair[7–13]. Meiotic cohesins also recruit additional axis proteins that influence and monitor synapsis and recombination and regulate cell cycle progression[8,10,14–17].

Remodeling of meiotic chromosomes to form an axis-loop structure is thought to be driven in part by expression of meiosis-specific cohesin subunits. All eukaryotes studied to date express one or more meiosis-specific kleisins, and some also have meiosis-specific SMC and/or Scc3/Stag isoforms[10,13,18]. However, it is largely unknown how the activities of meiotic cohesins differ from those in other cells, except that the Rec8 kleisin can be selectively protected from cleavage to keep sister chromatids together during the first meiotic division.

Less attention has been paid to meiotic roles or regulation of factors that modulate the loading, unloading, and dynamics of cohesins

[1]Department of Molecular and Cell Biology, University of California, Berkeley, Berkeley, CA, USA. [2]Howard Hughes Medical Institute, Chevy Chase, MD, USA. [3]Biological Systems and Engineering Division, Lawrence Berkeley National Laboratory, Berkeley, CA, USA. [4]California Institute for Quantitative Biosciences, Berkeley, CA, USA. ✉e-mail: afdernburg@berkeley.edu

on chromatin. These include the 'loading' complex Scc2–Scc4, the acetyltransferase Eco1/Ctf7, the 'release factor' Wapl, and Pds5. Wapl destabilizes cohesin binding to chromosomes by a mechanism that is independent of kleisin cleavage. This activity is inhibited by Eco1-mediated acetylation of Smc3 but is promoted in some contexts by Pds5 (refs. [19–27]).

Here, we investigate the mechanism of chromosome remodeling during the early meiotic (EM) prophase in *C. elegans*. This nematode expresses two types of meiotic kleisins: REC-8 and COH-3 and COH-4 (COH-3/4). COH-3/4 are closely related paralogs with overlapping roles and are thus regarded as a single type of kleisin[12]. REC-8 and COH-3/4 are essential for homolog pairing and synapsis but have distinct roles (Fig. 1a). REC-8 is expressed in premeiotic (PM) germ cells and is thus present during DNA replication[12]; REC-8 cohesins mediate sister chromatid cohesion (SCC) that prevents inter-sister recombination and synapsis[28]. COH-3/4 are expressed only after replication; these complexes likely create chromatin loops that emanate from the axis and are more abundant than REC-8 cohesins[12,15,28,29]. No other meiosis-specific cohesin proteins have been identified in *C. elegans*. During most of meiotic prophase, cohesin complexes containing REC-8 and COH-3/4 localize along the length of chromosome axes. Following crossover designation at mid-pachytene, the two types of cohesins become enriched on reciprocal 'arms' of the bivalent to mediate two sequential rounds of segregation[12,30].

Wapl/Rad61 is a widely conserved cohesin regulator that was identified in screens for radiation sensitivity in budding yeast and mitotic defects in *Drosophila*. Its best-known role is promoting the release of 'arm' cohesion during mitotic prophase. Here, we show that *C. elegans* WAPL-1 is downregulated by ATM-1 (ataxia telangiectasia mutated, ATM) at meiotic entry. This inhibition promotes or stabilizes COH-3/4 binding along chromosome axes. Surprisingly, we find that ATM-1 is activated at meiotic entry by the CHK-2 (Chk2) kinase, which, together with ECO-1 (Eco1) and PDS-5 (Pds5), preferentially protects cohesin complexes containing REC-8 from the effects of WAPL-1. Together, these findings reveal that constitutive activation of DDR kinases at meiotic entry reshapes the genome through cohesins to promote interhomolog interactions and meiotic recombination. Finally, we extend our observations to show that inhibition of WAPL by ATM promotes cohesin enrichment at sites of DNA damage in proliferating human cells.

## Results

### CHK-2 leads to downregulation of WAPL-1 at meiotic entry

Immunolocalization of *C. elegans* WAPL-1 (Wapl) reveals diffuse nuclear localization in most tissues, including the germline. Its localization to chromatin drops abruptly at meiotic entry[31,32]. We were intrigued by this finding because depletion of WAPL from mammalian cells during interphase can lead to formation of 'vermicelli,' linear cohesin chromosome cores that resemble meiotic chromosome axes[33]. This reduction was more pronounced in dissected, immunostained gonads than in

intact animals expressing green fluorescent protein (GFP)-tagged WAPL-1 (ref. [32]), suggesting that WAPL-1 probably dissociates from chromosomes, rather than being degraded in early meiosis.

WAPL-1 reaccumulates in oocyte nuclei during later stages of meiotic prophase and contributes to removing COH-3/4 cohesin from the axes as chromosomes condense during diplotene and diakinesis[31]. However, this role is nonessential; *wapl-1*-null mutants are viable and fertile, with normal meiotic segregation, as indicated by a low production of male progeny, which arise through X chromosome nondisjunction and are diagnostic for meiotic errors[31,32], in broods of self-fertilizing hermaphrodites. They do show hallmarks of reduced mitotic fidelity, including some embryonic and larval lethality and low-penetrance egg-laying and locomotion defects[32].

We found that loss of WAPL-1 from chromatin at meiotic entry requires the essential meiotic kinase CHK-2 (Fig. 1a). Either *chk-2* loss-of-function mutations or auxin-induced degradation of CHK-2 resulted in persistence of WAPL-1 on meiotic chromosomes (Fig. 1b,d). Loss of CHK-2 does not abolish axis assembly or recruitment of the HORMA domain protein HTP-3 (Fig. 1c). However, the abundance of COH-3 was greatly reduced following CHK-2 depletion (Fig. 1e,f), in accordance with prior evidence that WAPL-1 preferentially releases COH-3/4-containing cohesins from meiotic chromosomes during late prophase[31]. Co-depletion of WAPL-1 and CHK-2 fully restored COH-3/4 to wild-type levels (Fig. 1e,f). Interestingly, we also observed a marked increase in REC-8 intensity following this co-depletion (Fig. 1e,g), similar to observations in *wapl-1* mutants[34].

WAPL-1 contains four consensus phosphorylation sites for CHK-2 (R-X-X-S/T). However, mutation of all four potential sites to nonphosphorylatable residues (*wapl-1*⁴ˢᴬ) did not affect the localization of WAPL-1 in proliferating or meiotic nuclei (Extended Data Fig. 1a–d). Thus, we considered the possibility that regulation of WAPL-1 by CHK-2 might be indirect. We tested whether WAPL-1 downregulation requires the formation of meiotic DSBs, synapsis, and/or homologous pairing, three distinct meiotic events that depend on CHK-2 activity in early prophase[17,35,36]. Loss-of-function mutations[32] or auxin-induced depletion of SPO-11, which is essential for meiotic DSBs, did not alter WAPL-1 localization, indicating that WAPL-1 downregulation is independent of DSBs (Extended Data Fig. 1e–g). WAPL-1 localization was also unaffected when we depleted SYP-3, an essential component of the synaptonemal complex (SC), although chromosome synapsis was disrupted, confirming that depletion was effective (Extended Data Fig. 1e–g)[37,38]. Similarly, WAPL-1 localization was unaffected by co-depletion of PLK-1 and PLK-2, two orthologs of mammalian PLK1 that play partially overlapping roles in chromosome pairing and synapsis during early prophase[32,39,40].

Previous studies in mammalian cells have shown that ATM promotes genome-wide enhancement of cohesin binding in response to ionizing radiation (IR)-induced DNA damage[41]. DDR signaling also governs localized cohesin binding at DNA-damage loci in budding yeast and mammalian cells[42–45]. Moreover, WAPL has been identified as a target of ATM and ATR in *Arabidopsis*[46]. We found that depletion of ATM-1

**Fig. 1 | CHK-2 is required for downregulation of WAPL-1 at meiotic entry. a**, Diagram of a *C. elegans* gonad containing proliferating germline stem cells and meiotic nuclei. The dashed line between red and blue nuclei indicates the boundary between PM and EM cells. Important meiotic events and the behaviors of REC-8 and COH-3/4 cohesins in each stage are shown. **b**, WAPL-1 immunostaining in the distal tip region of gonads, showing the persistence of WAPL-1 on meiotic chromosomes following CHK-2 depletion. Here and elsewhere, 'control^AID' indicates treatment of a negative control strain with auxin in parallel with the experimental strain(s). The control strain is isogenic except that it lacks any degron-tagged genes, that is, it contains the same TIR1 transgene as the experimental strain(s). Red and blue rectangles outline regions containing PM germline stem cell and EM nuclei, respectively, which are enlarged in **c**. Scale bar, 10 μM. Meiotic nuclei but not PM nuclei display chromosomes marked by the HORMA domain protein HTP-3 (magenta). In the

merged images, WAPL-1 (W) is shown in green and HTP-3 (H) is shown in magenta. Scale bar, 2 μM. **d**, Quantification of WAPL-1 immunostaining in **b**. Lower and upper box ends represent the first and third quartiles, with the median indicated by the horizontal line within the box. All data points are shown, and the sample sizes are indicated. ****$P < 0.0001$ (two-sided Wilcoxon–Mann–Whitney test, adjusted by Bonferroni correction). a.u., arbitrary units. **e**, COH-3/4 and REC-8 immunolocalization (in green) in early pachytene nuclei, showing that WAPL-1 negatively regulates axial COH-3/4, but not REC-8, upon CHK-2 depletion. Scale bar, 2 μM. **f,g**, Quantification of the intensity of COH-3/4 and REC-8 immunostaining in auxin-treated animals of the indicated genotypes. **e**. Lower and upper box ends represent the first and third quartiles, with the median indicated by the horizontal line within the box. All data points are shown, and the sample sizes are indicated. ****$P < 0.0001$ (two-sided Wilcoxon–Mann–Whitney test, adjusted by Bonferroni correction). See 'Data presentation' for more details.

(ATM) in the *C. elegans* germline abolished WAPL-1 downregulation, whereas depletion of ATL-1 (ATR) had no effect on WAPL-1. Importantly, ATL-1 depletion phenocopied the effects of an *atl-1*-null mutation on the size and number of germline nuclei (Fig. 2b–d and Extended Data Fig. 2g)[47]. Alignment of WAPL-1 homologs also revealed a small but conserved cluster of S/T-Q residues (Fig. 2a). S/T-Q cluster domains

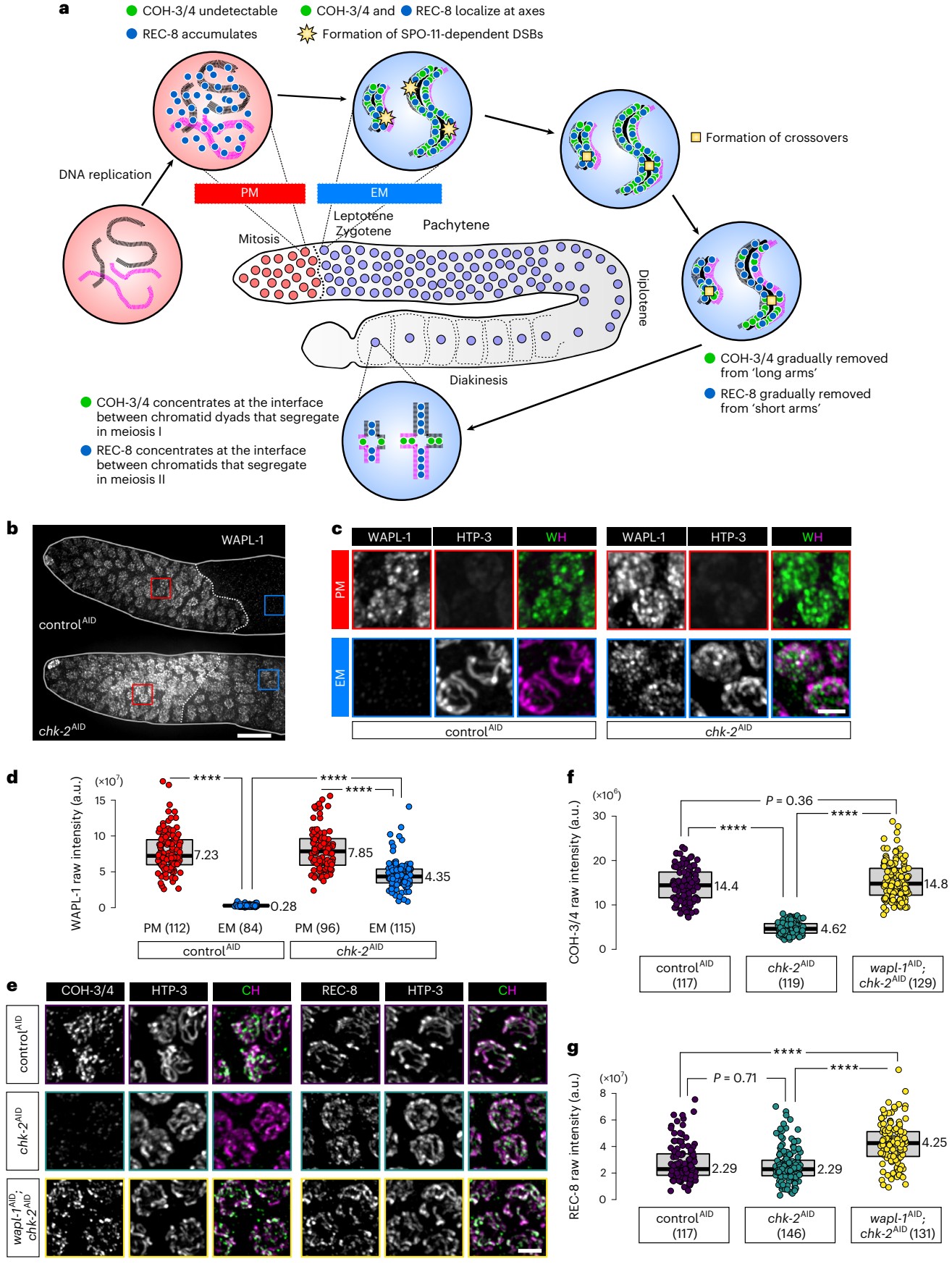

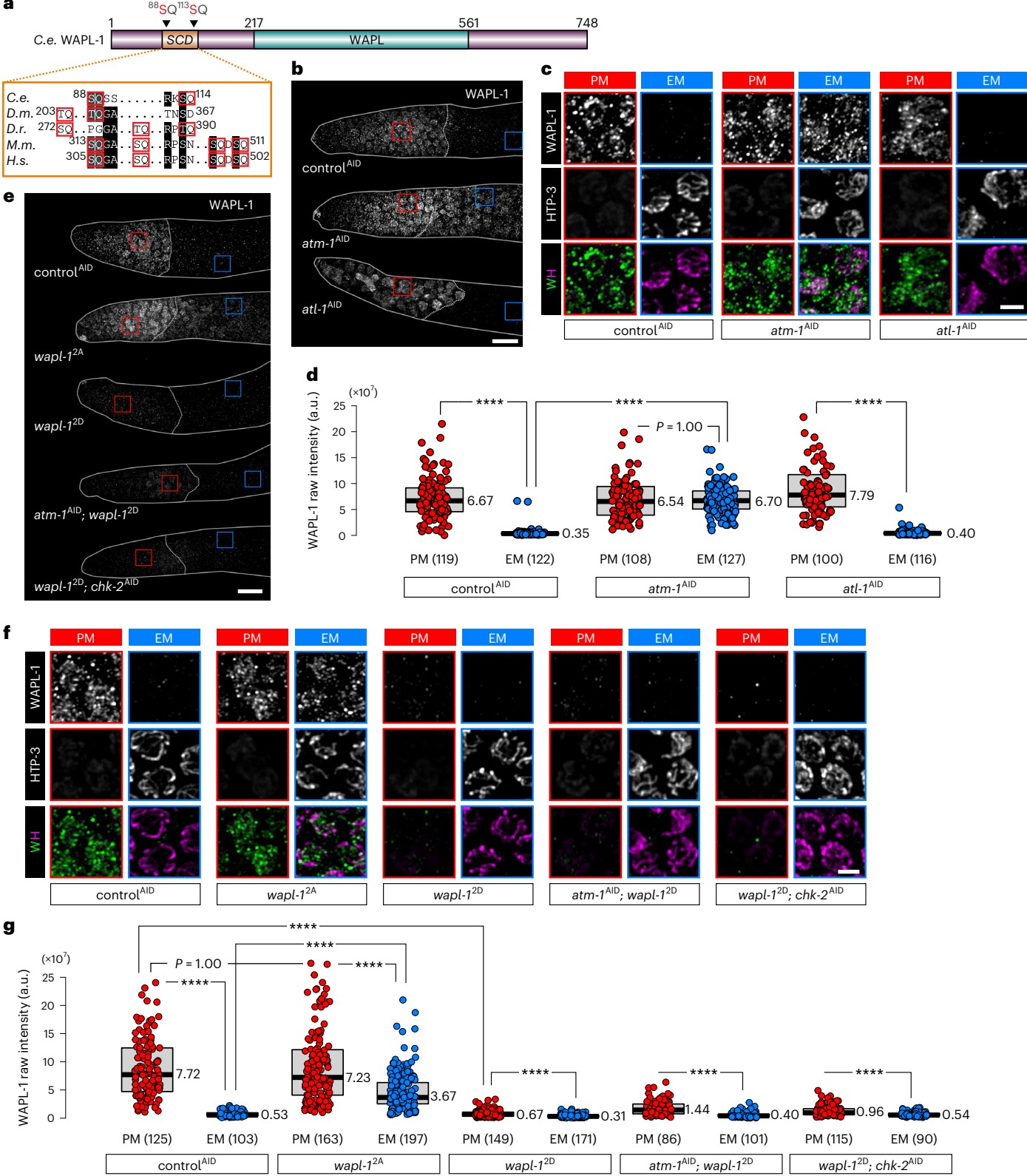

**Fig. 2 | WAPL-1 suppression depends on ATM-1 and a small target domain (mini-SCD). a**, Diagram illustrating the domain architecture of *C. elegans* WAPL-1. The N-terminal mini-SCD identified in this study and the two putative phosphorylation sites are shown. Partial alignment of amino acid sequences of corresponding regions of Wapl orthologs show conservation of this mini-SCD across species (*C.e.*, *Caenorhabditis elegans*; *D.m.*, *Drosophila melanogaster*; *D.r.*, *Danio rerio*; *M.m.*, *Mus musculus*; *H.s.*, *Homo sapiens*). SQs and TQs are highlighted and outlined. **b,e**, WAPL-1 immunostaining in the distal tip of gonads, showing that ATM-1 and WAPL-1 mini-SCDs are essential for WAPL-1 downregulation at meiotic entry. Scale bars, 10 μM. **c,f**, Enlarged images showing WAPL-1 immunostaining (in green) in PM nuclei and EM nuclei from **b** and **e**. HTP-3 is recruited to axes at meiotic entry. Scale bar, 2 μM. **d,g**, Quantification of the intensity of WAPL-1 immunostaining in **b** and **e**. Lower and upper box ends represent the first and third quartiles, with the median indicated as the horizontal line within the box. All data points are shown, and the sample sizes are indicated. ****$P < 0.0001$ (two-sided Wilcoxon–Mann–Whitney test, adjusted by Bonferroni correction).

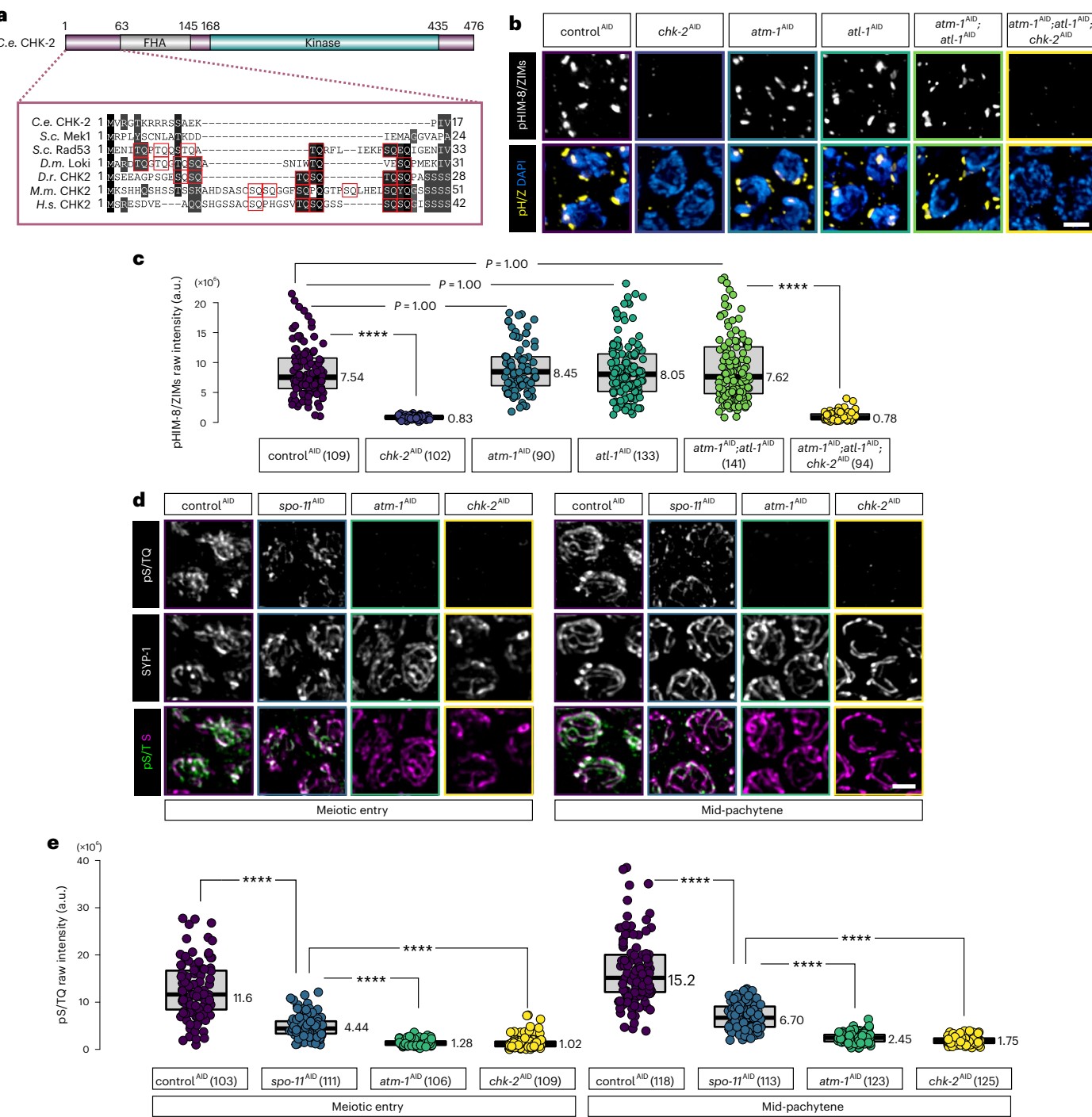

**Fig. 3 | CHK-2 activates ATM-1 to suppress WAPL-1 during EM prophase.**
**a**, Domain architecture of *C. elegans* CHK-2, showing the forkhead-associated (FHA) and kinase domains. Partial alignment of the N termini of several Rad53/CHK2 orthologs is shown below the schematic, with all S/TQs highlighted. *C. elegans* CHK-2 and *S. cerevisiae* Mek1 are meiosis-specific proteins that lack an N-terminal SCD. **b**, Immunofluorescence using a phospho-specific antibody that recognizes CHK-2 target motifs in HIM-8 and ZIM-1, ZIM-2, and ZIM-3 (pHIM-8/ZIM, or pH/Z), showing that CHK-2 activity is independent of ATM-1 and ATL-1. DAPI-stained DNA highlights meiotic nuclei. Scale bar, 2 μM. **c**, Quantification of pHIM-8/ZIM immunostaining in **b**. Lower and upper box ends represent the first and third quartiles, with the median indicated by the horizontal line within

the box. All data points are shown, and the sample sizes indicated. ****$P < 0.0001$ (two-sided Wilcoxon–Mann–Whitney test, adjusted by Bonferroni correction). **d**, pS/TQ immunostaining (in green) of EM nuclei (left) or mid-pachytene nuclei (right) under the indicated conditions, showing that ATM-1 activity depends on CHK-2. SYP-1 immunostaining (in magenta) shows the SC. Scale bar, 2 μM. **e**, Quantification of the intensity of pS/TQ immunostaining in **d**. Lower and upper box ends represent the first and third quartiles, with the median indicated by the horizontal line within the box. All data points are shown with the sample sizes indicated. ****$P < 0.0001$ (two-sided Wilcoxon–Mann–Whitney test, adjusted by Bonferroni correction).

(SCDs), defined as segments of 100 or fewer amino acids containing 3 or more S/T-Q motifs, are found in many substrates of ATM or ATR[48]. Although the two SQ motifs in *C. elegans* WAPL-1 do not meet this strict

definition of an SCD, we tested the function of this putative mini-SCD by replacing the serines with nonphosphorylatable (*wapl-1*[2A]) or phosphomimetic (*wapl-1*[2D]) amino acids (Fig. 2a). WAPL-1[2A] showed defective

downregulation (Fig. 2e), despite the activity of CHK-2 (Extended Data Fig. 2a–c) and ATM-1 (Extended Data Fig. 2d–f). By contrast, phosphomimetic WAPL-1[2D] was reduced on chromatin both before and after meiotic entry (Fig. 2e). Localization of WAPL-1[2D] was not restored by depletion of CHK-2 or ATM-1 (Fig. 2e–g). These results support the idea that ATM-1 inhibits the association of WAPL-1 with chromatin at meiotic entry by phosphorylating these SQ motifs.

## CHK-2 positively regulates ATM-1 activity

In the canonical DDR pathway in mammalian cells, CHK2 is an essential downstream transducer of ATM activity[49,50]. However, previous studies have found that *C. elegans* CHK-2 is dispensable for checkpoint activation in response to hydroxyurea and ionizing radiation in embryos and the adult germline, and is essential only for meiosis[35,36]. Like other meiosis-specific CHK2 orthologs, *C. elegans* CHK-2 lacks the amino-terminal SCD that mediates activation by ATM (Fig. 3a). Additionally, CHK-2-dependent phosphorylation of the nuclear envelope protein SUN-1 was observed in the absence of ATM-1 and ATL-1 (ref. [51]). We further found that pairing center proteins HIM-8 and ZIM-1, ZIM-2, and ZIM-3, which are direct targets of CHK-2 (ref. [17]), were phosphorylated when we depleted ATM-1, ATL-1, or both proteins (Fig. 3b,c), confirming that CHK-2 activity is independent of ATM-1 and ATL-1.

Since WAPL-1 persists on chromatin following depletion of either CHK-2 or ATM-1, we tested whether ATM-1 activity might depend on CHK-2. The intensity of immunostaining using an antibody against phosphorylated SQ/TQ (pS/TQ)[52], which recognizes ATM/ATR substrates, was greatly reduced in meiotic nuclei following depletion of either CHK-2 or ATM-1 (Fig. 3d).

Formation of meiotic DSBs in *C. elegans* requires CHK-2 (refs. [35,53,54]). We tested whether ATM-1 activity depends on DSBs by depleting the SPO-11 endonuclease, and found that this also reduced pS/TQ immunofluorescence, albeit less so than depletion of ATM-1 (Figs. 3d,e). This is consistent with our evidence that SPO-11 depletion does not affect WAPL-1 downregulation at meiotic entry (Extended Data Fig. 1a–c). Together, these results indicate that CHK-2 promotes basal levels of ATM-1 activity in the absence of meiotic DSBs.

## A CHK-2 consensus site is essential for ATM-1 activity

Previous studies have shown that high concentrations of Chk2 can lead to self-activation in vitro and phosphorylation of H2AX and other S/T-Q sites in vivo[55,56], suggesting that Chk2 may promote ATM/ATR activity under certain conditions. We aligned the amino acid sequences of ATM family proteins and found a conserved CHK-2 consensus phosphorylation motif (R-X-X-S/T) within their FAT domains, which are critical for ATM/ATR activation (Fig. 4a)[57–59]. A mutation in this motif was identified in a person with leukemia with ATM deficiency[60].

Mutation of the arginine and serine residues in this motif (*atm-1*[KA]) greatly reduced pS/T-Q immunofluorescence in meiotic nuclei (Fig. 4a–c and Extended Data Fig. 3), while the corresponding phosphomimetic mutation did not appear to alter ATM-1 activity (Fig. 4b,b). Additionally, phosphomimetic ATM-1[S1853D] showed detectable, albeit reduced, activity, even when CHK-2 was depleted (Fig. 4b–d). Depletion of either CHK-2 or SPO-11 in animals expressing ATM-1[S1853D] resulted in similar levels of anti-pS/TQ immunofluorescence (Fig. 4e,f). Together, these observations suggest that break-independent phosphorylation of S1853 by CHK-2 promotes a basal level of ATM-1 activity, and DSBs further elevate its activity.

Together, these findings indicate that the persistence of WAPL-1 on meiotic chromosomes upon CHK-2 depletion may be a consequence of a failure to activate ATM-1. To further test this idea, we examined WAPL-1 immunostaining in *atm-1*[KA] and *atm-1*[S1853D], alleles that showed defective and normal kinase activity, respectively, in early meiosis. CHK-2 showed normal activity in both cases (Fig. 4b–d). Animals expressing only the nonphosphorylatable *atm-1*[KA] allele showed aberrant WAPL-1 staining, similar to that seen following depletion of CHK-2 or ATM-1

(Fig. 5a–c). By contrast, the phosphomimetic *atm-1*[S1853D] allele resulted in normal downregulation of WAPL-1, even when CHK-2 was depleted (Fig. 5a–c). Moreover, nonphosphorylatable WAPL-1[2A] persisted on EM chromosomes in *atm-1*[S1853D], indicating that ATM-1[S1853D] regulates WAPL-1 through its N-terminal mini-SCD (Extended Data Fig. 5d,e). We tested whether *atm-1*[S1853D] could bypass the requirement for CHK-2 in DSB induction, and found that RAD-51 foci were absent, indicating that CHK-2 promotes breaks independently of ATM activation (Extended Data Fig. 4a,b), which also validated the depletion efficacy of SPO-11.

Together, our results indicate that CHK-2 activates ATM even in the absence of DSBs (Fig. 3d,e), and that this CHK-2-dependent ATM-1 activity downregulates WAPL-1, resulting in stabilization of cohesins along chromosomes. Consistent with this interpretation, phosphomimetic mutations in ATM-1 or WAPL-1 restored robust axis localization of COH-3/4 when CHK-2 was depleted (Fig. 5d,e).

## CHK-2 synergizes with cohesin acetylation to promote axis assembly

The evidence above indicates that CHK-2 regulates COH-3/4 localization through ATM-1 and WAPL-1. However, loss of CHK-2 reduced COH-3/4 along the axis more dramatically than loss of ATM-1 or expression of nonphosphorylatable WAPL-1 (Extended Data Fig. 5a), suggesting that CHK-2 may also stabilize cohesins through other mechanisms. Our evidence that COH-3/4 localization is fully restored by WAPL-1 depletion in *chk-2* mutants suggests that CHK-2 may promote an activity that antagonizes WAPL-1. Sororin in vertebrates and *Drosophila*[61–63], and cohesin acetylation by Eco1, ESCO1, ESCO2, or ECO-1 (refs. [64–67]), are both known to counteract cohesin destabilization by WAPL in other contexts. We thus investigated the role of the likely *C. elegans* Eco1 ortholog F08F8.4 (now ECO-1).

Depletion of ECO-1 alone did not reduce COH-3/4 localization (Fig. 6a–c). However, co-depletion of ECO-1 and ATM-1, or depletion of ECO-1 in *wapl-1*[2A], showed additive effects on COH-3/4 localization, nearly recapitulating the effects of CHK-2 depletion (Fig. 6a–c and Extended Data Fig. 5a–c). These results suggest that ECO-1 can antagonize WAPL-1 in early meiosis, but that WAPL-1 downregulation normally makes this unnecessary.

Previous studies have shown that Eco1/Eso1/ESCO1/2 antagonizes Wapl-dependent cohesin release by acetylating cohesin subunits[68,69], including two conserved lysine sites on the ATPase head of Smc3 (refs. [19,23,26,27]). We mutated the corresponding lysines in *C. elegans* SMC-3 to glutamine to mimic acetylation (K106Q K107Q; *smc-3*[QQ]). Axial COH-3/4 localization and axis morphogenesis in early meiosis appeared normal in *smc-3*[QQ] mutants (Fig. 6d). These mutations partially restored COH-3/4 localization upon CHK-2 depletion (Fig. 6d,e), suggesting that CHK-2 activity may promote acetylation of SMC-3 (see 'Discussion').

## PDS-5 protects REC-8 from WAPL-mediated release

Prior work and our observations indicate that WAPL-1 has a greater impact on COH-3/4 than on REC-8 localization (Fig. 1e–g)[31]. However, Wapl can promote release of Rec8 cohesin in yeast, plant, and human meiocytes[70–73]. Therefore, we wondered why *C. elegans* REC-8 cohesin is more resistant to WAPL-1 activity than COH-3/4 during meiotic prophase.

The cohesin regulator Spo76/EVL-14/Pds5/PDS-5 is required to establish and maintain sister chromatid cohesion in mitosis and meiosis[74–77]. Intriguingly, Pds5 can either recruit Wapl to release cohesin or prevent Wapl from accessing cohesin, depending on the context[61,66,74,76,78]. Importantly, chromosome condensation defects seen in budding yeast Pds5 mutants are rescued by loss of Rad61/Wpl1 (Wapl), suggesting a direct antagonism between Pds5 and Wapl[79,80].

PDS-5 (also known as EVL-14) is essential for gonad development in *C. elegans*, presumably owing to its mitotic functions[77]. The protein localizes to nuclei throughout the germline and is enriched along chromosome axes in meiotic nuclei (Extended Data Fig. 6a,b). Its abundance

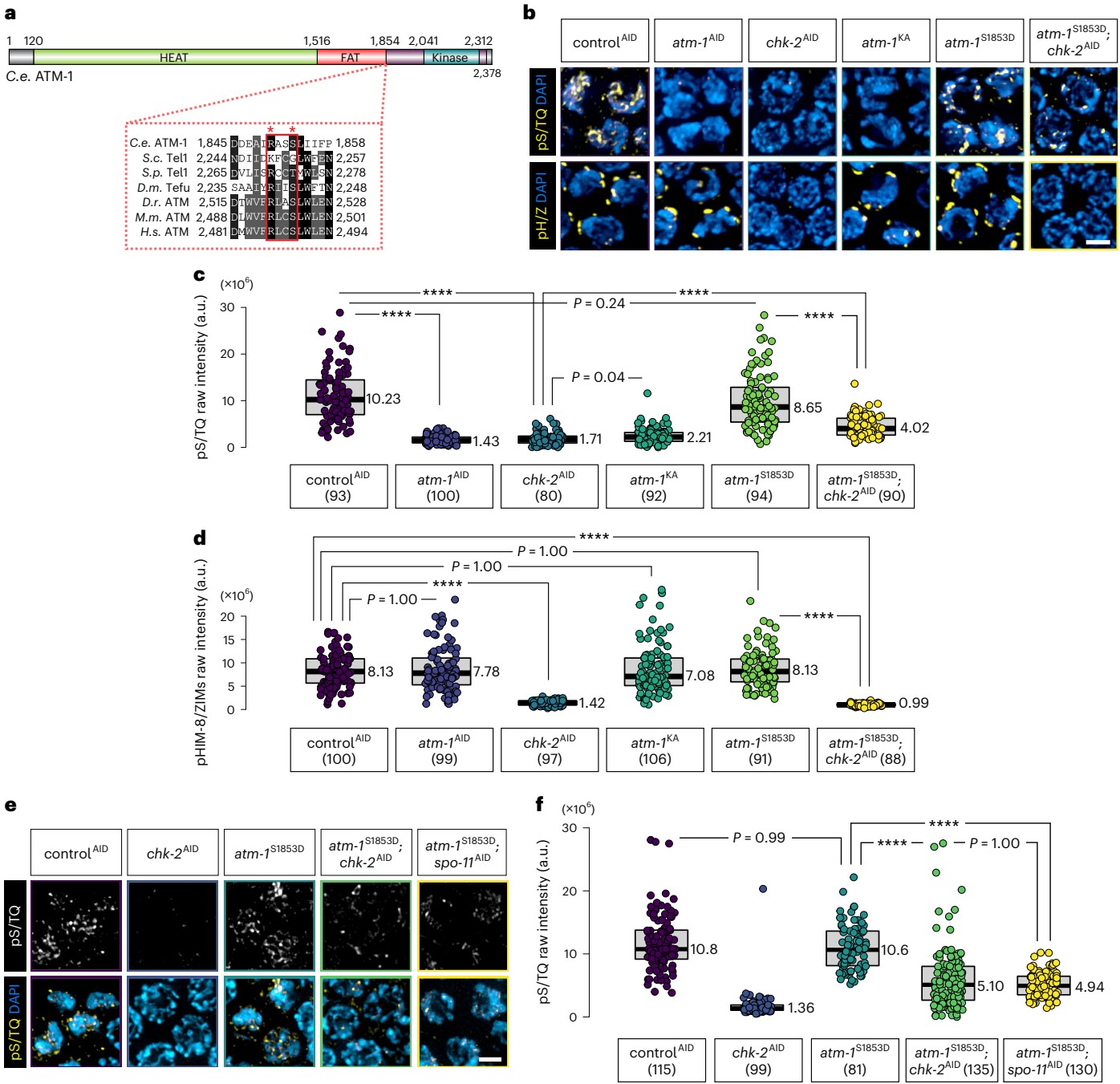

**Fig. 4 | A CHK-2 consensus motif in the FAT domain of ATM-1 mediates ATM-1 activity. a**, Domain architecture of *C. elegans* ATM-1, including the HEAT (Huntington, Elongation Factor 3, PR65/A, and TOR) and FAT (FRAP, ATM, and TRRAP) domains. An alignment of the C-terminal end of FAT domains from several ATM orthologs is shown below the schematic, with the Rad53/CHK2 consensus motif outlined. The conserved arginine and the putative phospho-serine/threonine site of the consensus motif are indicated with asterisks. *S.p.*, *Schizosaccharomyces pombe*. **b**, A phosphomimetic mutation in ATM-1 results in CHK-2-independent activity in EM nuclei. pS/TQ immunostaining is used as a proxy for ATM-1/ATL-1 activity, while phosphorylation of conserved motifs on HIM-8 and the ZIM proteins is indicative of CHK-2 activity. DAPI-stained DNA highlights meiotic nuclei. Scale bar, 2 μM. **c**,**d**, Quantification of pS/TQ immunofluorescence intensity (**c**) and pHIM-8/ZIM intensity (**d**) (see 'Data presentation' for more details). Lower and upper box ends represent the first and third quartiles, with the median indicated by the horizontal line within the box. All data points are shown, and the sample sizes are indicated. ****$P < 0.0001$ (two-sided Wilcoxon–Mann–Whitney test, adjusted by Bonferroni correction). **e**, pS/TQ immunostaining shows comparable kinase activity of ATM-1[S1853D] upon depletion of CHK-2 and SPO-11. Scale bar, 2 μM. **f**, Quantification of pS/TQ immunofluorescence in **e**. Lower and upper box ends represent the first and third quartiles, with the median indicated by the horizontal line within the box. All data points are shown, and the sample sizes are indicated. ****$P < 0.0001$ (two-sided Wilcoxon–Mann–Whitney test, adjusted by Bonferroni correction).

on axes was not affected by loss of CHK-2 (Extended Data Fig. 6a–c), despite the reduction in COH-3/4 (above), supporting the idea that PDS-5 preferentially associates with REC-8 cohesin.

Depletion of PDS-5 did not affect the abundance of REC-8 in PM nuclei, much of which is likely nucleoplasmic, but in early meiosis REC-8 was markedly reduced along axes (Fig. 7a,b,e). This

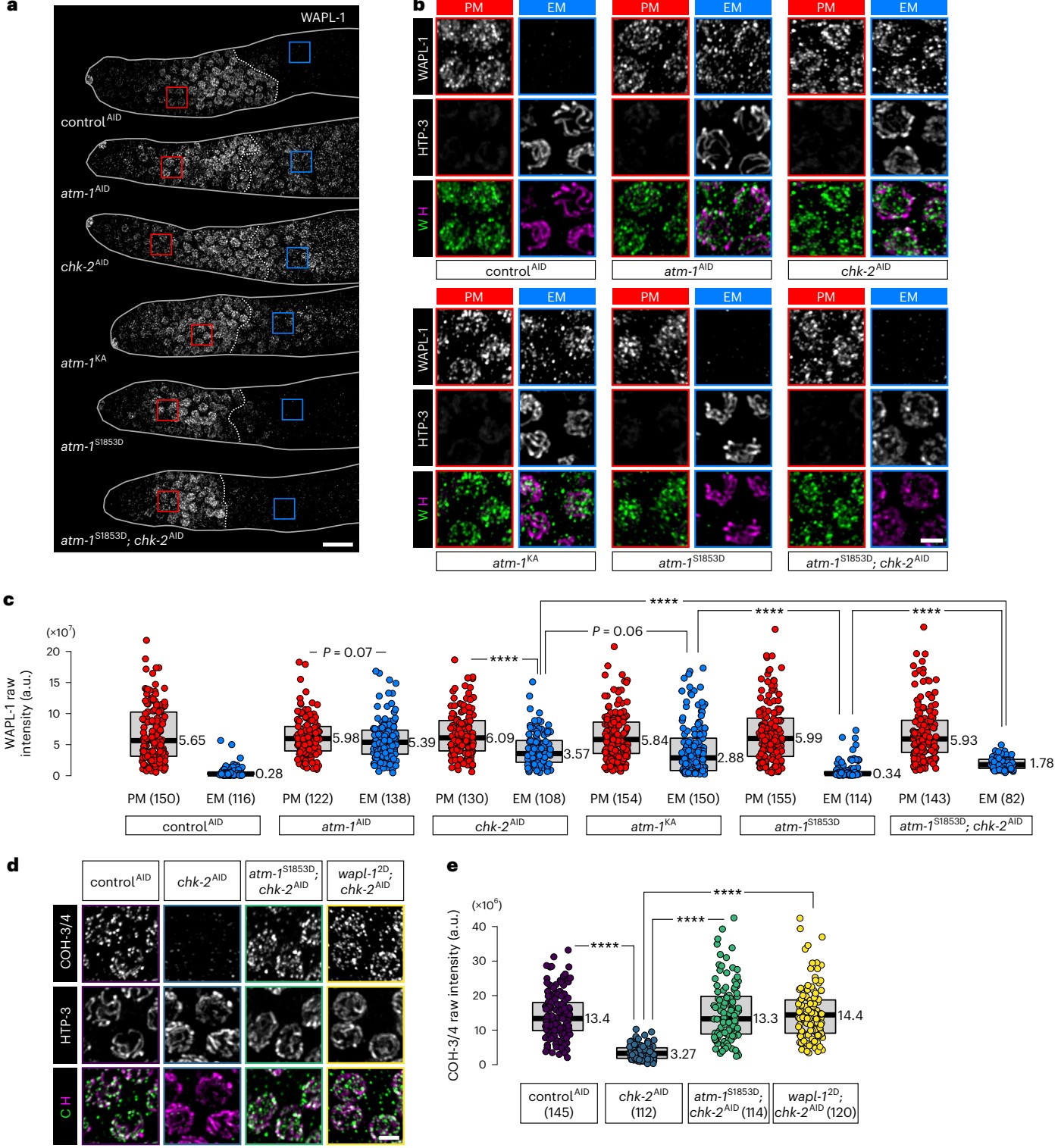

**Fig. 5 | CHK-2 suppresses WAPL-1 by activating ATM-1. a**, A phosphomimetic mutation in ATM-1 is sufficient to suppress WAPL-1 at meiotic entry upon CHK-2 depletion. Scale bar, 10 μM. **b**, Enlarged images of the regions boxed in **a**. HTP-3 localizes to axes starting at meiotic entry. Scale bar, 2 μM. **c**, Quantification of the intensity of WAPL-1 immunostaining in **a**. Lower and upper box ends represent the first and third quartiles, with the median indicated by the horizontal line within the box. All data points are shown, and the sample sizes are indicated. ****$P < 0.0001$ (two-sided Wilcoxon–Mann–Whitney test, adjusted by Bonferroni

correction). **d**, COH-3/4 immunostaining (in green) of early pachytene nuclei, showing that either mimicking ATM-1 activation or WAPL-1 phosphorylation is sufficient to restore COH-3/4 localization upon CHK-2 depletion. Scale bar, 2 μM. **e**, Quantification of the intensity of COH-3/4 immunostaining in **d**. Lower and upper box ends represent the first and third quartiles, with the median indicated by the horizontal line within the box. All data points are shown, and the sample sizes are indicated. ****$P < 0.0001$ (two-sided Wilcoxon–Mann–Whitney test, adjusted by Bonferroni correction).

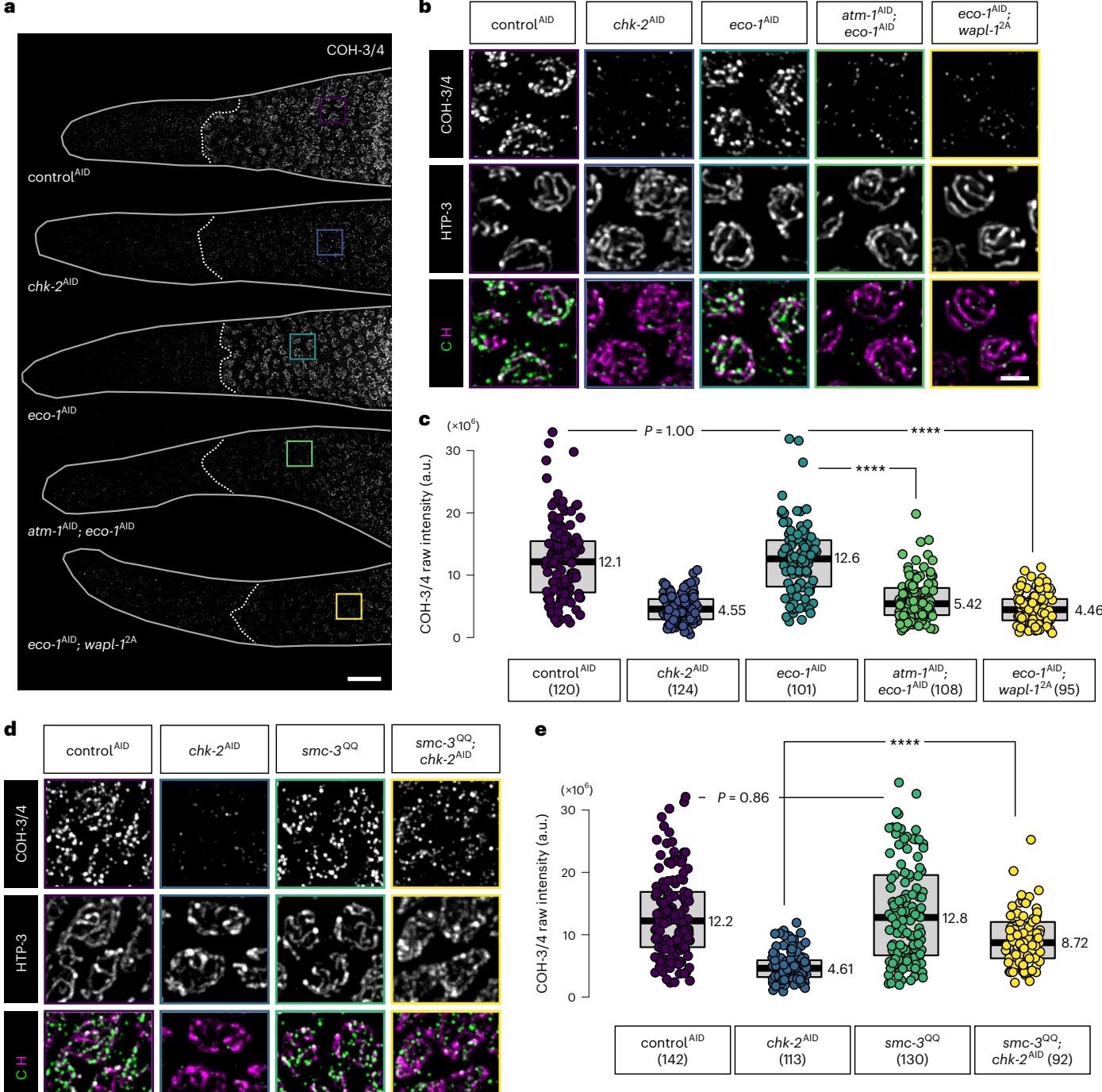

**Fig. 6 | ECO-1-dependent cohesin acetylation contributes to stabilization of axial cohesin against WAPL-1. a**, ECO-1 is required for robust COH-3/4 localization in EM nuclei if ATM-1-dependent WAPL-1 phosphorylation is defective. Scale bar, 10 μM. **b**, Enlarged views of the regions indicated in **a**. Bright HTP-3 staining along axes is detected starting at meiotic entry. Scale bar, 2 μM. **d**, COH-3/4 immunostaining (in green) of meiotic entry nuclei, showing that the SMC-3 acetylation-mimetic mutation antagonizes WAPL-1-dependent COH-3/4 release upon CHK-2 depletion. Scale bar, 2 μM. **c**,**e**, Quantification of the intensity of COH-3/4 immunostaining in **b** and **d**, respectively. Lower and upper box ends represent the first and third quartiles, with the median indicated by the horizontal line within the box. All data points are shown, and the sample sizes are indicated. ****$P < 0.0001$ (two-sided Wilcoxon–Mann–Whitney test, adjusted by Bonferroni correction).

is similar to findings in fission yeast but contrasts with observations from budding yeast and mammals, where loss of Pds5 has little effect on Rec8 binding to chromosomes[81–85]. By contrast, the localization of COH-3/4 cohesin was unaltered by depletion of PDS-5, supporting the idea that PDS-5 preferentially stabilizes REC-8 (Fig. 7c,d,f).

Co-depletion of CHK-2 and PDS-5 resulted in dramatic reduction of REC-8 along chromosome axes (Fig. 7g), although the abundance of REC-8 in PM cells was again unaffected (Fig. 7g,h). Importantly, upon co-depletion of PDS-5 and CHK-2, REC-8 localization was fully rescued by co-depletion of WAPL-1, indicating that PDS-5 protects REC-8 from release by WAPL-1 (Fig. 7g–i and Extended Data Fig. 6e,f).

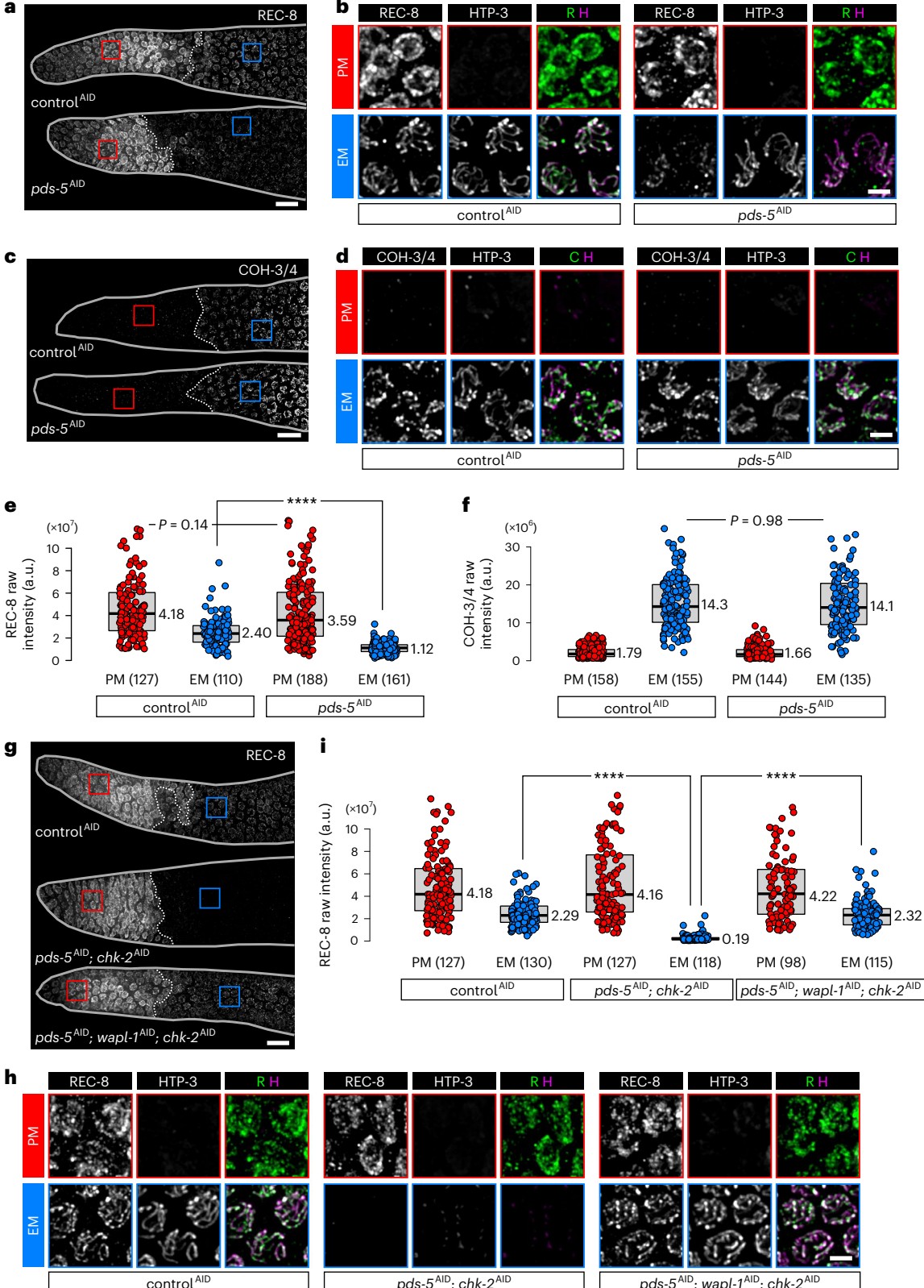

**Fig. 7 | PDS-5 protects REC-8 cohesin from WAPL-1-dependent release in early meiosis. a,c,g**, REC-8 (**a** and **g**) and COH-3/4 (**c**) immunofluorescence in the distal tip of gonads, showing that WAPL-1 can release REC-8 cohesin upon co-depletion of PDS-5 and CHK-2. Scale bar, 10 μM. **b,d,h**, Enlarged views of the regions indicated in **a**, **c**, and **g**, respectively. Scale bar, 2 μM. **e,f,i**, Quantification of the intensity of REC-8 (**e** and **i**) and COH-3/4 (**f**) immunostaining. Lower and upper box ends represent the first and third quartiles, with the median indicated by the horizontal line within the box. All data points are shown, and the sample sizes are indicated. ****$P < 0.0001$ (two-sided Wilcoxon–Mann–Whitney test, adjusted by Bonferroni correction).

The HORMA domain protein HTP-3 can be recruited to axes in the absence of either REC-8 or COH-3/4 cohesin, but not both[12]. Co-depletion of CHK-2 and PDS-5 resulted in loss of HTP-3 from axes (Fig. 7g–i), corroborating the conclusion that both classes of cohesin were severely disrupted[12]. Localization of REC-8 and HTP-3 was restored by co-depletion of WAPL-1 (Fig. 7g–i). Together, these results reveal that downregulation of WAPL-1 and protection of REC-8 by PDS-5 are parallel pathways that contribute to cohesin stability and axis assembly at meiotic entry. Acetylation of cohesins by ECO-1 likely also contributes to axis formation, but this is only evident when WAPL-1 cannot be properly downregulated.

## Cohesin enrichment at DNA damage foci is regulated by WAPL

Our identification of a conserved mini-SCD in Wapl homologs prompted us to explore whether WAPL downregulation is important in contexts other than meiosis. Studies in yeast and human cells have shown that the DDR kinases ATM and ATR mediate cohesin enrichment at DNA-damage loci, which promotes repair[41,44,86]. ATM has also been found to promote the enrichment of CTCF, a cohesin binding partner, at DNA-damage sites in human cells[87]. We thus investigated whether the establishment of cohesin-enriched domains at DNA-damage foci depends on ATM activity and/or WAPL downregulation.

We treated HeLa cells with DNA-damage-inducing agents, including the radiomimetic DNA-cleaving agent bleomycin and the topoisomerase II poison etoposide (ETO), both of which lead to DSBs that activate ATM. Following treatment with either agent, we observed foci that were positive for both γH2A.X (S139-phosphorylated histone H2A.X) (Fig. 8a)[88,89] and pS/TQ (Fig. 8b)[52]. The mitotic kleisin Rad21 was concentrated at many of these damage foci (Fig. 8a–d).

We next induced DNA damage after treating cells with specific inhibitors of ATM, ATR, and DNA-PK (Fig. 8e), three paralogous DNA-damage transducer kinases. Inhibition of ATM, but not ATR or DNA-PK, resulted in markedly reduced γH2A.X upon ETO treatment (Fig. 8e and Extended Data Fig. 7a). Inhibition of ATM also largely eliminated damage-induced Rad21 foci (Fig. 8e,i).

To corroborate the specificity of the chemical inhibitors, we performed short interfering RNA (siRNA)-mediated knockdown of ATM and ATR (Fig. 8f). We confirmed knockdown of ATM using a commercial antibody (Extended Data Fig. 7c,d). Although we lacked a similar tool to monitor ATR abundance, we noted that nuclear volume was reduced by either ATR inhibition or ATR knockdown (Extended Data Fig. 7b,g). Consistent with our observations with small-molecule inhibitors, ATM knockdown resulted in a substantial decrease in nuclear γH2A.X intensity (Fig. 8f and Extended Data Fig. 7f). Importantly, Rad21 was no longer enriched at damage foci marked by either γH2A.X or pS/TQ (Fig. 8f,j and Extended Data Fig. 7e). Together, these results indicate

that the activity of ATM, but not ATR or DNA-PK, is required for cohesin enrichment at DNA-damage foci.

We next tested whether WAPL downregulation contributes to the assembly of these cohesin-enriched domains. We reasoned that if ATM promotes the formation of cohesin foci by phosphorylating WAPL, overexpression of nonphosphorylatable WAPL might impair focus formation. Human WAPL has two potential mini-SCDs (Fig. 8g). Importantly, neither overlaps with the FGF or YSR motifs, which mediate the interaction between WAPL and PDS5 (refs. 24,90). Overexpression of GFP-tagged wild-type WAPL or mutant proteins did not affect the appearance of γH2A.X following DNA damage (Fig. 8g–h), indicating that ATM signaling was not perturbed (Fig. 8h and Extended Data Fig. 7h). Overexpression of wild-type WAPL also did not significantly affect nuclear RAD21 foci following damage (Fig. 8h,k). However, overexpression of nonphosphorylatable GFP-WAPL (WAPL^5A) blocked the formation of RAD21 foci very effectively, whereas GFP-WAPL^5D lacked this activity (Fig. 8h,k). To test whether the reduction of RAD21 foci in cells expressing GFP-WAPL^5A is due to WAPL-dependent cohesin release, we overexpressed a nonphosphorylatable WAPL protein lacking four amino acids (1116MEDC1119) that are critical for release activity (Fig. 8g)[91,92]. Although this mutant protein (WAPL^5AΔ4) localized to nuclei, it had no effect on RAD21 foci following DNA damage (Fig. 8h,k). Interestingly, we also found that GFP-WAPL^5AΔ4 overexpression caused nucleus-wide clustering of cohesin, similar to the 'vermicelli' phenotype, although the cohesin threads appeared to be much thinner (Fig. 8h)[33], suggesting that this protein may act in a dominant negative fashion. Taken together, these results indicate the role of both ATM activity and WAPL mini-SCD in regulating local enrichment of cohesin at damage foci in mammalian cultured cells, similar to results for *C. elegans* meiotic chromosome axes.

## Discussion

We have found that downregulation of WAPL by ATM promotes cohesin localization along meiotic chromosome axes in *C. elegans* and at DNA-repair foci in mammalian cells (Extended Data Fig. 8). A key function of meiotic axes is to regulate the outcome of repair of induced DSBs[93,94], so it makes teleological sense that this assembly would be regulated by DDR signaling[95]. Our findings illuminate the role of cohesin regulators and how they are orchestrated by DDR during the unique cell cycle state of meiotic prophase. Expression of meiosis-specific cohesins is necessary but not sufficient for axis formation and function[96,97]. We find that CHK-2 promotes break-independent activation of ATM-1 at meiotic entry, which in turn promotes axis assembly through downregulation of WAPL-1, a key regulator of cohesin dynamics[21,22]. Previous studies have shown that WAPL reduces cohesin residence time on chromatin and modulates cohesin clustering and cohesin-dependent loop extrusion[33,98]. We find that downregulation of WAPL-1 is important

**Fig. 8 | ATM-mediated WAPL downregulation regulates cohesin concentration at DNA-damage foci. a,b,** Immunostaining of phosphorylated H2A.X (γH2A.X) and RAD21 in nuclei of HeLa cells that were treated with either bleomycin (BLEO) or ETO, showing the concentration of cohesin at DNA-damage foci. DMSO was used as a solvent control. In the merged images, 'R' indicates RAD21 and 'γH' indicates γH2A.X. Scale bar, 10 μM. **c,d,** Quantification of the number of RAD21 foci under conditions shown in **a** and **b**. Lower and upper box ends represent the first and third quartiles, with the median indicated by the horizontal line within the box. All data points are shown, and the sample sizes are indicated. ****P < 0.0001 (two-sided Wilcoxon–Mann–Whitney test, adjusted by Bonferroni correction). **e,** Immunostaining of γH2A.X and RAD21 in nuclei of HeLa cells treated with kinase inhibitors, followed by ETO to induce DNA damage, showing that ATM activity is required for cohesin concentration at DNA-damage foci. The chemical inhibitors, which affected specific kinases, were as follows: KU55933 (ATMi), VE-821 (ATRi), and NU7441 (DNA-PKi). The control group (CONi) was treated with DMSO. Scale bar, 10 μM. **f,** RAD21 immunostaining in nuclei of HeLa cells depleted of ATM

and/or ATR prior to ETO-induced DNA damage, showing that ATM is required for cohesin concentration at DNA damage foci. siRNAs specific for each kinase were used for knockdown. A non-targeting siRNA pool (Methods) was used for control knockdowns (siCON). Scale bar, 10 μM. **g,** Domain architecture of human WAPL, indicating the positions of the YSR (tyrosine-serine-arginine) motif, FGF (phenylalanine-glycine-phenylalanine) motifs, the two SCD domains, the MEDC (methionine-glutamate-aspartate-cysteine) sequence, and the residues that were mutated in our transgenic constructs. **h,** Immunofluorescence of HeLa cell nuclei expressing GFP or GFP-WAPL, showing that overexpression of nonphosphorylatable WAPL inhibits cohesin concentration at DNA-damage foci. Scale bar, 10 μM. **i,j,k,** Quantification of the number of RAD21 foci in **e**, **f**, and **h**, as described in 'Image analysis' and 'Data presentation'. Lower and upper box ends represent the first and third quartiles, with the median indicated by the horizontal line within the box. All data points are shown, and the sample sizes are indicated. ****P < 0.0001 (two-sided Wilcoxon–Mann–Whitney test, adjusted by Bonferroni correction).

for meiotic axis assembly. This is likely a conserved feature of meiosis, as loop anchors emerge along with axis compaction and reduced cohesin dynamics during meiosis in many species[33,98–100].

Our findings also reveal that phosphorylation by ATM inhibits WAPL. A role for ATM in meiotic axis morphogenesis has also been demonstrated in *Arabidopsis*[101]. ATM and ATR localize sequentially to chromosome axes during EM prophase in mice, consistent with roles in regulating cohesin along the axis[102]. Our results

also reveal a key function for this regulatory pathway in the DDR in proliferating cells[103,104].

We also find that the acetyltransferase ECO-1 contributes to the stability of axial cohesins, although this was apparent only when downregulation of WAPL was defective. This is consistent with observations from yeast, *Drosophila*, and *Arabidopsis*[64,105,106]. Our results suggest that ECO-1 may also be regulated by CHK-2. In budding yeast, phosphorylation of the mitotic kleisin Mcd1 by Chk1 promotes Eco1-dependent Mcd1 acetylation,

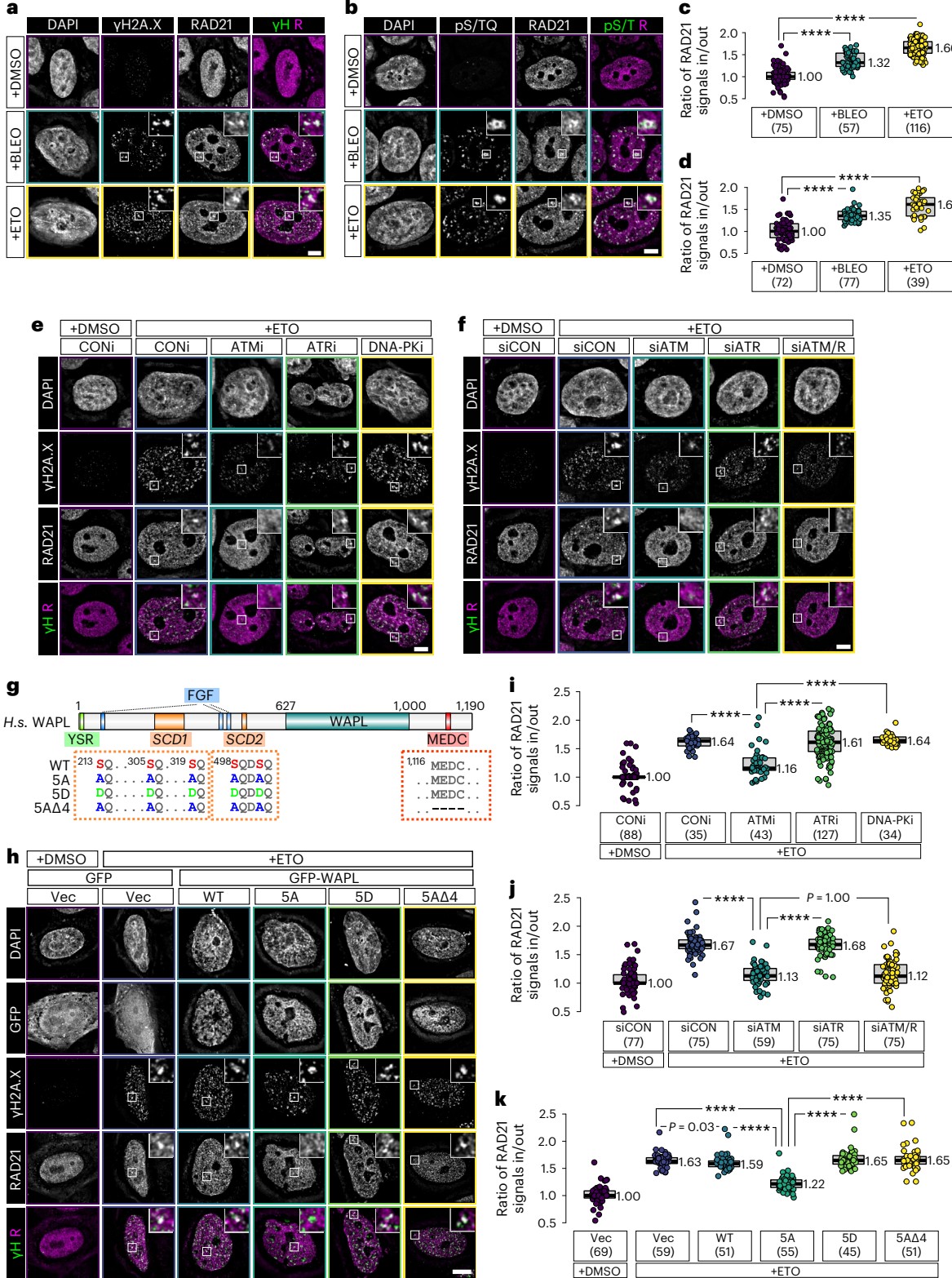

which in turn antagonizes Wapl and promotes cohesion[68,107]. Analogous regulation of *C. elegans* COH-3/4 by CHK-2 has also been proposed[12].

Studies across diverse eukaryotes have established critical roles for Pds5 in regulating cohesin-dependent chromosomal events[20,24,25,74,76,108–110]. It plays a widely conserved role in meiosis and is often regarded as an axis component[75], but its specificity seems to vary across species. In *C. elegans*, PDS-5 stabilizes REC-8 against WAPL-1, analogous to evidence from fission yeast[81]. However, loss of Pds5 in budding yeast or of PDS5 in *Arabidopsis* has little effect on the association of Rec8 (REC8) with meiotic chromosomes[82,83,111]. Nevertheless, budding yeast *PDS5* mutants show SC formation between sister chromatids, rather than homologous chromosomes, a phenotype also seen in *C. elegans* and mouse spermatocytes lacking REC-8 or Rec8, respectively[28,82]. Thus, Pds5 may promote cohesive activity of Rec8 cohesin, even if it is dispensable for Rec8 localization to chromosome axes.

The functional interplay between Pds5 and Wapl in different organisms has been enigmatic. Some results have indicated that these factors act as a complex; in other cases, Pds5 antagonizes Wapl activity, as shown here. We found that WAPL-1 antagonizes cohesin localization even when PDS-5 is depleted (Fig. 7g,h), indicating that it can function independently of PDS-5. Most importantly, WAPL-1 depletion restores axial REC-8 cohesin upon PDS-5 depletion. Notably, the FGF and YSR motifs in vertebrate Wapl that mediate binding to Pds5 (refs. [24,90]) are both absent from *C. elegans* WAPL-1 (ref. [71]). *C. elegans* PDS-5 also has a relatively long, unstructured domain that may modulate its activities through mechanisms analogous to the function of Sororin in vertebrates and Dalmatian in *Drosophila*, both of which are cohesin-protecting factors that antagonize WAPL-dependent cohesin release[61].

Taken together, the results of our work show that axis assembly is driven by the specialized roles of meiotic cohesins and their interactions with cohesin regulators, which in turn are controlled by specialized DDR signaling during meiotic prophase. Additionally, our study reveals a pathway that regulates cohesin activity to promote programmed induction and repair of DSBs in meiotic cells and repair of exogenous breaks in proliferating cells.

## Online content

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

## Methods

### Strain maintenance

All *C. elegans* strains were maintained on standard nematode growth medium (NGM) plates seeded with OP50 bacteria at 20 °C[112]. Young adult hermaphrodites (20–24 hours (h) post-L4) were used for immunofluorescence analysis. *C. elegans* is a laboratory model nematode that does not require ethical approval to study.

### Strain construction

All new alleles used in this study were generated by CRISPR–Cas9-mediated genome editing. See Supplementary Table 1 for the list of strains used in this study. Briefly, Alt-R CRISPR–Cas9 crRNAs specific for target sites were mixed with a *dpy-10*-specific crRNA[113] at a molar ratio of 8:1. These were denatured and annealed to an equal quantity of tracrRNA (Integrated DNA Technologies) by being heated to 95 °C for 5 minutes (min), followed by 5 min at 25 °C. 1 µL of 100 µM hybridized tracrRNA/crRNA was combined with 2.5 µL of 40 µM *S. pyogenes* Cas9-NLS purified protein (QB3 MacroLab) and incubated at room temperature for 5 min. Next, 0.5 µL of 100 µM stock of an Ultramer DNA oligonucleotide (IDT) repair template containing 35–45 bp homology arms and the desired epitope/degron or mutation sequence was added to the mixture, for a total volume of 5 µL, and injected into the gonads of young adult hermaphrodites aged 24 h from the late L4 stage. Hermaphrodites were injected and maintained on individual plates at 20 °C for 3–4 days. Roller and Dumpy F1 progeny were singled, maintained at 20 °C for 3 days, and screened by PCR for the desired mutation or epitope tag. Candidate alleles were verified by Sanger sequencing. See Supplementary Table 2 for a complete list of crRNAs, repair templates, and genotyping primer sequences used in this study. Validations of key constructed strains are available in Supplementary Table 4.

### Worm viability and fertility

To quantify brood sizes, male self-progeny, and embryonic viability, L4 hermaphrodites were plated individually and transferred to new plates daily for four consecutive days. Eggs were counted twice a day to minimize counting errors. Viable progeny and males were scored when they reached young adulthood.

### Auxin-induced protein depletion in worms

Auxin-induced depletion of degron-tagged proteins was performed as previously described[114]. Hermaphrodites at the L4 stage were transferred to seeded plates containing 1 mM indole-3-acetic acid (IAA, auxin) and incubated for 24 h before analysis. For each experiment, strains being compared were treated in parallel using the same batch of auxin-containing plates. Degron and epitope tags were inserted into genes of interest by genome editing in a strain expressing TIR1 in the germline (see Supplementary Table 1 for detailed information), which was used as a control and treated in parallel with all other strains in each assay. The stability and function of degron-tagged proteins in the absence of auxin were validated by localization and/or phenotypic assays. The kinetics and efficacy of depletion were analyzed by immunolocalization, functional assays, and, where feasible, by western blots.

### Plasmids

To express human WAPL in HeLa cells, sequences were inserted into the pcDNA3-acGFP vector, obtained from Addgene (cat. no. 128047). The WAPL coding sequence was divided into four ~1-kb fragments (sequences are available in Supplementary Table 3) and synthesized by Twist Bioscience. These fragments were inserted at the 3′ end of the GFP coding sequence using Gibson assembly[115] and verified by Sanger sequencing.

### Antibodies and reagents

Primary antibodies were purchased from commercial sources or have been described in previous studies, and were diluted as follows: rabbit anti-RAD-51 (1:500)[39], rabbit anti-pHIM-8/ZIMs (1:500)[17], goat anti-SYP-1 (1:300)[39], chicken anti-HTP-3 (1:500)[116], mouse anti-HA (1:400, Thermo Fisher 26183)[117], mouse anti-FLAG (1:500, Sigma F3165)[114], mouse anti-V5 (1:500, Thermo Fisher R960-25)[117], rabbit anti-V5 (1:250, Millipore Sigma V8137)[117], mouse anti-WAPL (1:500, Santa Cruz sc-365189), rabbit anti-γH2A.X antibody (1:500, Cell Signaling, cat. no. 2577), mouse anti-ATM antibody (1:500, Thermo Fisher, cat no. MA1-23152), rabbit anti-pS/TQ antibody (1:500, Cell Signaling, cat. no. 6966), rabbit anti-COH-3/4 antibody (1:500, SDQ3972, ModENCODE project)[118], rabbit anti-REC-8 antibody (1:500, SDQ0802, ModENCODE project), rabbit anti-WAPL-1 antibody (1:500, SDQ3963, ModENCODE project). Secondary antibodies raised in donkey and labeled with Alexa Fluor 488, Cy3, Cy5, or Alexa Fluor 647 (Jackson ImmunoResearch Laboratories, Alexa Fluor 488-donkey anti-mouse no. 715-545-151, Alexa Fluor 488-donkey anti-chicken no. 703-545-155, Alexa Fluor 488-donkey anti-goat no. 705-545-147, Cy3-donkey anti-mouse no. 715-165-151, Cy3-donkey anti-rabbit no. 711-165-152, Cy3-donkey anti-chicken no. 703-165-155, Cy5-donkey anti-mouse no. 715-175-151, Cy5-donkey anti-chicken no. 703-175-155, Alexa Fluor 647-donkey anti-mouse no. 715-605-151, Alexa Fluor 647-donkey anti-goat no. 705-605-147, Alexa Fluor 647-donkey anti-rabbit no. 711-605-152) and were used at 1:400 dilution. Kinase inhibitors included VE-821 (Selleckchem S8007); NU7441 (Selleckchem S2638); and KU55933 (Selleckchem S1092). DNA-damage-inducing agents included ETO (Sigma cat. no. E1383) and bleomycin (Fisher cat. no. B397210MG).

### siRNA-mediated knockdown

The following ON-TARGETplus SMARTpool siRNAs were purchased from Horizon Discovery: non-targeting control pool (negative control pool), cat. no. D-001810-10-05; WAPL siRNA, cat. no. L-026287-01-0005; ATM siRNA, cat. no. L-003201-00-0005; ATR siRNA, cat. no. L-003202-00-0005. HeLa cells were cultured on coverslips in 6-well plates to 25% confluency, and siRNA knockdown was performed using DharmaFECT, according to the manufacturer's recommendations. Cells were fixed and analyzed 72 h after siRNA transfection. DNA-damage-inducing agents and/or kinase inhibitors were added 24 h before fixation.

### Transient transfection

For WAPL overexpression, HeLa cells were grown on coverslips in 6-well plates to 50% confluency. Then, 2.5 µg of purified plasmid DNA was mixed with 5 µL Lipofectamine 3000 (Thermo Fisher) and used for transfection according to the manufacturer's protocol. Cells were fixed for imaging 48 h after transfection. DNA-damage-inducing agents and/or kinase inhibitors were added 24 h before fixation.

### Chemical treatments

DNA damage was induced by addition of 0.8 µM ETO or 0.4 µM bleomycin for 24 h. KU55933, VE-821, and NU7441 were added at 1 µM for 24 h. Chemicals were dissolved in DMSO (dimethylsulfoxide).

### Immunofluorescence assays

Adult hermaphrodites were dissected on a clean coverslip in egg buffer (25 mM HEPES pH 7.4, 118 mM NaCl, 48 mM KCl, 2 mM EDTA, 0.5 mM EGTA) containing 0.01% tetramisole and 0.1% Tween-20. Samples were fixed for 2 min in egg buffer containing 1% formaldehyde and then transferred to a 1.5-mL tube containing PBS + 0.1% Tween-20 (PBST). After 5 min, the buffer was replaced with ice-cold methanol and incubated at −20 °C for an additional 10 min. Worms were washed twice with PBST, blocked with Roche blocking reagent diluted into PBST, and stained with primary antibodies diluted in blocking solution at 4 °C overnight. Samples were then washed with PBST and incubated with secondary antibodies that were diluted in blocking solution at room temperature for 1 h. Worms were washed twice with PBST and mounted in ProLong Diamond with DAPI (Invitrogen) before imaging.

For immunofluorescence of HeLa cells, coverslips in 6-well plates were washed with PBS and then fixed with 4% formaldehyde in PBS at room temperature for 10 min. After 3 washes with PBS, cells were permeabilized by addition of 0.5% Triton X-100 in PBS at room temperature for 5 min. They were rinsed with PBS and blocked with 5% BSA in PBS at room temperature for 1 h. Cells were then washed with PBS and incubated with primary antibodies diluted in 1% BSA at room temperature for 2 h. After another PBS wash, cells were incubated in secondary antibodies diluted in 1% BSA in PBS at room temperature for 1 h in the dark. Cells were then washed again with PBS and mounted in ProLong Diamond with DAPI before imaging.

### Microscopy
All images were acquired as z-stacks of optical sections at 0.2-μm intervals using a DeltaVision Elite microscope (GE) with a ×100, 1.4 numerical aperture (NA) or ×60, 1.42 NA oil-immersion objective. Iterative three-dimensional (3D) deconvolution, image projection, and colorization were performed using the softWoRx package, ImageJ/Fiji (v1.53t), and Adobe Photoshop CC 2017, respectively.

### Image analysis
To quantify the abundance of proteins in *C. elegans* germline nuclei, additive projections were generated from raw (undeconvolved) 3D data stacks after background subtraction using the rolling ball tool in ImageJ. Individual nuclei (regions of interest, ROIs) were manually segmented based on DAPI staining in ImageJ, and the integrated intensity within each ROI was calculated. For each condition, 80-200 nuclei from 3-4 representative gonads were quantified.

To quantify protein abundance in HeLa cell nuclei, individual nuclei (ROIs) were first segmented on the basis of DAPI fluorescence in an equatorial optical section from a 3D image stack using the 2D watershed tool (scikit-image library v0.18, Python 3.9). Protein abundance (integrated intensity) within this region was calculated from additive Z projections, similar to the approach we used to quantify proteins in *C. elegans* germline nuclei.

To quantify RAD21 enrichment at sites of DNA damage in HeLa cells, we developed an automated method to ensure consistency and minimize potential investigator bias. Following empirical optimization, the method was applied to each dataset using ImageJ macros. For experiments involving expression of GFP or GFP-WAPL, only GFP-positive cells were included; these were identified based on GFP fluorescence in equatorial sections using the Auto Threshold tool in ImageJ in 'Li' maximum entropy mode. Nuclear ROIs were segmented as described above. Peaks of immunofluorescence of DNA-damage markers (γH2A.X or pS/TQ) were segmented using the Auto Threshold tool in MaxEntropy mode. The resulting binary masks of damage-marker-enriched nuclear regions were used to segment and calculate the average intensity (integrated intensity/area) of RAD21 at DNA-damage regions from additive Z projections. The average background RAD21 intensity was calculated from the nuclear regions outside of these masks.

### Data presentation
For data based on immunofluorescence in *C. elegans* germline nuclei, we show representative images of the distal regions of dissected gonads. All images are oriented with the distal tip on the left. They show the entire proliferative (PM) region and a similarly sized region containing nuclei in EM prophase. The boundary between PM and meiotic prophase is indicated by a dashed line. Figure labels indicate proteins that were depleted by auxin treatment. For immunofluorescence in HeLa cells, representative images of individual nuclei are shown with enlargements of fluorescent foci as insets.

For quantitative analysis of immunofluorescence, the integrated nuclear intensity or number of foci were measured as described above under 'Image analysis.' Tukey boxplots of data points from individual nuclei were generated using R. Boxes indicate the quartiles and median, and the median value is also indicated next to the box. The number of nuclei that were scored for each condition or group is shown in parentheses underneath the data points.

### Statistical analysis
We used the Shapiro–Wilk test to determine whether our data for each condition showed a normal distribution. We used Student's *t*-test to compare data sets that were found to show a normal distribution ($P > 0.05$ by the Shapiro–Wilk test); otherwise, we used the Wilcoxon–Mann–Whitney test. *P* values were adjusted by Bonferroni correction when statistical analyses involved multiple tests on the same dataset. The number of asterisks indicates the calculated *P* values: **$P < 0.01$, ***$P < 0.001$, ****$P < 0.0001$. The exact values of the *P* values that are greater or equal to 0.01 are indicated on the plots.

### Reporting summary
Further information on research design is available in the Nature Portfolio Reporting Summary linked to this article.

## Data availability
Source data are provided with this paper.

## Code availability
ImageJ macros and Python scripts to automate the image analysis process tasks are available at https://github.com/zhouliangyu.

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

## Acknowledgements
We thank L. Zhang and F. Wu for providing help in strain construction and preparation of experimental materials. We thank D. Koshland, V. Guacci, S. Xiang, L. Costantino, and K. Boardman, and all Dernburg lab members, for insightful discussions throughout this study and for critical reading of the manuscript. We also thank W. Zhang for help with the mammalian cell experiments, and C. Glazier for preliminary analysis of WAPL-1 function in *C. elegans*. The work was funded by the National Institutes of Health grant R01 GM065591 (A. F. D.) and an Investigator award from the Howard Hughes Medical Institute (A. F. D.). The funders had no role in study design, data collection and analysis, decision to publish or preparation of the manuscript.

## Author contributions
Z.Y. and A.F.D. conceived the study and designed the experiments. Z.Y. and H.J.K. performed experiments, with data analyzed by Z.Y.

under A.F.D.'s supervision. Z.Y. and A.F.D. wrote the manuscript, with contributions from H.J.K.

**Competing interests**

The authors declare no competing interests.

**Additional information**

**Extended data** is available for this paper at https://doi.org/10.1038/s41594-023-00929-5.

**Correspondence and requests for materials** should be addressed to Abby F. Dernburg.

**Peer review information** *Nature Structural & Molecular Biology* thanks Andreas Hochwagen and the other, anonymous, reviewer(s) for their contribution to the peer review of this work. Editor recognition statement (if applicable to your journal): Beth Moorefield, Florian Ullrich and Dimitris Typas were the primary editors on this article and managed its editorial process and peer review in collaboration with the rest of the editorial team. Peer reviewer reports are available.

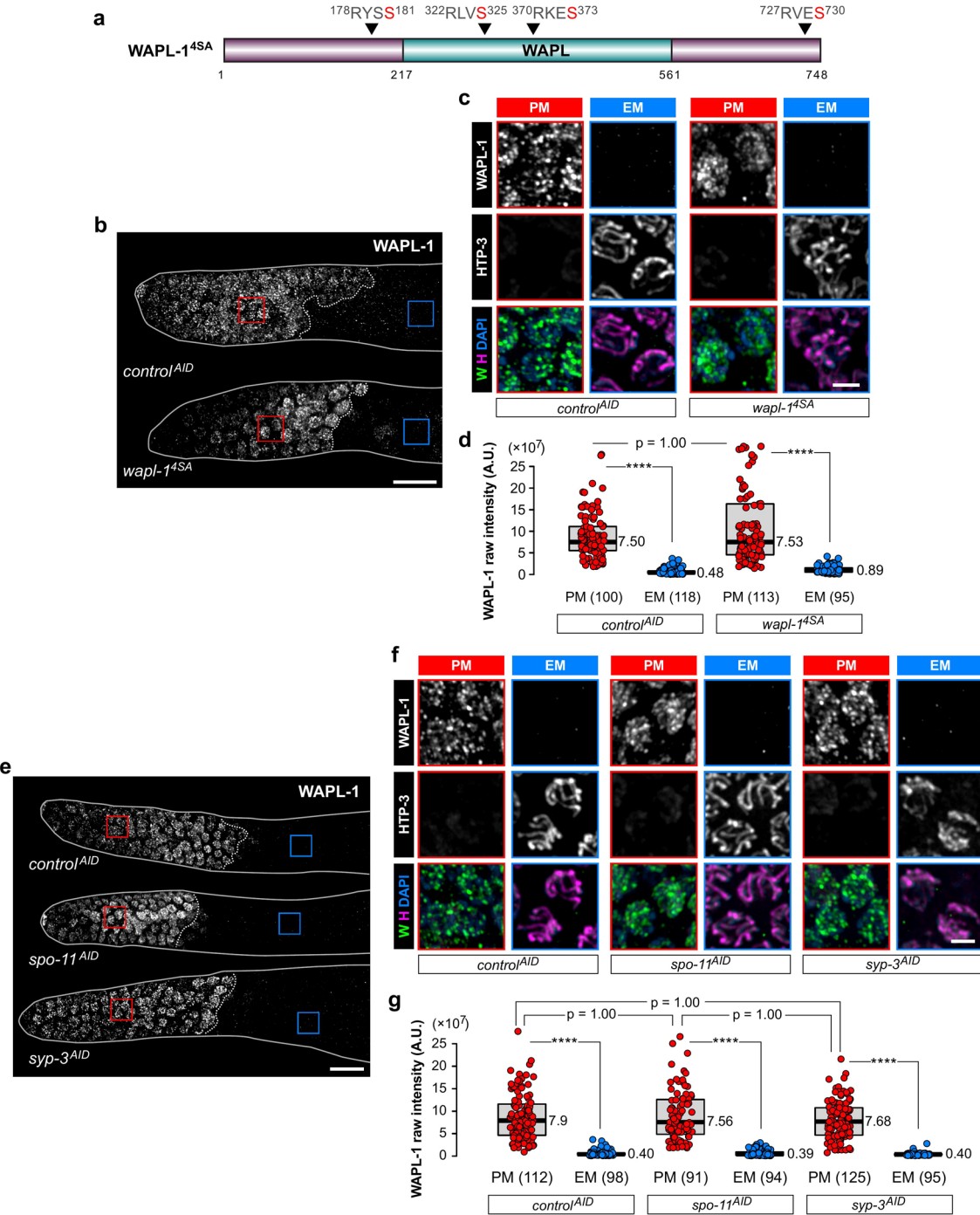

**Extended Data Fig. 1 | WAPL-1 downregulation does not require direct phosphorylation by CHK-2, DSBs, or synapsis. a**, WAPL-1 sequence indicating the positions of CHK-2 consensus motifs that were mutated in the wapl-1⁴ˢᴬ allele. **b,e**, WAPL-1 immunostaining germline nuclei, showing that neither direct CHK-2-dependent WAPL-1 phosphorylation, SPO-11-dependent meiotic DSB formation, nor SC components is required for WAPL-1 suppression at meiotic entry. Scale bar, 10 μM. **c,f**, Enlarged images of the regions outlined in **b** and **e**.

HTP-3 marks chromosome axes in meiotic prophase nuclei. Scale bar, 2 μM. **d,g**, Quantification of WAPL-1 intensity in **b** and **e**, respectively. Lower and upper box ends represent the first and third quartiles, with the median indicated as the horizontal line within the box. All data points are shown with the sample sizes indicated below the boxes. ****$P < 0.0001$ (two-sided Wilcoxon–Mann–Whitney test, adjusted by Bonferroni correction). Data for plots in **d** and **g** are available as source data.

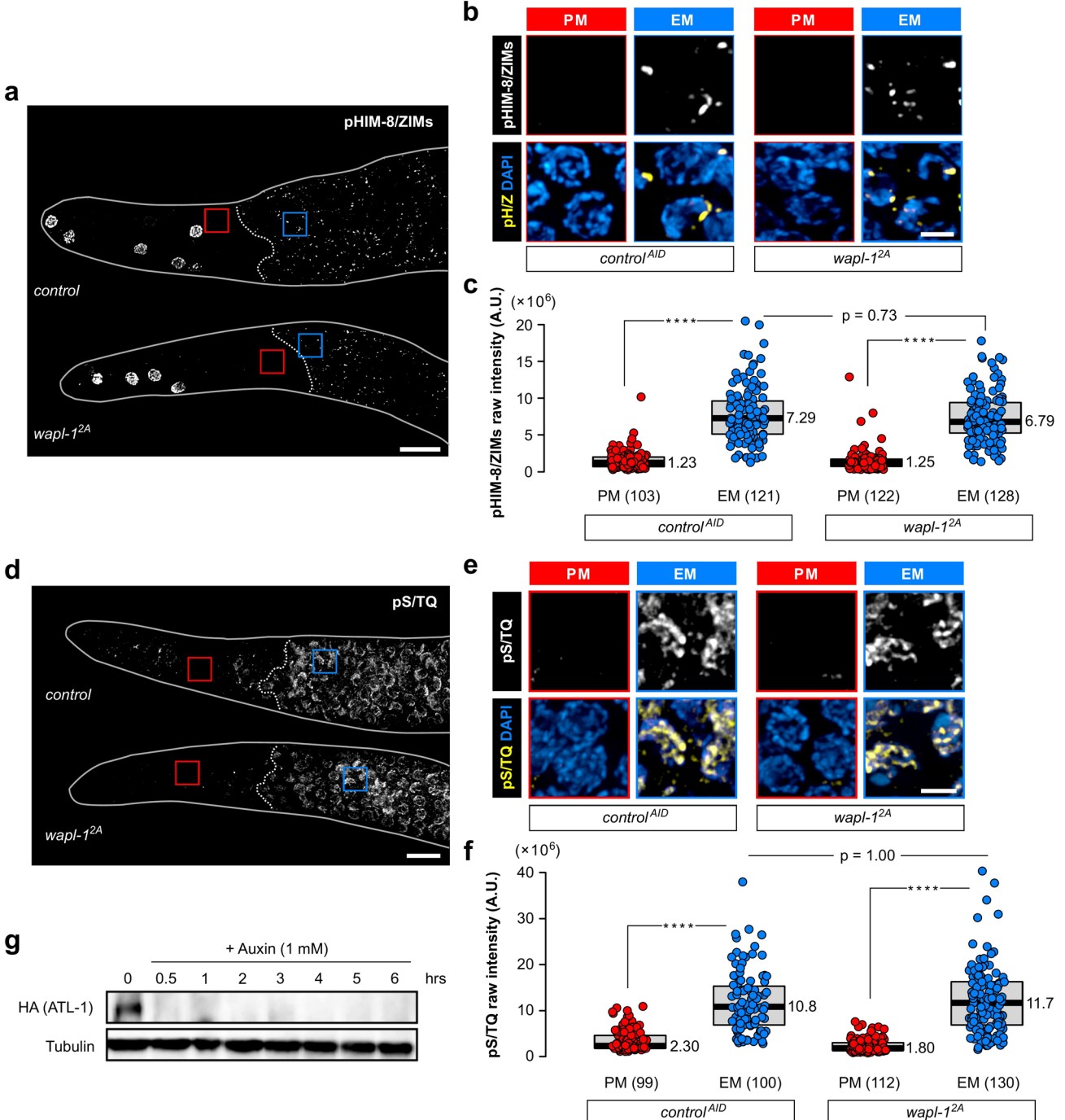

**Extended Data Fig. 2 | Mutation of ATM target sites in WAPL-1 does not affect CHK-2 or ATM-1 activity at meiotic entry. a,d**, pHIM-8/ZIM (**a**) and pS/TQ (**d**) immunostaining in the distal tip of gonads of wild-type and *wapl-1*[2A] worms, showing that the activity of CHK-2 and ATM-1 do not change in the nonphosphorylatable WAPL-1 mutant. 'Control' indicates animals from the same background strain carrying wild-type alleles. The pHIM-8/ZIM antibody also recognizes an unidentified, CHK-2-independent epitope in mitotic cells. Scale bar, 10 μM. **b** and **e**, Enlarged images of the regions outlined in **a** and **d**,

respectively. Scale bar, 2 μM. **c** and **f**, Quantification of the intensity of pHIM-8/ZIMs (**c**) and pS/TQ (**f**). Lower and upper box ends represent the first and third quartiles, with the median indicated as the horizontal line within the box. All data points are shown with the sample sizes indicated below the boxes. ****$P < 0.0001$ (two-sided Wilcoxon–Mann–Whitney test, adjusted by Bonferroni correction). **g**, Western blot analysis showing the depletion of HA-degron-tagged ATL-1 upon auxin addition. Unprocessed Western blot image for **g** and Data for plots in **c** and **f** are available as source data.

**a**

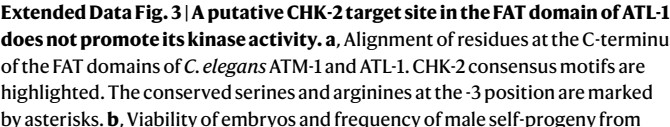

**b**

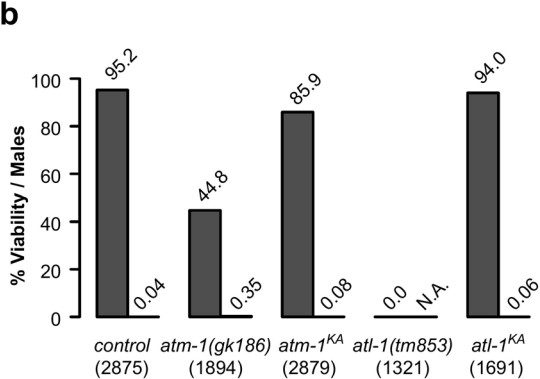

**Extended Data Fig. 3 | A putative CHK-2 target site in the FAT domain of ATL-1 does not promote its kinase activity. a**, Alignment of residues at the C-terminus of the FAT domains of *C. elegans* ATM-1 and ATL-1. CHK-2 consensus motifs are highlighted. The conserved serines and arginines at the -3 position are marked by asterisks. **b**, Viability of embryos and frequency of male self-progeny from hermaphrodites homozygous for the indicated alleles. Null alleles of both *atm-1* (*atm-1(gk186)*) and *atl-1* (*atl-1(tm853)*) were assayed for comparison. The number of eggs scored for each allele is shown in parentheses. Data for graphs in **b** are available as source data.

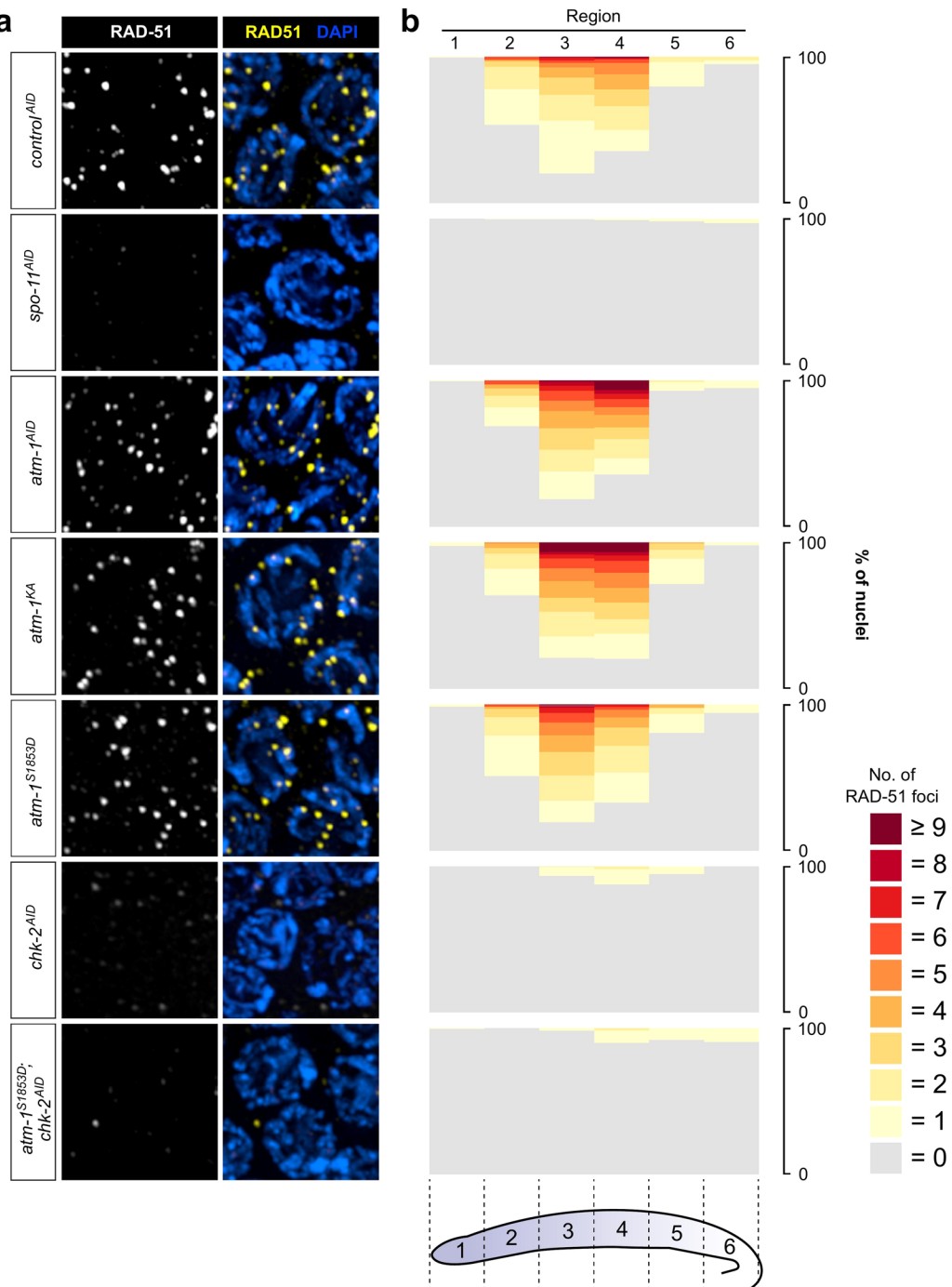

**Extended Data Fig. 4 | DSBs are dispensable for a basal level of ATM-1 activity in meiotic prophase nuclei. a**, RAD-51 foci indicate the presence or absence of meiotic DSBs in pachytene nuclei. Scale bar, 2 μM. *n* = 4 biological replicates. **b**, Quantitative analysis of RAD-51 foci. The region of the germline from the distal tip to late pachytene was divided into six zones of equal length. The distribution of RAD-51 foci per nucleus nuclei for each region is shown for each of the conditions represented by images in **a**. Data for plots in **b** are available as source data.

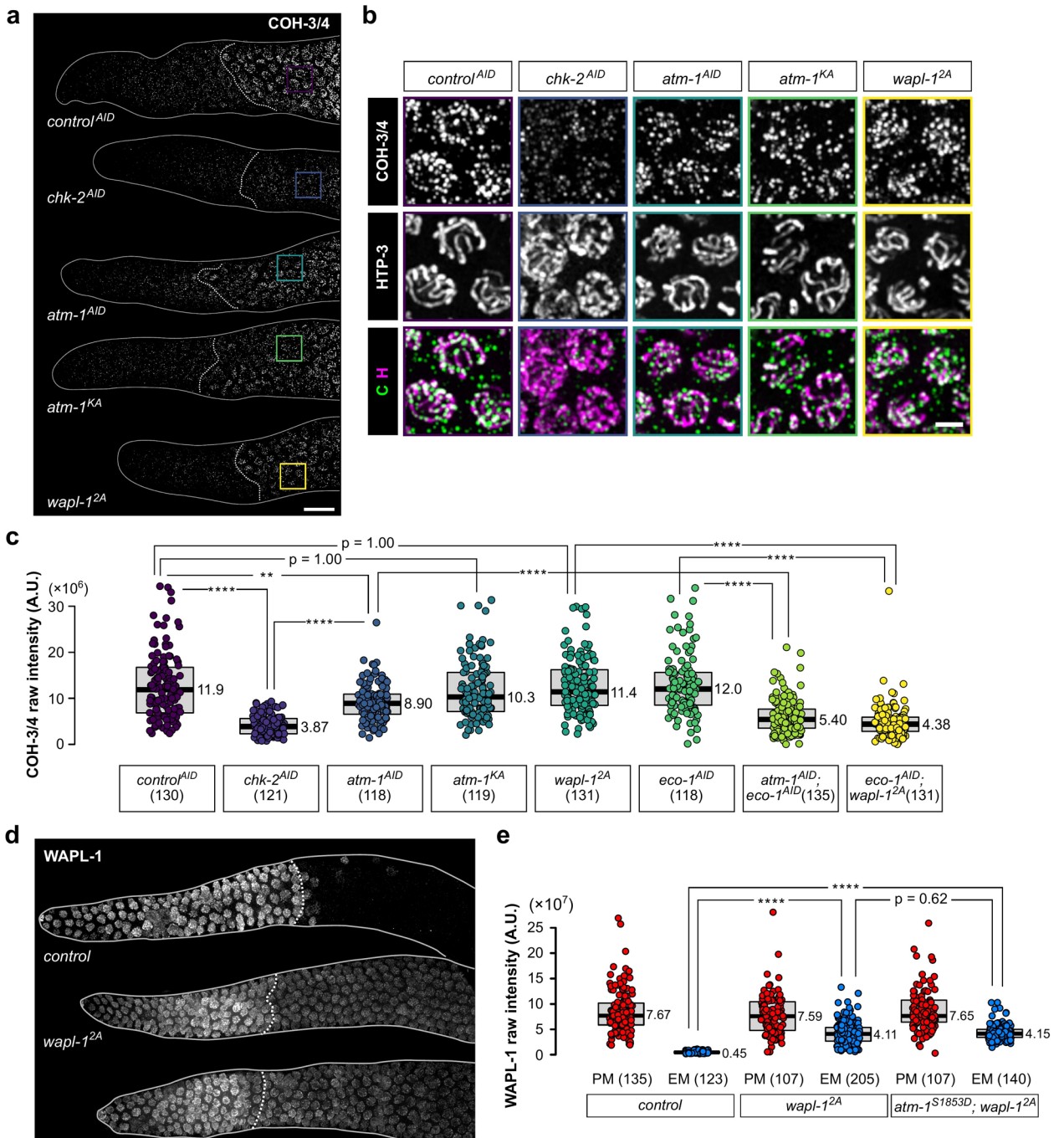

**Extended Data Fig. 5 | CHK-2 promotes axial cohesin stabilization. a**, COH-3/4 immunofluorescence in the distal region of gonads, showing that CHK-2 activity controls a cohesin-stabilizing activity that is independently of WAPL-1 suppression. Dashed lines indicate the boundaries between PM and meiotic germline. Scale bar, 10 μM. **b**, Enlarged images of the regions indicated in **a**. HTP-3 immunostaining (in magenta) marks chromosome axes. Scale bar, 2 μM. **c**, Quantification of the intensity of COH-3/4 immunostaining in **a** and in Fig. 6a. Lower and upper box ends represent the first and third quartiles, with the median indicated as the horizontal line within the box. All data points are shown with the sample sizes indicated below the boxes. ****$P < 0.0001$ (two-sided Wilcoxon–Mann–Whitney test, adjusted by Bonferroni correction). **$P < 0.01$ ($P = 0.0013$,

two-sided Wilcoxon–Mann–Whitney test, adjusted by Bonferroni correction) **d**, WAPL-1 immunostaining in the distal tip of gonads, showing that ATM-1$^{S1853D}$ fails to downregulate the nonphosphorylatable WAPL-1$^{2A}$ at meiotic entry. 'Control' indicates animals from the same background strain carrying wild-type alleles. Scale bar, 10 μM. **e**, Quantification of the intensity of WAPL-1 immunostaining in **d**. Lower and upper box ends represent the first and third quartiles, with the median indicated as the horizontal line within the box. All data points are shown with the sample sizes indicated below the boxes. ****$P < 0.0001$ (two-sided Wilcoxon–Mann–Whitney test, adjusted by Bonferroni correction). Data for plots in **c** and **e** are available as source data.

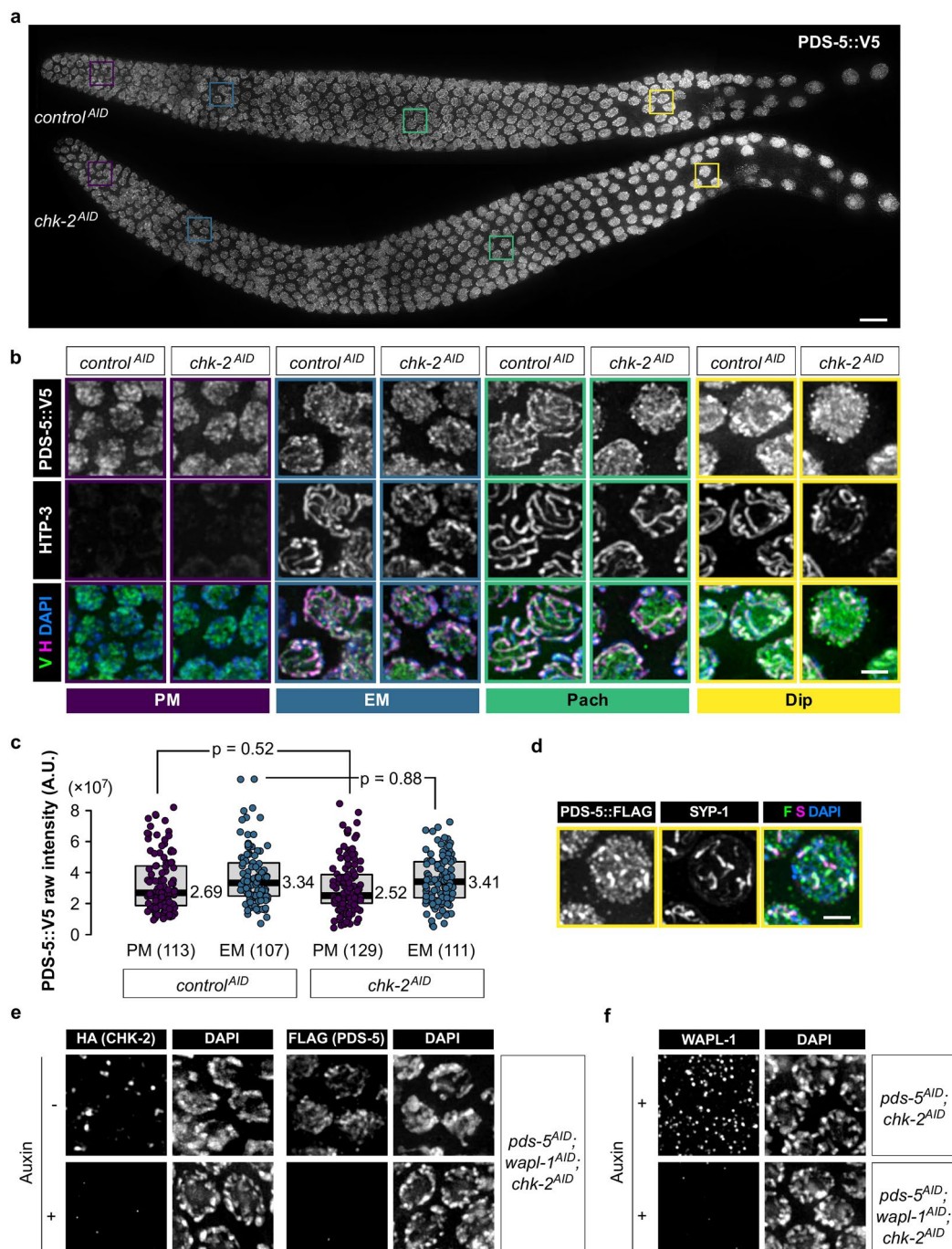

**Extended Data Fig. 6 | PDS-5 localizes to meiotic chromosome axes independently of CHK-2. a**, PDS-5::V5 immunofluorescence using anti-V5 in *C. elegans* gonads, showing that PDS-5 localization does not change in the absence of CHK-2. Scale bar, 10 μM. **b**, Enlarged images of the regions outlined in **a**. HTP-3 immunostaining (in magenta) marks chromosome axes. Scale bar, 2 μM. **c**, Quantification of the PDS-5::V5 intensity in **b**. Lower and upper box ends represent the first and third quartiles, with the median indicated as the horizontal line within the box. All data points are shown with the sample sizes indicated below the boxes. *P* values are calculated by two-sided Wilcoxon–Mann–Whitney test. **d**, PDS::FLAG immunofluorescence in diplotene nuclei, showing that PDS-5 localizes to 'short arms'. SYP-1 immunofluorescence in late diplotene nuclei is restricted to the 'short arm' of each bivalent. Scale bar, 2 μM. *n* = 3 biological replicates. **e**,**f**, Immunofluorescence of HA, FLAG and WAPL-1 in *C. elegans* gonads of *pds-5*[AID]; *wapl-1*[AID]; *chk-2*[AID], showing that the three proteins are depleted simultaneously upon auxin addition. Data for plots in **c** are available as source data.

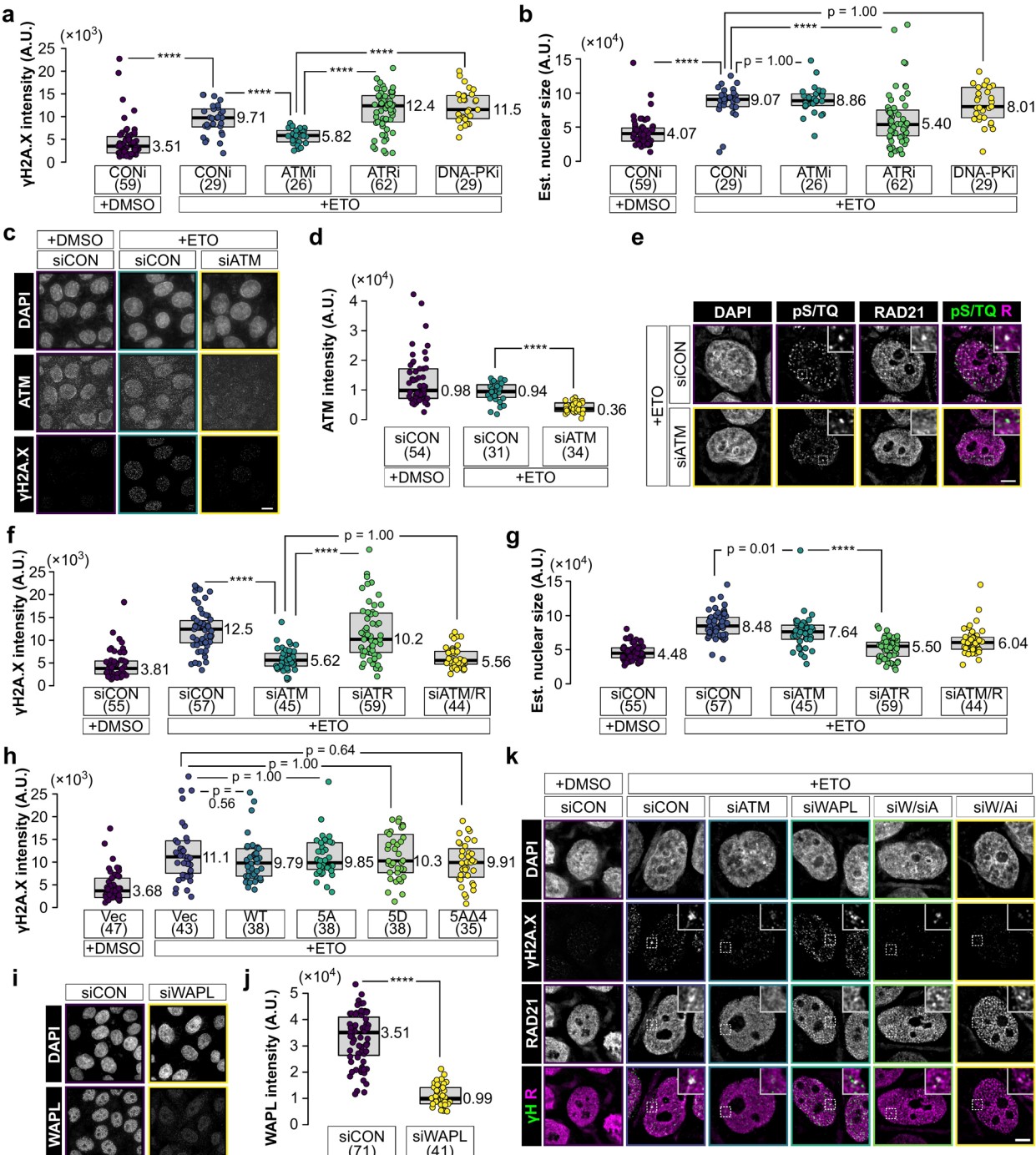

**Extended Data Fig. 7 | ATM-mediated WAPL downregulation promotes cohesin enrichment at DNA damage foci. a,b**, Quantification of nuclear γH2A.X intensity (**a**) and nuclear size (**b**) in cells that were exposed to ETO-induced following chemical inhibition of DNA damage kinases. ATM was inhibited by addition of KU55933 (ATMi), ATR by VE-821 (ATRi), and DNA-PK by NU7441 (DNA-Pki). **c**, Immunofluorescence of ATM in HeLa cells treated with control siRNA or siRNA targeting ATM, with or without ETO treatment. **d**, Quantification of nuclear ATM intensity in cells treated as in (**c**). Integrated ATM intensity was normalized against DAPI intensity for each nucleus. **e**, Immunofluorescence of RAD21 in nuclei of HeLa cells treated with either control siRNA or siRNA against ATM, and then ETO to induce DNA damage. DNA damage foci are marked by pS/TQ immunofluorescence. **f,g**, Quantification of nuclear γH2A.X intensity (**f**) and estimated nuclear size (**g**) in cells following siRNA-mediated knockdown of ATM and/or ATR. The integrated γH2A.X intensities were normalized by integrated DAPI intensities for each nucleus. **h**, Quantification of nuclear γH2A.X intensity

in transfected HeLa cells expressing GFP or GFP-WAPL fusion proteins, following ETO-induced DNA damage. **i**, Immunofluorescence of WAPL in nuclei of HeLa cells treated with either control siRNA or siRNA against WAPL. **j**, Quantification of nuclear WAPL intensity in HeLa cells, normalized against DAPI intensity in each nucleus (**i**). **k**, Immunofluorescence of RAD21 in nuclei of HeLa cells treated with the indicated siRNA and then ETO. WAPL siRNA was combined with KU55933 (ATMi) or siRNA targeting ATM. γH2A.X immunofluorescence marks DNA damage foci. *n* = 3 biological replicates, at least 25 cells for each condition are assayed for each condition. Data for plots in **a**, **b**, **d**, **f**, **g**, **h** and **j** are available as source data. In all quantitative analyses, lower and upper box ends represent the first and third quartiles, with the median indicated as the horizontal line within the box. All data points are shown with the sample sizes indicated below the boxes. *P* values are calculated by two-sided Wilcoxon–Mann–Whitney test, multiple comparisons are adjusted by Bonferroni correction. ****P < 0.0001. Scale bar, 10 μM.

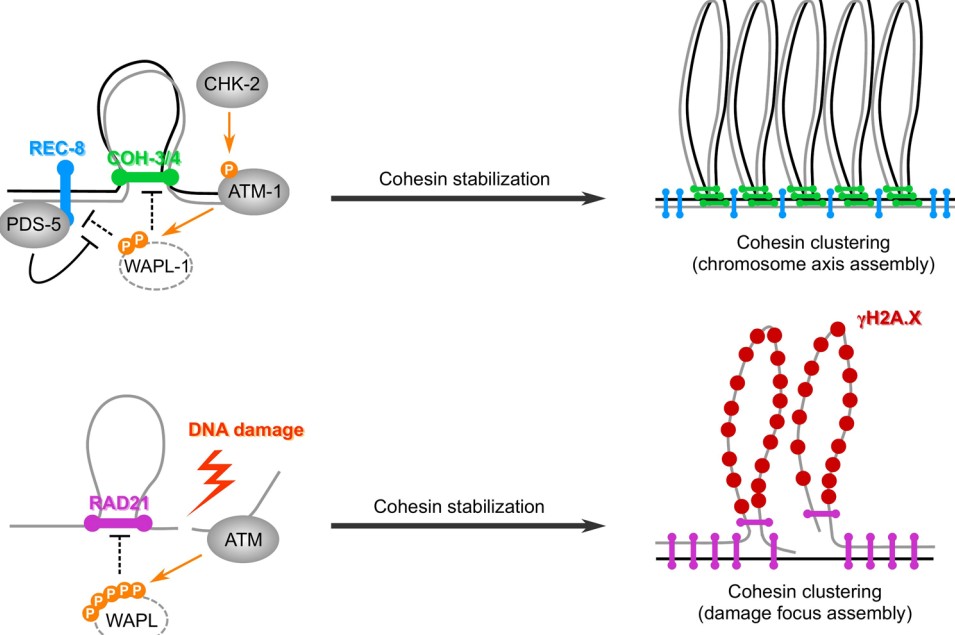

**Extended Data Fig. 8 | A conserved phosphoregulatory pathway promotes cohesin clustering in meiosis and proliferating cells.** In *C. elegans* meiosis, ATM-mediated WAPL downregulation promotes global cohesin clustering to form the meiotic chromosome axis. The same pathway promotes local cohesin clustering at DNA damage foci in mammalian proliferating cells.

# Reporting Summary

## Statistics

For all statistical analyses, confirm that the following items are present in the figure legend, table legend, main text, or Methods section.

| n/a | Confirmed | |
|---|---|---|
| ☐ | ☒ | The exact sample size (*n*) for each experimental group/condition, given as a discrete number and unit of measurement |
| ☐ | ☒ | A statement on whether measurements were taken from distinct samples or whether the same sample was measured repeatedly |
| ☐ | ☒ | The statistical test(s) used AND whether they are one- or two-sided *Only common tests should be described solely by name; describe more complex techniques in the Methods section.* |
| ☒ | ☐ | A description of all covariates tested |
| ☐ | ☒ | A description of any assumptions or corrections, such as tests of normality and adjustment for multiple comparisons |
| ☐ | ☒ | A full description of the statistical parameters including central tendency (e.g. means) or other basic estimates (e.g. regression coefficient) AND variation (e.g. standard deviation) or associated estimates of uncertainty (e.g. confidence intervals) |
| ☐ | ☒ | For null hypothesis testing, the test statistic (e.g. *F*, *t*, *r*) with confidence intervals, effect sizes, degrees of freedom and *P* value noted *Give P values as exact values whenever suitable.* |
| ☒ | ☐ | For Bayesian analysis, information on the choice of priors and Markov chain Monte Carlo settings |
| ☒ | ☐ | For hierarchical and complex designs, identification of the appropriate level for tests and full reporting of outcomes |
| ☒ | ☐ | Estimates of effect sizes (e.g. Cohen's *d*, Pearson's *r*), indicating how they were calculated |

*Our web collection on statistics for biologists contains articles on many of the points above.*

## Software and code

Policy information about availability of computer code

| Data collection | DeltaVision/softWoRx |
|---|---|
| Data analysis | ImageJ/Fiji (v1.53t), scikit-image library v0.18 (Python 3.9) and ImageJ macros and Python scripts to automate the image analysis tasks (https://github/zhouliangyu). Microscopy image blending is processed in Adobe Photoshop CC 2017. |

For manuscripts utilizing custom algorithms or software that are central to the research but not yet described in published literature, software must be made available to editors and reviewers. We strongly encourage code deposition in a community repository (e.g. GitHub). See the Nature Portfolio guidelines for submitting code & software for further information.

## Data

Policy information about availability of data

All manuscripts must include a data availability statement. This statement should provide the following information, where applicable:
- Accession codes, unique identifiers, or web links for publicly available datasets
- A description of any restrictions on data availability
- For clinical datasets or third party data, please ensure that the statement adheres to our policy

Source data are provided with this paper.

# Human research participants

Policy information about studies involving human research participants and Sex and Gender in Research.

| | |
|---|---|
| Reporting on sex and gender | *Use the terms sex (biological attribute) and gender (shaped by social and cultural circumstances) carefully in order to avoid confusing both terms. Indicate if findings apply to only one sex or gender; describe whether sex and gender were considered in study design whether sex and/or gender was determined based on self-reporting or assigned and methods used. Provide in the source data disaggregated sex and gender data where this information has been collected, and consent has been obtained for sharing of individual-level data; provide overall numbers in this Reporting Summary. Please state if this information has not been collected. Report sex- and gender-based analyses where performed, justify reasons for lack of sex- and gender-based analysis.* |
| Population characteristics | *Describe the covariate-relevant population characteristics of the human research participants (e.g. age, genotypic information, past and current diagnosis and treatment categories). If you filled out the behavioural & social sciences study design questions and have nothing to add here, write "See above."* |
| Recruitment | *Describe how participants were recruited. Outline any potential self-selection bias or other biases that may be present and how these are likely to impact results.* |
| Ethics oversight | *Identify the organization(s) that approved the study protocol.* |

Note that full information on the approval of the study protocol must also be provided in the manuscript.

# Field-specific reporting

Please select the one below that is the best fit for your research. If you are not sure, read the appropriate sections before making your selection.

☒ Life sciences   ☐ Behavioural & social sciences   ☐ Ecological, evolutionary & environmental sciences

For a reference copy of the document with all sections, see nature.com/documents/nr-reporting-summary-flat.pdf

# Life sciences study design

All studies must disclose on these points even when the disclosure is negative.

| | |
|---|---|
| Sample size | We quantified at least 25 cells for each mutant/group/condition in every assay, because a sample that has a size of >25 is generally considered to be large enough to reflect if the data is distributed normally. In most cases in the study, the numbers (sample size) ranged from several tens to above a hundred. |
| Data exclusions | We quantified all intact cells that are not close to the edges of any image we collected. For C. elegans germline images, we only quantified meiotic cells that are in the stage of meiotic onset/early meiosis (within 10-20 rows of cells after the transition from mitosis to meiosis). For mammalian cell culture images we excluded mitotic cells and only quantified interphase cells. This is because cohesin-enriched damage foci are only visible in interphase nuclei. |
| Replication | For data that were collected from each mutant/group/condition, we quantified meiotic cells of at least three gonads from individual worms. All replicate experiments are successful and meaningful. For DNA damage foci analysis, we analyzed cells from at least three biological replicates. From all replicates the results we obtained are similar. |
| Randomization | Worms freely crawl on plates and we picked worms from each mutant/group/condition at random. When acquiring images, of either C. elegans gonads or mammalian cell cultures, we picked gonads/fields of cells at random. |
| Blinding | Complete blinding is not applicable, because the person who carried out the experiments knew the genetic backgrounds/conditions of worms/cells from each group before image acquisition. However, when analyzing the images we acquired, we apply the same criteria/scripts on the raw data across all conditions to minimize human intervention. |

# Reporting for specific materials, systems and methods

We require information from authors about some types of materials, experimental systems and methods used in many studies. Here, indicate whether each material, system or method listed is relevant to your study. If you are not sure if a list item applies to your research, read the appropriate section before selecting a response.

## Materials & experimental systems

| n/a | Involved in the study |
|---|---|
| ☐ | ☒ Antibodies |
| ☐ | ☒ Eukaryotic cell lines |
| ☒ | ☐ Palaeontology and archaeology |
| ☐ | ☒ Animals and other organisms |
| ☒ | ☐ Clinical data |
| ☒ | ☐ Dual use research of concern |

## Methods

| n/a | Involved in the study |
|---|---|
| ☒ | ☐ ChIP-seq |
| ☒ | ☐ Flow cytometry |
| ☒ | ☐ MRI-based neuroimaging |

## Antibodies

| | |
|---|---|
| Antibodies used | Primary antibodies were purchased from commercial sources or have been described in previous studies, and were diluted as follows: rabbit anti-RAD-51 (Harper 2011, 1:500), rabbit anti-pHIM-8/ZIMs (Kim 2015, 1:500), goat anti-SYP-1 (Harper 2011, 1:300), chicken anti-HTP-3 (MacQueen 2005, 1:500), mouse anti-HA (Thermo Fisher 26183, 1:400), mouse anti-FLAG (Sigma F3165, 1:500), mouse anti-V5 (Thermo Fisher R960-25, 1:500), rabbit anti-V5 (Millipore Sigma V8137, 1:250), mouse anti-WAPL (Santa Cruz sc-365189, 1:500), rabbit anti-γH2A.X antibody (Cell Signaling, Cat No. 2577, 1:500), mouse anti-ATM antibody (Thermo Fisher, Cat No. MA1-23152, 1:500), rabbit anti-pS/TQ antibody (Cell Signaling, Cat No. 6966, 1:500), rabbit anti-COH-3/4 antibody (SDQ3972, ModENCODE project (Gerstein 2010), 1:500), rabbit anti-REC-8 antibody (SDQ0802, ModENCODE project (Gerstein 2010), 1:500), rabbit anti-WAPL-1 antibody (SDQ3963, ModENCODE project (Gerstein 2010), 1:500). Secondary antibodies raised in donkey and labeled with Alexa 488, Cy3, or Cy5 (Jackson ImmunoResearch Laboratories, 1:400, Alexa Fluor 488 Donkey anti-mouse #715-545-151, Alexa Fluor 488 Donkey anti-chicken #703-545-155, Alexa Fluor 488 Donkey anti-goat #705-545-147, Cy3 Donkey anti-mouse #715-165-151, Cy3 Donkey anti-rabbit #711-165-152, Cy3 Donkey anti-chicken #703-165-155, Cy5 Donkey anti-mouse #715-175-151, Cy5 Donkey anti-chicken #703-175-155, Alexa Fluor 647 Donkey anti-mouse #715-605-151, Alexa Fluor 647 Donkey anti-goat #705-605-147, Alexa Fluor 647 Donkey anti-rabbit #711-605-152). |
| Validation | Antibodies that are commercially available were validated by the manufacturers: Thermo anti-HA #26183 was validated by IP, IF, ICC, WB using mammalian cells/tissues and bacteria lysates. Sigma anti-FLAG F3165 was validated by IB, IP, ICC, IF, ELISA, ChIP, EIA, electron microscopy, flow cytometry. Thermo anti-V5 R960-25 was validated by WB, IHC, ICC, IF, ELISA, IP, ChIP, RIP and flow cytometry. Sigma anti-V5 V8137 was validated by IF, ICC, IP and WB. Santa Cruz anti-WAPL sc-365189 was validated by WB, IP, IF, ELISA, using human cultured cells. Cell signaling anti-gammaH2A.X antibody was validated by IF, WB and flow cytometry using mammalian cultured cells. Thermo anti-ATM #MA1-23152 was validated by IHC, WB using human cultured cells/tissues. Cell signaling anti-pS/TQ #6966 was validated by IP, WB, IHC, ChIP, IF using mammalian cultured cells and claimed to cross-react with C. elegans. Other antibodies were validated in previous studies (see above). |

## Eukaryotic cell lines

Policy information about cell lines and Sex and Gender in Research

| | |
|---|---|
| Cell line source(s) | No new mammalian cell lines were generated in the study. The HeLa cells used in the study were obtained from UC Berkeley cell culture facility. |
| Authentication | No formal cell line authentication was carried out by the authors, only visual examination of the cell identity were performed every day during the cells were in use. Regular verification of the identities of cell lines was performed in the cell culture facility. |
| Mycoplasma contamination | Tested negative before being used in the study. |
| Commonly misidentified lines (See ICLAC register) | No misidentified lines were used in the study. |

## Animals and other research organisms

Policy information about studies involving animals; ARRIVE guidelines recommended for reporting animal research, and Sex and Gender in Research

| | |
|---|---|
| Laboratory animals | This study utilized the invertebrate model organism C. elegans. Details of the genetic backgrounds of all alleles used in this study were listed in Source Data Table 1 in the manuscript. |
| Wild animals | No wild animals used in the study. |
| Reporting on sex | C. elegans can develop as one of two sexes, male or hermaphrodite, the latter being the major form of the organism. We analyzed C. elegans hermaphrodites (young adults) as a general practice for meiosis studies using this organism unless addressing male-specific questions. |
| Field-collected samples | No field-collected samples in the study. |
| Ethics oversight | C. elegans as a non-vertebrate model organism has no ethical restrictions. |

Note that full information on the approval of the study protocol must also be provided in the manuscript.

