## [Peer Review File · Nature Structural & Molecular Biology]

Peer Review Information

Manuscript Title: TM signaling modulates cohesin behavior in meiotic prophase and proliferating cells

Corresponding author name(s): Abby F. Dernburg

Reviewer Comments & Decisions:

Decision Letter, initial version:
--

Message: 6th Apr 2022

Dear Dr. Yu,

Thank you again for submitting your manuscript "DNA damage signaling modulates cohesin activity in meiotic prophase and proliferating cells". I apologize for the delay while we awaited the reports (copied below) from the 3 reviewers who evaluated your paper. In light of their reports, we remain interested in your study and would like to see your response to the comments of the referees, in the form of a revised manuscript.

I hope that you will be pleased to see that all 3 reviewers are quite positive about the potential interest of the work and appreciate the unexpected findings. Each also highlights aspects of the data or its presentation that should be strengthened by revisions, and makes specific suggestions to improve the work.

Reviewer #1 finds that more precise language is needed to more accurately reflect conclusions that are directly supported by data (including the title), and that the findings need to be placed in context of literature. They also suggest improvements to the model and request more rigorous statistical reporting. Reviewer #2 also suggests more nuanced wording of some aspects of the text, as well as additional data to strengthen the conclusions—particularly regarding the assays in human cells. They note that limitations of the mammalian system make it difficult to obtain compelling supporting data, but offer suggestions of how images showing accumulation of DNA damage markers may be improved, and note that quantitation is required to critically evaluate these data. Reviewer #3 raises concerns about the experimental design and approaches that limit the strength of the conclusions that can be drawn, and finds that the data obtained from the inducible Degron system should be supported by genetic analyses and, in some cases, additional controls. The reviewer concurs that the statistical analysis needs to be more rigorous and that the microscopic images are not always clear, and makes suggestions to improve data presentation. Editorially, we agree that these suggestions would strengthen the study, and ask that they be included in a revised manuscript.

Please be sure to address/respond to all concerns of the referees in full in a point-by-point response and highlight all changes in the revised manuscript text file. If you have comments that are intended for editors only, please include those in a separate cover letter.

When revising the manuscript, please bear in mind the following guidelines for our journal's Article format:

- abstract should be maximum 150 words, no references
- main text is typically between 3,000 and 4,500 words, and should be organized as introduction, results (with subheadings) and discussion.
- display items (figures and tables): typically between 6 and 8.
- supplementary items: Supplementary Figures should be a maximum of 10; other allowed supplementary items are Suppl Table, Note, Video, Data Set.
- uncropped images of gels and blots should be presented in a Supplementary Data Set.

We expect to see your revised manuscript within 6 weeks. If you cannot send it within this time, please contact me to discuss an extension; we would still consider your revision, provided that no similar work has been accepted for publication at NSMB or published elsewhere.

Reporting Summary:

When submitting the revised version of your manuscript, please pay close attention to our [href="https://www.nature.com/nature-research/editorial-policies/image-integrity">Digital Image Integrity Guidelines. and to the following points below:](https://www.nature.com/nature-research/editorial-policies/image-integrity)

- that unprocessed scans are clearly labelled and match the gels and western blots presented in figures.
- that control panels for gels and western blots are appropriately described as loading on sample processing controls

-- all images in the paper are checked for duplication of panels and for splicing of gel lanes.

FOR MS WITH CROPPED GELS: Please note that all key data shown in the main figures as cropped gels or blots should be presented in uncropped form, with molecular weight markers. These data can be aggregated into a single supplementary figure item. While these data can be displayed in a relatively informal style, they must refer back to the relevant figures. These data should be submitted with the final revision, as source data, prior to acceptance, but you may want to start putting it together at this point.

Data availability: this journal strongly supports public availability of data. All data used in accepted papers should be available via a public data repository, or alternatively, as Supplementary Information. If data can only be shared on request, please explain why in your Data Availability Statement, and also in the correspondence with your editor. Please note that for some data types, deposition in a public repository is mandatory - more information on our data deposition policies and available repositories can be found below: <https://www.nature.com/nature-research/editorial-policies/reporting-standards#availability-of-data>

Please use the link below to submit your revised manuscript and related files when they are ready:

[Redacted]

With kind regards,

Beth

Beth Moorefield, Ph.D.
Senior Editor
Nature Structural & Molecular Biology

Referee expertise:

Referee #1: DDR/meiotic genome stability

Referee #2: DDR/mammalian meiosis

Referee #3: C. elegans meiosis/cohesin

Reviewers' Comments:

Reviewer #1:

Remarks to the Author:

Meiotic chromosomes assume a highly ordered architecture that is important for meiotic recombination and chromosome segregation. In this manuscript, Yu and Dernburg show that the DNA-damage kinase ATM regulates the deposition of meiotic cohesin complexes in *C. elegans* by affecting the chromosomal abundance of the cohesin modulator WAPL. The signal pathway involving ATM in this context is highly unusual because ATM acts downstream of CHK2 (an effector kinase that typically is activated by ATM) and independently of meiotic double-strand break formation. In addition, the effects of several other cohesion regulators are analyzed. The effect of ATM on WAPL and cohesin appears at least partially conserved in proliferating mammalian cells.

The results in worms are strong and for the most part support the conclusions. The finding that ATM acts through WAPL to regulate cohesin level is novel and is expected to be of substantial interest to a broad readership. However, several points about data presentation should be addressed. In addition, the discussion section is not well developed.

1. In general, the authors are strongly encouraged to use more precise language to avoid overstating their results. For example, the title makes a strong point about DNA-damage signaling modulating cohesin activity. This statement is imprecise in two ways. First, in

meiosis, this modulation actually does not seem to be DNA-damage dependent based on their analyses but depends on DSB-independent activation of ATM by CHK2. So, it would be more precise to say that ATM signaling modulates cohesin. Second, there is no analysis of cohesin activity in this manuscript. The effect is on cohesin binding to chromosomes. Unless the authors present analysis of cohesin activity such as sister chromatid cohesion or loop formation, this statement also needs to be toned down.

2. The first three paragraphs of the results section should be moved to the introduction because they are largely restating published work and do not present primary data. Along those lines, the fact that WAPL-1 is downregulated at meiotic entry has been shown several years ago and should not be used as the title of the first section. The section title (and the associated caption of figure 1) should state the novel finding, i.e. that CHK-2 mediates the downregulation of WAPL-1.

3. Please also reword the conclusion of the last section because there is no direct data about WAPL downregulation in mammalian cells. Do chromosomal WAPL levels in fact decrease upon ATM activation as is seen in worms? Please provide this analysis to support the statement in the title of this section. Otherwise, the statements in this section claiming downregulation should be reworded.

4. The discussion section needs a substantial amount of editing to better place the findings in context. In the current version, a large number of facts are listed without clear explanation of how they connect to the presented observations. The overall lack of logical flow makes the discussion exceedingly difficult to read.

5. The model figure (figure 9) needs work. The positive and negative roles of Pds5, Eco1 and Wapl on cohesin binding are not intuitive. It is also unclear what the point of the arrows is. The arrows showing loop extrusion are the same in all the images and the presented data do provide not evidence that loop extrusion (or cohesin clustering) is altered by these factors in meiosis or upon somatic DNA damage. I suggest leaving out the arrows and clarifying the observed effects on cohesion deposition.

6. For all statistical analyses that involve multiple t-tests on the same dataset, please include a correction for multiple hypothesis testing (e.g. Bonferroni correction).

Minor points:

1. Please provide a callout to Figure S4 when first discussing the use of spo11-AID as a way to test DSB dependence. This panel shows that the AID allele is a complete null which is essential information for drawing the conclusions about DSB dependence.

2. Lines 231-232 and 236-237 are somewhat repetitive.

3. I think calling the N-terminal region of WAPL an SCD cluster domain is inappropriate. The definition of an SCD is 3 or more SQ or TQ sites within 100 amino acids. None of the examples in Figure 2A match this definition. This loose interpretation of the SCD is also confusing because the definition of the human SCD differs between panels 2A and 8G. To avoid confusion in the field I strongly suggest using a different terminology.

4. Please clarify that the MEDC deletion was performed in the overexpression plasmid.

Reviewer #2:

Remarks to the Author:

The manuscript provides new insights into the regulation and function of cohesins in relation to chromosome axis formation and recombination in meiosis in *C.elegans*. Cohesin complexes exist in multiple flavours in meiosis. The regulation and the function of distinct meiotic complexes are poorly understood despite their importance. The manuscript makes the exciting discovery that CHK-2 and ATM-1 kinases inhibit the cohesin antagonist WAPL-1 thereby enabling cohesin accumulation in chromosome cores. Interestingly, CHK-2 and ATM-1 do not require DNA break for this function which is surprising given that ATM-1 is normally activated by DNA breaks. The manuscript also reveals that two distinct cohesin reinforcing pathways, (1) a PDS-5-mediated stabilization of REC-8 cohesin and (2) enhancement of COH-3/4 cohesin by WAPL-1 inhibition, redundantly enable chromosome axis formation. These discoveries are highly significant for an understanding of cohesin control in meiosis. They will likely interest researchers in the cell-cycle and recombination fields. High-end genetics is used and the experiments seem to be done according to high standards. Therefore, the data are high quality, and they mostly support the conclusions. Accordingly, the presented experiments are, in principle, sufficient for the main messages of the manuscript. Nevertheless, some of the interpretations should be nuanced to allow for alternatives, and the authors may also optionally choose to add further experiments to strengthen their message.

Below I list a few points where the manuscript could be improved:

1.

I am most concerned about the conclusions of the human cell culture experiments. In my opinion, it would be better to omit some of these experiments as they are not absolutely essential for the main meiotic analysis. Further, because of the limitation of the experimental system, the results of the human cell culture experiment is only consistent with, but cannot strongly support, the main conclusion of the manuscript. If these experiments are kept in the manuscript, more convincing analysis of the RAD21 foci and a more nuanced discussion would be needed.

These experiments are problematic on two levels:

First, these experiments cannot rigorously test if ATM-mediated phosphorylation of WAPL leads to RAD21 accumulation at DNA repair sites, because these experiments do not test the effect of ATM phosphorylation on endogenous WAPL (this would be very difficult to test). These experiments only test if overexpression of various forms (WT, non-phosphorylatable or phospho-mimetic mutants) of WAPL-1 is sufficient to interfere with normal cohesion accumulation at DNA repair sites. This is not equivalent to testing if phosphorylation of endogenous WAPL-1 is the main driver of cohesion accumulation at DNA damage sites under otherwise unperturbed conditions. Thus, the current experimental design allows only very limited conclusions.

Second, based on the images in Fig. 8, the quantification of RAD21 foci seems very difficult in somatic cells due to the strong non-homogenous RAD21 background on chromatin. At the resolution that is available in the pdf from the journal's website, it is not immediately obvious if RAD21 reproducibly accumulated at sites of DNA damage and how the overexpression of various versions of WAPL affected RAD21 focus formation. It might be more convincing to present quantification of the enrichment of RAD21 signal in gamma H2A.X rich chromatin relative to gamma H2A.X-poor chromatin instead of RAD21 focus numbers, given that it is very difficult to see what is a RAD21 focus and what is only

background. One may also show the percentage of gamma H2AX rich foci where RAD21 is enriched above background levels. In my opinion these alternative measures would be necessary to make more convincing conclusions about WAPL-1's and ATM-1's role in RAD21 accumulation at DNA break sites.

Minor issues:

1.

Fig1.

In the meiosis field, it is customary to state that homologous chromosomes and sister chromatids segregate in the first and second division, but, strictly speaking, this statement is inaccurate. Homolog chromosomes and sister chromatids do not exist after the first meiotic prophase as DNA molecules are recombined. It would be more accurate to state that dyads of recombined chromatids and single chromatids segregate in meiosis I and II, respectively. For the same reasons, in Fig. 1, it is confusing to state that COH-3/4 localizes at the interface between homologous chromosomes. This gives the impression that COH-3/4 was repurposed to connect homologous DNA molecules instead of sister DNA molecules. To the best of my knowledge this is not the case. Therefore, I would rather state that COH-3/4 localizes to the interface between chromatid dyads that segregate in meiosis I, and REC8 localizes to the interface between chromatids that segregate in meiosis II.

2.

Line 130.

"WAPL-12A showed defective WAPL-1 downregulation (Fig. 2E), despite normal levels of CHK-2 (Fig. S2A-S2C) and ATM-1 (Fig. S2D-S2F)."

The supplementary figure does not show CHK-2 or ATM-1 levels, further bulk phosphorylation levels cannot prove that the phosphorylation activity of these kinases is normal as it cannot test if patterns of phosphorylation are normal. Hence a more nuanced conclusion is needed. I would rephrase the sentence:

"WAPL-12A showed defective WAPL-1 downregulation (Fig. 2E), despite the presence/seemingly normal levels of CHK-2 (Fig. S2A-S2C) and ATM-1 (Fig. S2D-S2F) activity. "

3.

Line 166-167

"atm-1S1853D showed high ATM-1 activity even in the absence of CHK-2, supporting the conclusion that CHK-2 activates ATM-1 through this site (Fig. 4B-4D)."

This sentence overstates the levels of ATM-1 activity. According to figure 4c the ATM-1 activity is considerably reduced in the double mutant as compared to WT or the single ATM-1 mutant indicating that CHK-2-mediated activation of ATM-1 involves both the s1853 site and also other unidentified mechanisms. Is it possible that CHK-2 activates ATM-2 also by alternative means? Could CHK-2 phosphorylate alternative sites on ATM-1, or does lack of DSB cause a drop in ATM activity in the atm-1S1853D chk-2AID double mutant? The observation that loss of SPO11 does not affect WAPL-1 levels suggest that DSB dependent ATM-1 activation has little role (Fig. S1). Was atm-1S1853D Spo-11 double mutant tested, is it similar to atm-1S1853D or atm-1S1853D chk-2AID? Related to this point, it would be appropriate to show statistical comparisons between atm-1S1853D chk-2AID and both chk-2AID and atm-1S1853D in fig. 4c and 5c. Currently, the double mutant is compared only to a one of the single mutants.

4.

A central conclusion of the manuscript is that CHK-2 mediated phosphorylation of ATM-1 activates ATM-1, which, in turn, drives WAPL-1 depletion from chromosomes by phosphorylating WAPL-1 in meiosis. This conclusion rests on the observation that a nonphosphorylatable mutant of WAPL-1 accumulates on chromatin in meiosis, and that the constitutively active/phosphomimetic ATMS1853D enables WAPL-1 depletion in the absence of CHK-2. However, it is formally possible that the phosphomimetic ATMS1853D mutant is able to induce WAPL-1 depletion in the absence of CHK-2 without directly phosphorylating WAPL-1 in its SCD. Whereas this is not the most straightforward scenario, it may be prudent to exclude it by testing if *atm-1S1853D wapl-12A* double mutant resembles *wapl-12A*. The prediction is that WAPL-12A should appear and COH-3/4 should disappear in meiosis in the double mutant if the model of the manuscript is correct. I note that there is no strong reason to doubt that ATM-1 primarily regulates WAPL-1 by phosphorylating the SCD within WAPL-1 as suggested in the manuscript. Therefore, examination of *atm-1S1853D wapl-12A* is not absolutely essential albeit it would strengthen the conclusions in my opinion.

5.

Lines 194-198:

"co-depletion of ECO-1 and ATM-1 or depletion of ECO-1 in *wapl-12A* showed synergistic effects on COH-3/4 localization (Fig. S5A-S5C), nearly recapitulating the effects of CHK-2 depletion (Fig. 6A-6C)."

Incorrect reference to Fig. S5A-S5C. There is no co-depletion shown in this figure. The sentence should refer only to 6A-C. Further, it would be much better to show the quantifications of Fig. S5C and 6C in the same graph. This would allow more relevant statistical comparisons. For example, it would be meaningful to directly compare *atm-1AID eco1AID* double mutant with *atm-1AID* in addition to *eco-1AID* given that the experiment seems to mainly aim to test the effect of ECO-1 depletion in an *atm-1AID* background.

6.

Lines 203-204

"These results suggest that CHK-2 promotes cohesin stabilization through at least two mechanisms, including downregulation of WAPL-1 and acetylation by ECO-1."

This conclusion is not fully accurate. The experiments are consistent with a possible activation of ECO-1 by CHK-2 but they did not directly test if CHK-2 promoted cohesin stability and acetylation by controlling ECO-1. The experiments only showed that CHK-2 loss and ECO-1 activity/acetylation-mimetic modifications of SMC-3 has opposing consequences on COH-3/4 levels on chromatin. Contrary to the conclusion in the manuscript it is also possible that CHK-2 activates pro-cohesion pathways that are independent of ECO-1 (for example a potential target could be the cohesin loader SCC-2/NIPBL). Thus, it is possible that ECO-1 mediated acetylation is not directly regulated by CHK-2. Hence a more nuanced conclusion is needed.

7.

Connected to the previous point, if CHK-2 controls both ECO-1 and WAPL-1, then COH-3/4 levels were predicted to be similar in *chk-2AID smc3QQ*, *atm-1AID smc3QQ* and *wapl-12a smc3QQ*. This is because presumed loss of ECO-1 activity in the absence of CHK-2 would not alter cohesin stability due to the presence of an acetylation-mimetic SMC-3 form.

Thus, like *atm-1AID* and *wapl-12a*, *chk-2AID* would antagonize cohesion only by activating WAPL-1 in the *smc3QQ* background. To the contrary, if *chk-2AID* antagonized cohesion by an *ECO-1* independent pathway in addition to enabling abnormal WAPL-1 activation, then we would expect to see lower COH-3/4 levels in *chk-2AID smc3QQ* as compared to *atm-1AID smc3QQ* and *wapl-12a smc3QQ*.

Reviewer #3:

Remarks to the Author:

The manuscript by Yu and Dernburg reports that assembly of the meiotic chromosomal axis around which chromatin loops are organized is promoted at the onset of meiosis in *C. elegans* by a DNA damage response (DDR) signaling pathway in which the kinase ATM downregulates the cohesin regulator WAPL-1 (WAPL or Rad61 in other organisms) to stabilize cohesin. *ECO-1*, the worm ortholog of an acetyltransferase known to stabilize chromosomal cohesin complexes in other organisms, and *PDS-5*, a cohesin-associated protein with complex roles in cohesin stabilization and destabilization, are also shown participate in axis assembly by stabilizing cohesin associated with meiotic axes. Finally, experiments in cultured HeLa cells suggest that DDR signaling performs a conserved function, downregulating WAPL to facilitate cohesin loading and hence break repair.

The findings are potentially of considerable interest and importance to the fields of meiosis and DNA damage repair. However, there are some major issues that must be resolved before publication. Perhaps more importantly, certain features of the experimental design result in data that are less rigorous than would have been possible using conventional methods that are standard in the field. This raises concerns about the conclusions and must be justified in the text, if possible, and demands controls that are not included in the current manuscript.

Note that the line numbering is different in the PDF and MS Word versions of the documents available through web portal. Line numbers used below relate to the PDF version.

Major points:

1. Nearly every experiment conducted in *C. elegans* makes use of an auxin-inducible degron (AID) system to deplete proteins in the germline. In certain cases, this approach makes sense: the AID system, like RNAi, allows analysis of meiotic roles of genes that have essential functions in mitosis or development. For example, *C. elegans* *PDS-5/EVL-14* is required for gonadal development and fertility, making analysis of roles in meiosis difficult if not impossible using existing alleles. However, in many cases in this manuscript, the approach is not justified because there are existing null alleles that give meiosis-specific defects and that have been used with great success in numerous studies of meiosis (examples include, *spo-11*, *chk-2*, *syp-3*, *atm-1*, *wapl-1*). Instead of these well characterized alleles, the authors use degron-tagged genes and then do not provide adequate characterization of the extent of depletion, the reproducibility of the knockdown, or how the AID-induced phenotype compares to that of existing null alleles. It is unclear why the authors chose this approach, when in most cases there is no apparent advantage over others. If the authors chose the AID approach due to concerns about pleiotropy, they should state the evidence for pleiotropy and demonstrate that the AID approach circumvents this problem and yields data that are at least as good if not better than

existing alleles. Ideally, this would be done for each target depleted by AID.

2. Even when justified, the degron experiments need to be treated similarly to an experiment using a pharmacological inhibitor or RNAi knockdown, or at least like the characterization of a new allele. How effective is the knockdown (western blot or immunofluorescence)? How reproducible is the knockdown (controls need to be included in each experiment. Typically, AID-mediated knockdown should be treated like a partial loss-of-function mutation – indeed, there has been considerable interest in newer generations of the degron tag and of the auxin analog that allow faster and more complete knockdown than the version of the system used here. Critically, strains being compared should be treated in parallel in the same experiment using the same batch of auxin/IAA-containing plates, etc, and this should be stated explicitly in the methods section.

3. The methods state that nuclei from 2-4 gonads were quantified. Analysis of nuclei from two gonads is unquestionably insufficient to capture any animal-to-animal variability within the experiment. Variability between animals is unavoidable when staining worm germlines, since not all dissections are equally successful. It is of even greater concern when using drug treatment or RNAi to knock down proteins.

4. It has been shown in *C. elegans* and other organisms that simultaneous RNAi knockdown of multiple targets can result in less depletion of each target than observed when knocked down individually. A similar phenomenon could occur when multiple targets are knocked down simultaneously by the AID degron system. Controls (western blots or IF of the targets) are needed to show this is not the case.

5. Insufficient information is given regarding the strains and alleles used. Supplementary Table S1 appears to list only new strains generated in this study. Other strains are not described anywhere in the methods (for example, *spo-11(AID)*, *syp-3(AID)* and others). The authors need to include allele numbers, strain identifiers, complete genotypes, and references for all strains used in the study. For strains created in this study, the authors should include strain numbers and complete genotypes. The authors statement in the reporting summary that “Details of the genetic backgrounds of all alleles used in this study were listed in Table S1 in the manuscript” is incorrect.

6. The AID system involves adding a degron tag to each protein of interest. Although small, it is possible that addition of the tag impacts the function of the protein even in the absence of IAA. This needs to be addressed by one (or ideally more) of the following: Does the knock-in create a high incidence of males phenotype indicative of meiotic chromosome segregation defects? Does the knock-in result in embryonic lethality? Are the localization and/or levels of the tagged protein altered compared to wt? For *CHK-2* and *ATM-1*, does addition of the tag change the levels of substrate phosphorylation (phospho-HIM/ZIM or pSQ/TQ)?

7. A similar concern regarding protein stability/folding/localization applies to the non-phosphorylatable/phosphomimetic/non-acetylatable mutants.

8. Based on consensus motifs, homology to known phosphosites in orthologs, and point mutations to make unphosphorylatable, unacetylatable, and phosphomimetic versions of proteins, it is suggested that *CHK-2* phosphorylates *ATM-1*, *ATM-1* phosphorylates *WAPL-1*, *ECO-1* acetylates *SMC-3*, etc. While these models seem reasonable, data proving the existence of these post-translational modifications (mass spec, modification-specific antibodies, etc.) would greatly strengthen the paper.

9. The phenotypes analyzed stop at levels of *REC-8* and *COH-3/4* cohesin. Do any of the changes observed have consequences on DSB repair, interhomolog crossover recombination and bivalent formation, meiotic chromosome segregation, generation of haploid gametes, embryonic viability? How relevant are the increases and/or decreases in the intensity of the proteins analyzed to reproductive fitness?

10. Lines 236-237: “Importantly, axial localization of *COH-3/4*, *REC-8*, and *HTP-3* was

restored by co-depletion of WAPL-1 (Fig. 7G-7I).” I may have missed it, but I did not see the data for COH-3/4.

11. Regarding statistical analysis and p-values: According to the methods, statistical comparisons were done using the Student’s t-test unless otherwise specified. It is possible that I missed it, but I did not see anywhere where another test was used. It is not clear that the t-test is the best suited test for the quantification shown here. Is the data normal? Might a non-parametric test be better?

12. Validation of reagents: in the reporting summary, the authors state that antibodies that are commercially available were validated by the manufacturers, and other antibodies were validated in previous studies. This is only sometimes true. For example, commercially available antibodies rarely validated by the manufacturers by demonstrating absence of staining in mutants or other strains lacking the target. Occasionally, this has been done by the company or academic labs. If antibodies have been validated in *C. elegans*, references should be included.

13. The DAPI signal is very difficult to see in Fig. 1C, Fig. 1E, and most figures where it is shown. Often, the blue channel can be easier to see if the hue is changed to one that appears more like DAPI viewed through a microscope (i.e. lighter blue, not “true blue.” As shown, the DAPI staining does not add much because it does not show colocalization of anything with chromosomes. In fact, it hurts because it makes colocalization between other proteins more difficult to see. A separate panel showing DAPI in greyscale would be more helpful than the overlay. Alternatively, perhaps a merge of just the protein staining without DAPI would be easier to interpret.

14. Fig. 1G shows that the intensity of REC-8 staining in *wapl-1; chk-2* mutants is significantly higher than in controlAID animals, suggesting that WAPL-1 destabilizes REC-8 even when it is not detectable. The authors should discuss this (if they do not already do so).

15. Fig. 6C seems to be missing important controls: *atm-1(AID)* single mutant and *wapl-1(2A)* single mutant. These have been shown elsewhere, but given the nature of alleles being used, it is important that the experiments be done in parallel at the same time.

16. All figure legends would be more helpful if they stated a result instead of what is shown.

Other comments:

1. It is unclear whether ATM-1(KA) is a mutation to alanine of the same residue altered in ATM-1(S1853D). I assume this is the case, but the nomenclature is confusing.

2. The authors need to be careful when making statements like “Mutation of this putative phosphosite greatly reduced pS/T-Q immunofluorescence in meiotic nuclei, while the corresponding phosphomimetic mutation resulted in normal ATM-1 activity” (lines 164-165). The mean pS/TQ intensity shown for *atm-1(D1853D)* in figure 4C is quite a bit lower than that in controlAID. If the authors wish to make this claim, they should at the very least show that the difference is not statistically significant. Similar potentially misleading statements occur elsewhere.

3. Line 169-170: the statement, “These findings indicate that the persistence of WAPL in meiotic nuclei lacking CHK-2 is a consequence of a failure to activate ATM-1” comes after a paragraph analyzing pS/T-Q IF in different *atm-1* mutants. The pS/T-Q analysis does not indicate anything about WAPL levels, and the authors need to be careful in their use of the word indicate throughout the manuscript. It is often used when substantially weaker wording like “suggests that XYZ may be a consequence...” would be much more appropriate.

4. Lines 177-178: “...indicating that CHK-2 promotes breaks directly rather than through

- ATM activation." Even if the requirement for CHK-2 in DSB formation is not mediated through ATM, it need not be direct.
5. The ControlAID treatment is never described. It also is not described how mutants like the non-phosphorylatable and putative phosphomimetic mutants were handled. Were they grown on plates containing IAA even though they were not degron tagged? If not, appropriate controls need to be included – one cannot compare, for example, atm-1(AID) grown on auxin plates to a mutant grown on normal plates and draw any conclusion (even a conclusion that the two strains appear the same is fraught).
 6. Lines 488-490: "Unless otherwise indicated, hermaphrodites at the L4 stage were transferred to seeded plates containing 1 mM IAA and incubated for 24 hours..." I may have missed it, but I did not see anyplace where other methods for IAA treatment were indicated.
 7. Lines 165-166: Fig. 4A and 4B should be 4B and 4C.
 8. Line 195: synergistic effects are usually substantially stronger than additive/multiplicative effects. The data shown do not appear to support any synergism.
 9. Statistical significance is shown categorically (e.g. n.s., *, **, ***, ****). The journal's editorial policies include "Give P values as exact values whenever suitable," and this would indeed be suitable and better for the data in this manuscript.
 10. Image analysis: what radius was used for the rolling ball tool used for image analysis? What was done to determine that this setting was appropriate?
 11. Fig. 1A and elsewhere: in some figures/legends, premeiosis and early meiosis are shown. In others, meiotic entry and midpachytene are shown. Is meiotic entry the same as early meiosis? Can midpachytene be indicated on the figure? It is nitpicky, but the Leptotene/Zygotene label seems like it is too far proximal and overlaps with where midpachytene would be found.
 12. Fig. 3A: it would be helpful in the legend to state which CHK2 orthologs are meiosis-specific and which are not.
 13. Fig. 3B: here and elsewhere, state what stage is being shown.
 14. Fig. 4C: pS/TQ staining in atm-1(S1853D) is lower than in controlAID. Is the difference significant? This potentially lends credence to the concern that the point mutations are destabilizing the protein or otherwise impacting function in unexpected ways.
 15. Line 413: "The conserved arginine and phospho-serine/threonine site." It should not be called a conserved phosphosite unless phosphorylation has been demonstrated.
 16. Figure 8, all IF panels: it is very difficult to see the boxed region indicating the part of the image from which the inset was taken.
 17. Figure 8: CONi and siCON should be defined in the legend, and the methods should clearly state what the controls were.
 18. Fig S3 is never referred to in the text.

Author Rebuttal to Initial comments
--

Reviewers' Comments:

Reviewer #1:

Remarks to the Author:

Meiotic chromosomes assume a highly ordered architecture that is important for meiotic recombination and chromosome segregation. In this manuscript, Yu and Dernburg show that the DNA-damage kinase ATM regulates the deposition of meiotic cohesin complexes in *C. elegans* by affecting the chromosomal abundance of the cohesin modulator WAPL. The signal pathway involving ATM in this context is highly unusual because ATM acts downstream of CHK2 (an effector kinase that typically is activated by ATM) and independently of meiotic double-strand break formation. In addition, the effects of several other cohesion regulators are analyzed. The effect of ATM on WAPL and cohesin appears at least partially conserved in proliferating mammalian cells.

The results in worms are strong and for the most part support the conclusions. The finding that ATM acts through WAPL to regulate cohesin level is novel and is expected to be of substantial interest to a broad readership. However, several points about data presentation should be addressed. In addition, the discussion section is not well developed.

1. In general, the authors are strongly encouraged to use more precise language to avoid overstating their results. For example, the title makes a strong point about DNA-damage signaling modulating cohesin activity. This statement is imprecise in two ways. First, in meiosis, this modulation actually does not seem to be DNA-damage dependent based on their analyses but depends on DSB-independent activation of ATM by CHK2. So, it would be more precise to say that ATM signaling modulates cohesin. Second, there is no analysis of cohesin activity in this manuscript. The effect is on cohesin binding to chromosomes. Unless the authors present analysis of cohesin activity such as sister chromatid cohesion or loop formation, this statement also needs to be toned down.

We agree that the title may be confusing to many readers and have now changed it to “ATM signaling modulates cohesin behavior in meiotic prophase and proliferating cells.” We acknowledge that our cytological assays measure the abundance of cohesin along chromosomes, rather than directly measuring its biochemical activities, and have changed the language in the title and elsewhere to avoid implying otherwise.

2. The first three paragraphs of the results section should be moved to the introduction because they are largely restating published work and do not present primary data. Along those lines, the fact that WAPL-1 is downregulated at meiotic entry has been shown several years ago and should not be used as the title of the first section. The section title (and the associated caption of figure 1) should state the novel finding, i.e. that CHK-2 mediates the downregulation of WAPL-1.

We appreciate this suggestion and have moved the text describing previous results from the Results to the Introduction.

We have also changed the headings in the Results section and the caption of Figure 1 to better reflect the novel aspects of our findings. They are now: “CHK-2 activity leads to the downregulation of WAPL-1 at meiotic entry” and “CHK-2 is required for downregulation of WAPL-1 at meiotic entry” respectively.

3. Please also reword the conclusion of the last section because there is no direct data about WAPL downregulation in mammalian cells. Do chromosomal WAPL levels in fact decrease upon ATM activation as is seen in worms? Please provide this analysis to support the statement in the title of this section. Otherwise, the statements in this section claiming downregulation should be reworded.

We have changed the last heading in the Results section to “ATM-dependent phosphorylation of WAPL mediates cohesin concentration at mammalian DNA damage foci”.

We also revised the last Results section to avoid overstatement.

4. The discussion section needs a substantial amount of editing to better place the findings in context. In the current version, a large number of facts are listed without clear explanation of how they connect to the presented observations. The overall lack of logical flow makes the discussion exceedingly difficult to read.

We have revised the Discussion to improve its readability.

5. The model figure (figure 9) needs work. The positive and negative roles of Pds5, Eco1 and Wapl on cohesin binding are not intuitive. It is also unclear what the point of the arrows is. The arrows showing loop extrusion are the same in all the images and the presented data do provide not evidence that loop extrusion (or cohesin clustering) is altered by these factors in meiosis or upon somatic DNA damage. I suggest leaving out the arrows and clarifying the observed effects on cohesion deposition.

We revised the model figure according to reflect our main observations and removed the confusing arrows.

6. For all statistical analyses that involve multiple t-tests on the same dataset, please include a correction for multiple hypothesis testing (e.g. Bonferoni correction).

We thank the reviewer for noting this issue and performed *p*-value adjustment on all statistical analyses across the manuscript using *post-hoc* Bonferroni correction. We also now include *p*-values greater or equal to 0.01 in the figures, rather than denoting them as * or n.s. We revised the Methods section to reflect these changes.

Minor points:

1. Please provide a callout to Figure S4 when first discussing the use of spo11-AID as a way to test DSB dependence. This panel shows that the AID allele is a complete null which is essential information for drawing the conclusions about DSB dependence.

We thank the reviewer for noting this omission and have added a callout.

2. Lines 231-232 and 236-237 are somewhat repetitive.

We have revised these statements to eliminate redundancy.

3. I think calling the N-terminal region of WAPL an SCD cluster domain is inappropriate. The definition of an SCD is 3 or more SQ or TQ sites within 100 amino acids. None of the examples in Figure 2A match this definition. This loose interpretation of the SCD is also confusing because the definition of the human SCD differs between panels 2A and 8G. To avoid confusion in the field I strongly suggest using a different terminology.

We now mention this definition of “SCD,” and explained why we consider the WAPL sites to be a “mini-SCD” although they do not meet the strict definition.

4. Please clarify that the MEDC deletion was performed in the overexpression plasmid.

We revised the last paragraph of the Results to make this clearer.

Reviewer #2:

Remarks to the Author:

The manuscript provides new insights into the regulation and function of cohesins in relation to chromosome axis formation and recombination in meiosis in *C.elegans*. Cohesin complexes exist in multiple flavours in meiosis. The regulation and the function of distinct meiotic complexes are poorly understood despite their importance. The manuscript makes the exciting discovery that CHK-2 and ATM-1 kinases inhibit the cohesin antagonist WAPL-1 thereby enabling cohesin accumulation in chromosome cores. Interestingly, CHK-2 and ATM-1 do not require DNA break for this function which is surprising given that ATM-1 is normally activated by DNA breaks. The manuscript also reveals that two distinct cohesin reinforcing pathways, (1) a PDS-5-mediated stabilization of REC-8 cohesin and (2) enhancement of COH-3/4 cohesin by WAPL-1 inhibition, redundantly enable chromosome axis formation. These discoveries are highly significant for an understanding of cohesin control in meiosis. They will likely interest researchers in the cell-cycle and recombination fields. High-end genetics is used and the experiments seem to be done according to high standards. Therefore, the data are high quality, and they mostly support the conclusions. Accordingly, the presented experiments are, in principle, sufficient for the main messages of the manuscript. Nevertheless, some of the interpretations should be nuanced to allow for alternatives, and the authors may also optionally choose to add further experiments to strengthen their message.

We thank the referee for the positive comments and suggestions for improvement.

Below I list a few points where the manuscript could be improved:

1.

I am most concerned about the conclusions of the human cell culture experiments. In my opinion, it would be better to omit some of these experiments as they are not absolutely essential for the main meiotic analysis. Further, because of the limitation of the experimental system, the results of the human cell culture experiment is only consistent with, but cannot strongly support, the main conclusion of the manuscript. If these experiments are kept in the manuscript, more convincing analysis of the RAD21 foci and a more nuanced discussion would be needed.

These experiments are problematic on two levels:

First, these experiments cannot rigorously test if ATM-mediated phosphorylation of WAPL leads to RAD21 accumulation at DNA repair sites, because these experiments do not test the effect of ATM phosphorylation on endogenous WAPL (this would be very difficult to test). These experiments only test if overexpression of various forms (WT, non-phosphorylatable or phospho-mimetic mutants) of WAPL-1 is sufficient to interfere with normal cohesin accumulation at DNA repair sites. This is not equivalent to testing if phosphorylation of endogenous WAPL-1 is the main driver of cohesin accumulation at DNA damage sites under otherwise unperturbed conditions. Thus, the current experimental design allows only very limited conclusions.

We agree that it is essential to avoid overinterpretation of our experiments in human cells, but also feel that they are important to the paper since they indicate that cohesin regulatory mechanisms that underlie axis formation during meiosis are relevant in other contexts. We revised the last paragraph of the Results section to clarify the caveats of our experiments involving plasmid-based (over)expression of WAPL.

Second, based on the images in Fig. 8, the quantification of RAD21 foci seems very difficult in somatic cells due to the strong non-homogenous RAD21 background on chromatin. At the resolution that is available in the pdf from the journal's website, it is not immediately obvious if RAD21 reproducibly accumulated at sites of DNA damage and how the overexpression of various versions of WAPL affected RAD21 focus formation. It might be more convincing to present quantification of the enrichment of RAD21 signal in gamma H2A.X rich chromatin relative to gamma H2A.X-poor chromatin instead of RAD21 focus numbers, given that it is very difficult to see what is a RAD21 focus and what is only background. One may also show the percentage of gamma H2AX rich foci where RAD21 is enriched above background levels. In my opinion these alternative measures would be necessary to make more convincing conclusions about WAPL-1's and ATM-1's role in RAD21 accumulation at DNA break sites.

We agree that our image quantification method may be difficult for readers to understand or validate. We have now re-analyzed all image data from mammalian cells according to the reviewer's suggestion: instead of quantifying the number of RAD21 foci, we measured the ratio of RAD21 intensity within regions positive for DNA damage

markers (gamma-H2A.X and pS/TQ-) to similarly sized control regions in the same nuclei. This approach also supported our assertion that RAD21 accumulation at sites of damage is regulated by ATM and WAPL. These changes are reflected in our revised Figures 8C, 8D, 8I, 8J and 8K, and our description of the quantification in the Methods.

Minor issues:

1.

Fig1.

In the meiosis field, it is customary to state that homologous chromosomes and sister chromatids segregate in the first and second division, but, strictly speaking, this statement is inaccurate. Homolog chromosomes and sister chromatids do not exist after the first meiotic prophase as DNA molecules are recombined. It would be more accurate to state that dyads of recombined chromatids and single chromatids segregate in meiosis I and II, respectively. For the same reasons, in Fig. 1, it is confusing to state that COH-3/4 localizes at the interface between homologous chromosomes. This gives the impression that COH-3/4 was repurposed to connect homologous DNA molecules instead of sister DNA molecules. To the best of my knowledge this is not the case. Therefore, I would rather state that COH-3/4 localizes to the interface between chromatid dyads that segregate in meiosis I, and REC8 localizes to the interface between chromatids that segregate in meiosis II.

We appreciate this point. In organisms with localized centromeres, the original sister centromeres remain connected through Meiosis I and only separate during Meiosis II. Additionally, in the absence of crossing-over, sister chromatids typically co-segregate during the first meiotic division. For these reasons, it is often stated that homologs separate in MI and sisters in MII. That said, we fully agree that the definition of “sisters” and “homologs” becomes imprecise after crossing-over has occurred, particularly in holocentric organisms such as *C. elegans*, and have revised Figure 1 and the language used to describe this process more accurately.

2.

Line 130.

“WAPL-12A showed defective WAPL-1 downregulation (Fig. 2E), despite normal levels of CHK-2 (Fig. S2A-S2C) and ATM-1 (Fig. S2D-S2F).”

The supplementary figure does not show CHK-2 or ATM-1 levels, further bulk phosphorylation levels cannot prove that the phosphorylation activity of these kinases is normal as it cannot test if patterns of phosphorylation are normal. Hence a more nuanced conclusion is needed. I would rephrase the sentence:

“WAPL-12A showed defective WAPL-1 downregulation (Fig. 2E), despite the presence/seemingly normal levels of CHK-2 (Fig. S2A-S2C) and ATM-1 (Fig. S2D-S2F) activity. “

We agree and have revised the text according to the reviewer’s recommendation.

3.

Line 166-167

“*atm-1^{S1853D}* showed high ATM-1 activity even in the absence of CHK-2, supporting the conclusion that CHK-2 activates ATM-1 through this site (Fig. 4B-4D).”

This sentence overstates the levels of ATM-1 activity. According to figure 4c the ATM-1 activity is considerably reduced in the double mutant as compared to WT or the single ATM-1 mutant indicating that CHK-2-mediated activation of ATM-1 involves both the s1853 site and also other unidentified mechanisms. Is it possible that CHK-2 activates ATM-2 also by alternative means? Could CHK-2 phosphorylate alternative sites on ATM-1, or does lack of DSB cause a drop in ATM activity in the *atm-1^{S1853D} chk-2^{AID}* double mutant? The observation that loss of SPO11 does not affect WAPL-1 levels suggest that DSB dependent ATM-1 activation has little role (Fig. S1). Was *atm-1^{S1853D} Spo-11* double mutant tested, is it similar to *atm-1^{S1853D}* or *atm-1^{S1853D} chk-2^{AID}*?

Related to this point, it would be appropriate to show statistical comparisons between *atm-1^{S1853D} chk-2^{AID}* and both *chk-2^{AID}* and *atm-1^{S1853D}* in fig. 4c and 5c. Currently, the double mutant is compared only to a one of the single mutants.

We thank the reviewer for noting this important issue. We agree that immunofluorescence of pS/TQ in *atm-1^{S1853D}; chk-2^{AID}* is reduced relative to *atm-1^{S1853D}* and have revised our statement accordingly. To determine whether CHK-2 promotes ATM activity in part through DSBs, we have now examined the staining in *atm-1^{S1853D}; spo-11^{AID}* following auxin treatment and observed a similar level of ATM-1 activity as in *atm-1^{S1853D}; chk-2^{AID}*. We thus agree that CHK-2 may promote ATM-1 both through direct phosphorylation and through a DSB-dependent mechanism, and mention this in the manuscript. As the reviewer notes, ATM-1 activity in the absence of DSBs is sufficient to downregulate WAPL-1.

These results are now included in Figure 4.

We also compared the *atm-1*^{S1853D}; *chk-2*^{AID} to each single mutant in Figure 4C and Figure 5C, as suggested by the reviewer.

4.

A central conclusion of the manuscript is that CHK-2 mediated phosphorylation of ATM-1 activates ATM-1, which, in turn, drives WAPL-1 depletion from chromosomes by phosphorylating WAPL-1 in meiosis. This conclusion rests on the observation that a nonphosphorylatable mutant of WAPL-1 accumulates on chromatin in meiosis, and that the constitutively active/phosphomimetic ATMS1853D enables WAPL-1 depletion in the absence of CHK-2. However, it is formally possible that the phosphomimetic ATMS1853D mutant is able to induce WAPL-1 depletion in the absence of CHK-2 without directly phosphorylating WAPL-1 in its SCD. Whereas this is not the most straightforward scenario, it may be prudent to exclude it by testing if *atm-1*^{S1853D} *wapl-12A* double mutant resembles *wapl-12A*. The prediction is that WAPL-12A should appear and COH-3/4 should disappear in meiosis in the double mutant if the model of the manuscript is correct. I note that there is no strong reason to doubt that ATM-1 primarily regulates WAPL-1 by phosphorylating the SCD within WAPL-1 as suggested in the manuscript. Therefore, examination of *atm-1*^{S1853D} *wapl-12A* is not absolutely essential albeit it would strengthen the conclusions in my opinion.

We addressed this insightful comment by performing the suggested experiment (see below). We found that the nonphosphorylatable WAPL-1^{2A} persisted on meiotic chromosomes even in the presence of constitutively active ATM-1^{S1853D}, supporting the conclusion that phosphorylation of WAPL-1 by ATM-1 is essential for its downregulation at meiotic entry. This new finding is included in the revised manuscript, Figure S5.

5.

Lines 194-198:

“co-depletion of ECO-1 and ATM-1 or depletion of ECO-1 in *wapl-12A* showed synergistic effects on COH-3/4 localization (Fig. S5A-S5C), nearly recapitulating the effects of CHK-2 depletion (Fig. 6A-6C).”

Incorrect reference to Fig. S5A-S5C. There is no co-depletion shown in this figure. The sentence should refer only to 6A-C. Further, it would be much better to show the quantifications of Fig, s5c and 6c in the same graph. This would allow more relevant statistical comparisons. For example, it would be meaningful to directly compare *atm-1AID eco1AID* double mutant with *atm-1AID* in addition to *eco-1AID* given that the experiment seem to mainly aim to test the effect of ECO-1 depletion in an *atm-1AID* background.

We appreciate the referee for catching this error. To address the concerns about our quantitative comparisons, we re-analyzed our image data and performed statistical comparisons between all genotypes in Figure 6 and Figure S5, to which we have added a comparison between *atm-1AID, eco-1AID, atm-1AID; eco-1AID* double and *eco-1AID; wapl-12A* double). We replaced the previous graph (Figure S5C) with the new version that includes all comparisons.

6.

Lines 203-204 “These results suggest that CHK-2 promotes cohesin stabilization through at least two mechanisms, including downregulation of WAPL-1 and acetylation by ECO-1.”

This conclusion is not fully accurate. The experiments are consistent with a possible activation of ECO-1 by CHK-2 but they did not directly test if CHK-2 promoted cohesion stability and acetylation by controlling ECO-1. The experiments only showed that CHK-2 loss and ECO-1 activity/acetylation-mimetic modifications of SMC-3 has opposing consequences on COH-3/4 levels on chromatin. Contrary to the conclusion in the manuscript it is also possible that CHK-2 activates pro-cohesion pathways that are independent of ECO-1 (for example a potential target could be the cohesion loader SCC-2/NIPBL). Thus, it is possible that ECO-1 mediated acetylation is not directly regulated by CHK-2. Hence a more nuanced conclusion is needed.

We agree with these comments and have revised the text accordingly.

7.

Connected to the previous point, if CHK-2 controls both ECO-1 and WAPL-1, then COH-3/4 levels were predicted to be similar in *chk-2AID smc3QQ*, *atm-1AID smc3QQ* and *wapl-12a smc3QQ*. This is because presumed loss of ECO-1 activity in the absence of CHK-2 would not alter cohesion stability due to the presence of an acetylation-mimetic SMC-3 form. Thus, like *atm-1AID* and *wapl-12a*, *chk-2AID* would antagonize cohesion only by activating WAPL-1 in the *smc3QQ* background. To the contrary, if *chk-2AID* antagonized cohesin by an ECO-1 independent pathway in addition to enabling abnormal WAPL-1 activation, then we would expect to see lower COH-3/4 levels in *chk-2AID smc3QQ* as compared to *atm-1AID smc3QQ* and *wapl-12a smc3QQ*.

We agree that it might be informative to compare COH-3/4 levels in various genetic backgrounds expressing the acetylation-mimetic SMC-3^{QQ}. However, we decided not to pursue these experiments because the strains would be extremely difficult to engineer and the results would likely be uninformative. As we note in the Discussion, in budding yeast, Eco1 has been found to acetylate not only Smc3 but also Mcd1/Scc1 (the kleisin subunit) when it is phosphorylated by Chk1, and this acetylation also antagonizes Wapl (Heidinger-Pauli et al, 2008, Mol Cell; Heidinger-Pauli et al, 2009, Mol Cell). If similar regulation occurs in *C. elegans*, this would complicate interpretation of the suggested experiments. We have now revised the manuscript to state that CHK-2 may promote ECO-1-dependent acetylation of cohesin subunits to antagonize WAPL-1 and may also regulate cohesin through other mechanisms.

Reviewer #3:

Remarks to the Author:

The manuscript by Yu and Dernburg reports that assembly of the meiotic chromosomal axis around which chromatin loops are organized is promoted at the onset of meiosis in *C. elegans* by a DNA damage response (DDR) signaling pathway in which the kinase ATM downregulates the cohesin regulator WAPL-1 (WAPL or Rad61 in other organisms) to stabilize cohesin. ECO-1, the worm ortholog of an acetyltransferase known to stabilize chromosomal cohesin complexes in other organisms, and PDS-5, a cohesin-associated protein with complex roles in cohesin stabilization and destabilization, are also shown participate in axis assembly by stabilizing cohesin associated with meiotic axes. Finally, experiments in cultured HeLa cells suggest that DDR signaling performs a conserved function, downregulating WAPL to facilitate cohesin loading and hence break repair.

The findings are potentially of considerable interest and importance to the fields of meiosis and DNA damage repair. However, there are some major issues that must be resolved before publication. Perhaps more importantly, certain features of the experimental design result in data that are less rigorous than would have been possible using conventional methods that are standard in the field. This raises concerns about the conclusions and must be justified in the text, if possible, and demands controls that are not included in the current manuscript. Note that the line numbering is different in the PDF and MS Word versions of the documents available through web portal. Line numbers used below relate to the PDF version.

Major points:

1. Nearly every experiment conducted in *C. elegans* makes use of an auxin-inducible degron (AID) system to deplete proteins in the germline. In certain cases, this approach makes sense: the AID system, like RNAi, allows analysis of meiotic roles of genes that have essential functions in mitosis or development. For example, *C. elegans* PDS-5/EVL-14 is required for gonadal development and fertility, making analysis of roles in meiosis difficult if not impossible using existing alleles. However, in many cases in this manuscript, the approach is not justified because there are existing null alleles that give meiosis-specific defects and that have been used with great success in numerous studies of meiosis (examples include, *spo-11*, *chk-2*, *syp-3*, *atm-1*, *wapl-1*). Instead of these well characterized alleles, the authors use degron-tagged genes and then do not provide adequate characterization of the extent of depletion, the reproducibility of the knockdown, or how the AID-induced phenotype compares to that of existing null alleles. It is unclear why the authors chose this approach, when in most cases there is no apparent advantage over others. If the authors chose the AID approach due to concerns about pleiotropy, they should state the evidence for pleiotropy and demonstrate that the AID approach circumvents this problem and yields data that are at least as good if not better than existing alleles. Ideally, this would be done for each target depleted by AID.

As the referee notes, we used the AID system extensively in this work to deplete proteins that are essential for meiosis, in addition to several that are required more

broadly for mitosis and thus for viability. We did this because we have found that AID-mediated depletion can often (albeit not always) quantitatively and reproducibly recapitulate the effects of a null mutation, and there are major advantages to using conditional alleles. First, loss-of-function mutations that result in inviability or infertility must be maintained in heterozygous animals over genetic balancers, which complicates strain construction and maintenance. Conditional alleles, which can be maintained in a homozygous state, greatly facilitate the construction of strains carrying additional mutations or markers, such as those used in many of the experiments reported in this work. In addition, the use of conditional alleles often allows controls to be performed in parallel using the identical strain, simply by omitting auxin treatment, which obviates concerns about potential differences in genetic background. In previous publications using this system we have documented quantitative degradation of several meiotic proteins, even in cases where up to 4 proteins were simultaneously targeted (Zhang et al, 2015, *Development*; Zhang et al., 2017, *eLife*). Thus, the AID system does not seem to be saturable as readily as the RNAi machinery, which is perhaps unsurprising given that cells constantly target many proteins for ubiquitin-mediated proteolysis.

For several of the key proteins studied in this work, there is clear prior evidence or a strong expectation of pleiotropic effects of null or strong loss-of-function mutations. Mutations affecting core cohesins or their regulators have been demonstrated to cause somatic phenotypes, as would be expected for genes that affect mitotic division (e.g. as reported/summarized in Liu NQ et al, 2020 *Nature Genetics*; Pauli A et al, 2010 *Current Biology*; Xiong B et al, 2010 *Annual Review of Biochemistry*; Kline AD et al, 2015 *Reproductive and Developmental Genetics*; Liu D et al, 2015 *BMC Plant Biology*). Unsurprisingly, null mutations in *C. elegans atm-1* result in genome instability (a “mutator phenotype”), mitotic defects, and subfertility (Jones et al., 2012; Stergiou et al., 2007). We have observed spo-11-independent DSBs in meiotic cells in *atm-1* mutants, which was also a key reason to avoid using such alleles.

2. Even when justified, the degon experiments need to be treated similarly to an experiment using a pharmacological inhibitor or RNAi knockdown, or at least like the characterization of a new allele. How effective is the knockdown (western blot or immunofluorescence)? How reproducible is the knockdown (controls need to be included in each experiment. Typically, AID-mediated knockdown should be treated like a partial loss-of-function mutation – indeed, there has been considerable interest in newer generations of the degon tag and of the auxin analog that allow faster and more complete knockdown than the version of the system used here. Critically, strains being compared should be treated in parallel in the same experiment using the same batch of auxin/IAA-containing plates, etc, and this should be stated explicitly in the methods section.

We appreciate the suggestions and have now added to the Methods section that we only compare results across the same batch of plates. This is, however, a minor issue since one key advantage of the AID system is the stability of indole acetic acid (IAA, natural auxin) to light, heat, and other environmental conditions. All of the degron-tagged alleles reported here have been validated by assays based on localization and/or function of the tagged protein; i.e., we verify that the introduction of the degron, usually along with an epitope tag, does not affect viability or reproduction (see below for more details), and we test whether the protein is effectively depleted using a suitable combination of western blotting, immunofluorescence, and/or functional assays.

We have used some of the “improved” auxin systems and find that their performance in the germline is generally inferior to the simple version that we developed and use routinely in the lab: we have generally not found degradation of germline proteins to be “leaky” (i.e., we do not detect degradation in the absence of TIR1 expression or auxin treatment) and we thus find it unnecessary and cumbersome to use additional transgenes that have been reported to suppress leaky degradation. Additionally, the kinetics of degradation in the germline appear to depend most strongly on the rate of penetration of auxin into the tissue, so the use of larger, more expensive, and/or less water-soluble auxin analogs is detrimental.

For proteins that are ubiquitously expressed, western blots can reveal the kinetics but not the full extent of depletion since the TIR1 constructs we use are (by design) germline-enriched or germline-specific, so residual degron-tagged protein may remain in the soma. Some of the proteins we analyzed, particularly SPO-11 and ATM-1, are also very low in abundance and thus cannot be detected reliably on westerns. We thus use a combination of immunofluorescence (detecting either the degron-tagged protein or markers for its activity) and phenotypic analysis to validate the efficacy of depletion. Specific examples are listed below.

SPO-11 cannot be detected on western blots unless we immunoprecipitate it from large batches of worm extract. However, RAD-51 foci can be readily detected in the germline and in most cases they depend on SPO-11 (although *atm-1* mutations or prolonged depletion results in SPO-11-independent RAD-51 foci). In Figure S4, we show that either **SPO-11** or **CHK-2** depletion eliminates detectable RAD-51 foci in meiotic nuclei, similar to the effects of null mutations (Dernburg *et al.*, 1998; Alpi *et al.*, 2003). Depletion of **CHK-2** also results in other hallmarks of *chk-2* loss-of-function, including the absence of polarized nuclei at meiotic entry. This is revealed by DAPI staining shown in several figures in the manuscript (e.g. Figure 3B). Depletion of CHK-2 also disrupts homolog pairing and synapsis, as seen in *chk-2* null mutants (MacQueen and Villeneuve 2001) (e.g. revealed in Figure 1E by HTP-3 staining). In addition, CHK-2 depletion can be monitored by western blot, as shown below:

We also validated the function of [HA + degron-tagged] CHK-2, as well as the efficacy of CHK-2 depletion, using an established marker for CHK-2 activity, a phospho-specific antibody that recognizes CHK-2 dependent phosphorylation of the pairing center proteins HIM-8 and ZIM-1/2/3 (Kim et al., 2015). This also addresses the reviewer's concern in comment #6 below as to whether the degron might perturb function in the absence of auxin.

WAPL-1 – Although WAPL-1 association with chromosomes declines at meiotic entry, the protein can be robustly detected in premeiotic germline nuclei using epitope tags or anti-WAPL antibodies. The degron-tagged protein is strongly depleted following auxin treatment. *wapl-1* null mutants are viable and fertile, but degron-mediated depletion of WAPL-1 recapitulates the effects on COH-3/4 during late prophase.

ATM-1 is also undetectable by western blots. We used a commercial anti-pS/TQ antibody to monitor ATM activity; however, this probe also detects phosphorylation by ATL-1 (ATR) and other kinases. The strong defect in WAPL-1 downregulation following ATM depletion (e.g. shown in Figure 2B and Figure 5A etc) recapitulates the effects of *atm-1* mutants. Our interpretations do not depend on 100% degradation.

ATL-1 (ATR) can be detected by western blot (as below). In addition, IF reveals foci of ATL-1 that colocalize with RAD-51 in the germline. These ATL-1 foci are not detected following auxin treatment (not shown here). ATL-1 depletion also results in 100% embryonic lethality, which is consistent with *atl-1* null alleles.

PDS-5 displays both nucleoplasmic and axial localization in the germline. PDS-5 tagged with V5 + degron shows the same distribution. Following auxin treatment, only background (cytoplasmic) staining is detected in the germline.

ECO-1 IF is robustly detected in germline nuclei using FLAG or OLLAS epitope tags. The signal is stronger in premeiotic nuclei than in meiotic cells. Auxin treatment greatly reduces the signal.

In most cases, degron-tagged proteins of interest were degraded rapidly, typically within one hour.

3. The methods state that nuclei from 2-4 gonads were quantified. Analysis of nuclei from two gonads is unquestionably insufficient to capture any animal-to-animal variability within the experiment. Variability between animals is unavoidable when staining worm germlines, since not all dissections are equally successful. It is of even greater concern when using drug treatment or RNAi to knock down proteins.

We routinely check that data are consistent across the individual gonads that we use for measurements. We have collected new images and re-analyzed the data that were originally obtained from only two gonads (Figure 2D and Figure 3E), and replaced the

quantification results with the new ones. We also revised the method section to state that data were quantified from at least 3 gonads for each experiment, which is consistent with standards in the field.

4. It has been shown in *C. elegans* and other organisms that simultaneous RNAi knockdown of multiple targets can result in less depletion of each target than observed when knocked down individually. A similar phenomenon could occur when multiple targets are knocked down simultaneously by the AID degnon system. Controls (western blots or IF of the targets) are needed to show this is not the case.

As mentioned above, we have previously demonstrated that multiple proteins can be targeted by the AID system simultaneously (e.g. in Zhang et al, 2018 eLife in which 2 proteins were co-depleted and Zhang et al., 2021 bioRxiv, in which 4 were simultaneously targeted). In the current study, co-depletion of 2-3 proteins of interest simultaneously was validated by the observed phenotypes. For example, in Figure 7 we co-depleted PDS-5, CHK-2 and WAPL-1, and included controls for this triple depletion:

When adding auxin, both CHK-2 (tagged with HA-AID, showed by HA staining) and PDS-5 (tagged with FLAG-AID, showed by FLAG staining) were efficiently depleted simultaneously as below. We were also able to tell that CHK-2 was depleted efficiently by observing that crescent-shaped nuclei were absent from meiotic entry (by DAPI staining).

When WAPL-1 was also tagged with the degnon, it was effectively co-depleted along with PDS-5^{AID} and CHK-2^{AID}.

5. Insufficient information is given regarding the strains and alleles used. Supplementary Table S1 appears to list only new strains generated in this study. Other strains are not described anywhere in the methods (for example, *spo-11(AID)*, *syp-3(AID)* and others). The authors need to include allele numbers, strain identifiers, complete genotypes, and references for all strains used in the study. For strains created in this study, the authors should include strain numbers and complete genotypes. The authors statement in the reporting summary that “Details of the genetic backgrounds of all alleles used in this study were listed in Table S1 in the manuscript” is incorrect.

We regret that this information was incomplete and have added detailed information about all strains used in this study in Table S1 of the revised manuscript, including *spo-11^{aid}* and *syp-3^{aid}*.

6. The AID system involves adding a degron tag to each protein of interest. Although small, it is possible that addition of the tag impacts the function of the protein even in the absence of IAA. This needs to be addressed by one (or ideally more) of the following: Does the knock-in create a high incidence of males phenotype indicative of meiotic chromosome segregation defects? Does the knock-in result in embryonic lethality? Are the localization and/or levels of the tagged protein altered compared to wt? For CHK-2 and ATM-1, does addition of the tag change the levels of substrate phosphorylation (phospho-HIM/ZIM or pSQ/TQ)?

As mentioned above, we validated all of our degron-tagged alleles based on phenotypes, including brood counts. We now include these validation data as a supplementary table (Table S4). In brief, the tagging of target genes with an epitope plus the degron did not reduce viability or increase the incidence of male self-progeny.

Strain	Name	Eggs counted	Brood Size (\pm S.D.)	% viability (\pm S.D.)	% males (\pm S.D.)
WT	CA1199	3020	302 \pm 17	99.0 \pm 2.3	0.13 \pm 0.22
spo-11::aid::3xflag	CA1684	2949	295 \pm 21	99.2 \pm 3.4	0.07 \pm 0.14
ha::aid::syp-3	CA1642	2712	271 \pm 45	98.8 \pm 3.5	0.12 \pm 0.29
v5::aid::wapl-1	CA1685	2995	300 \pm 48	98.1 \pm 4.3	0.00 \pm 0.00
ha::aid::chk-2	CA1686	2982	298 \pm 28	99.1 \pm 8.6	0.07 \pm 0.15
3xflag::aid::atm-1	CA1688	2809	281 \pm 22	100.0 \pm 3.9	0.04 \pm 0.11
ha::aid::atl-1	CA1689	3080	308 \pm 37	99.3 \pm 3.0	0.00 \pm 0.00
3xflag::aid::eco-1	CA1699	2944	294 \pm 57	99.3 \pm 4.6	0.12 \pm 0.21
pds-5::AID::3xflag	CA1704	3095	310 \pm 25	98.9 \pm 2.5	0.10 \pm 0.32

Additionally, as mentioned in response to #2 & #4, above, tagging of CHK-2, WAPL-1, ECO-1, ATL-1 or PDS-5 with an [epitope + degron] did not alter their localization in the germline. We were unable to visualize degron-tagged SPO-11 and ATM-1 by IF, nor have these proteins been localized in previous studies, presumably due to their extremely low abundance. However, animals expressing only AID-tagged SPO-11 or ATM-1 showed no defects in worm growth, development, or reproduction, indicating that their functions were not markedly compromised by the addition of the degron or epitope tags.

As describe in our responses to comments #2 & #4, above, the knock-in of an epitope plus the degron to CHK-2 did not alter its localization or its activity (monitored by pHIM/ZIM staining) in the germline. We also monitored the activity of ATM-1 (by pS/TQ staining) after degron tagging and confirmed that it did not show major changes.

In summary, we think that the knock-in of the degron to the proteins of interest, including CHK-2 and ATM-1, did not disrupt their localization and function in the germline.

7. A similar concern regarding protein stability/folding/localization applies to the non-phosphorylatable/phosphomimetic/non-acetylatable mutants.

WAPL-1 – the nonphosphorylatable WAPL-1^{2A} could be visualized by our WAPL-1 antibody. It showed nuclear localization as wild type in the premeiotic region, but failed to be downregulated, similar to the localization of WAPL-1 in the *chk-2* and *atm-1* (shown in Figure 2E). These observations suggested that WAPL-1^{2A} could localize to where wild-type WAPL-1 localizes to but the downregulation was disrupted by the mutations. Moreover, when the activity of ECO-1 was depleted under the WAPL-1^{2A} background, meiotic cohesin showed localization defects that is similar to the WAPL-1-dependent cohesin release in *chk-2* (Figure 1E & 5A), suggesting that the nonphosphorylatable WAPL-1^{2A} is capable in releasing meiotic cohesin. These observations suggested that WAPL-1^{2A} localizes properly and possesses cohesin release activity. Whether the phosphomimetic WAPL-1^{2D} possesses cohesin release activity is hard to test – our results suggest that it is inactive, which we interpret as constitutive downregulation.

ATM-1 – using pS/TQ staining as the marker, we monitored the kinase activity of the phosphomimetic ATM-1^{S1853D} (Figure 4B). Statistical comparisons indicate that ATM-1^{S1853D} and wild-type ATM-1 have similar kinase activity meiotic nuclei, and that WAPL-1 is downregulated in animals expressing ATM-1^{S1853D}. As mentioned above, we were unable to detect ATM by IF or western blots so we relied on these assays as indications of proper folding/activity. The nonphosphorylatable ATM-1^{KA} showed defective ATM-1 activity, and we cannot rule out the possibility that this protein is inactive or unstable.

SMC-3 – The two ECO-1-dependent acetylation sites on Smc3 are conserved across diverse eukaryotes, including budding yeast and humans (Ünal et al, 2008, Science; Zhang et al, 2008, Mol Cell; Rowland et al, 2009, Mol Cell). Other studies have shown that the acetyl-mimetic Smc3^{QQ} assembles into cohesin complexes that localize to chromatin and antagonize Wapl-dependent cohesin release (Heidinger-Pauli et al, 2010, Genetics; Çamdere et al, 2015, eLife). In *C. elegans*, SMC-3 is thought to be an essential subunit of all cohesin complexes; given the viability and fertility of animals expressing only SMC-3^{QQ}, we can infer that the protein is functional in mitosis and meiosis.

8. Based on consensus motifs, homology to known phosphosites in orthologs, and point mutations to make unphosphorylatable, unacetylatable, and phosphomimetic versions of proteins, it is suggested that CHK-2 phosphorylates ATM-1, ATM-1 phosphorylates WAPL-1, ECO-1 acetylates SMC-3, etc. While these models seem reasonable, data proving the existence of these post-translational modifications (mass spec, modification-specific antibodies, etc.) would greatly strengthen the paper.

We agree with the reviewer's recommendation. Due to the constraint of time and our expertise, we did not perform Mass Spec analysis on the potential phosphosites of WAPL-1 and ATM-1, nor generating any phospho-specific antibodies that could recognize these potential phosphosites. We would like to perform these analyses in a follow-up study to further characterize these phosphoregulation.

9. The phenotypes analyzed stop at levels of REC-8 and COH-3/4 cohesin. Do any of the changes observed have consequences on DSB repair, interhomolog crossover recombination and bivalent formation, meiotic chromosome segregation, generation of haploid gametes, embryonic viability? How relevant are the increases and/or decreases in the intensity of the proteins analyzed to reproductive fitness?

Our results indicate that the stabilization of meiotic cohesin is governed in part by the ATM-WAPL regulatory axis, but also by other CHK-2-dependent pathways. CHK-2 is essential for meiotic DSB formation and homologous pairing and synapsis, whereas the effects of *atm-1* mutations on meiosis are relatively subtle. Depletion of WAPL-1 does not restore homolog pairing, synapsis, or DSB induction in the absence of CHK-2, making it difficult to determine how axis stability affects DNA repair, chromosome segregation, or embryonic viability, all of which depend on DSBs. We agree that these are interesting questions, but they are beyond the scope of the current study.

10. Lines 236-237: "Importantly, axial localization of COH-3/4, REC-8, and HTP-3 was restored by co-depletion of WAPL-1 (Fig. 7G-7I)." I may have missed it, but I did not see the data for COH-3/4.

We have revised the text to more accurately reflect the data shown.

11. Regarding statistical analysis and p-values: According to the methods, statistical comparisons were done using the Student's t-test unless otherwise specified. It is possible that I missed it, but I did not see anywhere where another test was used. It is not clear that the t-test is the best suited test for the quantification shown here. Is the data normal? Might a non-parametric test be better?

We appreciate this point and have revised our statistical analysis accordingly: for all data that were subjected to analysis, we performed normality test on the distribution pattern of the data. If we were to compare normally distributed data, we performed t-test to compute the p -values. For the rest, we performed Wilcoxon test, which does not require the data to be normally distributed, to compute p -values. In the case of multiple comparison using the same dataset, we performed p -value adjustment using the Bonferroni method to control for the probability of committing a type I error.

12. Validation of reagents: in the reporting summary, the authors state that antibodies that are commercially available were validated by the manufacturers, and other antibodies were validated in previous studies. This is only sometimes true. For example, commercially available antibodies rarely validated by the manufacturers by demonstrating absence of staining in mutants or other strains lacking the target. Occasionally, this has been done by the company or academic labs. If antibodies have been validated in *C. elegans*, references should be included.

We agree with the reviewer's recommendation and revised the Materials and Methods section accordingly to include the references where the commercial antibodies were tested in *C. elegans*.

13. The DAPI signal is very difficult to see in Fig. 1C, Fig. 1E, and most figures where it is shown. Often, the blue channel can be easier to see if the hue is changed to one that appears more like DAPI viewed through a microscope (i.e. lighter blue, not "true blue." As shown, the DAPI staining does not add much because it does not show colocalization of anything with chromosomes. In fact, it hurts because it makes colocalization between other proteins more difficult to see. A separate panel showing DAPI in greyscale would be more helpful than the overlay. Alternatively, perhaps a merge of just the protein staining without DAPI would be easier to interpret.

We agree with the reviewer's recommendation, and removed the DAPI channel from the merged images in Figures 1C, 1E, 2C, 2F, 3D, 5B, 5D, 6B, 6D, 7B, 7D, 7H and S5B to make them clearer and easier to interpret.

14. Fig. 1G shows that the intensity of REC-8 staining in *wapl-1; chk-2* mutants is significantly higher than in controlAID animals, suggesting that WAPL-1 destabilizes REC-8 even when it is not detectable. The authors should discuss this (if they do not already do so).

As the reviewer mentioned, it's possible that low levels of WAPL-1 (undetectable by IF) at meiotic entry could destabilize REC-8, based on our observation that WAPL-1 depletion resulted in the increase of meiotic entry REC-8 levels in Fig. 1G. We observed this using AID, and others observed a similar increase of REC-8 levels in a *wapl-1* loss-of-function allele (Woglar et al, 2020 PLOS Biology, cited in the manuscript when mentioning Fig. 1G). However, not like COH-3/4 cohesin, which is loaded at meiotic entry exclusively, REC-8 cohesin was visualized in high intensities as early as in the premeiotic region nuclei in *C. elegans* germline where WAPL-1 also localizes to (Severson et al. 2014 eLife). Also discussed in Woglar et al 2020, this fact makes the interpretation of the increase of meiotic entry REC-8 in the absence of the WAPL-1-dependent cohesin release activity complex.

15. Fig. 6C seems to be missing important controls: *atm-1(AID)* single mutant and *wapl-1(2A)* single mutant. These have been shown elsewhere, but given the nature of alleles being used, it is important that the experiments be done in parallel at the same time.

This was also noted by Reviewer #2, comment #3 (above), and we have added data for these controls to Figure S5C.

16. All figure legends would be more helpful if they stated a result instead of what is shown.

We agree with the reviewer's recommendation and revised the figure legends of main and supplementary figures to state the results.

Other comments:

1. It is unclear whether ATM-1(KA) is a mutation to alanine of the same residue altered in ATM-1(S1853D). I assume this is the case, but the nomenclature is confusing.

We apologize for the confusion. The CHK-2 consensus motif is composed by a conserved arginine at the -3 position relative to the potentially phosphorylated S/T at +1 position (Neill TO et al, 2002, JBC). The ATM-1(KA) is the mutant that has both the two conserved sites (R1850 → K & S1853 → A) mutated. The ATM-1(S1853D) is the mutant has only the potential phosphorylation site S1853 mutated to D.

2. The authors need to be careful when making statements like "Mutation of this putative phosphosite greatly reduced pS/T-Q immunofluorescence in meiotic nuclei, while the

corresponding phosphomimetic mutation resulted in normal ATM-1 activity” (lines 164-165). The mean pS/TQ intensity shown for atm-1(D1853D) in figure 4C is quite a bit lower than that in controlAID. If the authors wish to make this claim, they should at the very least show that the difference is not statistically significant. Similar potentially misleading statements occur elsewhere.

We agree with the reviewer’s recommendation and revised the text accordingly.

3. Line 169-170: the statement, “These findings indicate that the persistence of WAPL in meiotic nuclei lacking CHK-2 is a consequence of a failure to activate ATM-1” comes after a paragraph analyzing pS/T-Q IF in different atm-1 mutants. The pS/T-Q analysis does not indicate anything about WAPL levels, and the authors need to be careful in their use of the word indicate throughout the manuscript. It is often used when substantially weaker wording like “suggests that XYZ may be a consequence...” would be much more appropriate.

We have revised many statements throughout the manuscript to better differentiate between strong conclusions and suggestive data.

4. Lines 177-178: “...indicating that CHK-2 promotes breaks directly rather than through ATM activation.” Even if the requirement for CHK-2 in DSB formation is not mediated through ATM, it need not be direct.

We agree with the reviewer’s recommendation and revise the text accordingly.

5. The ControlAID treatment is never described. It also is not described how mutants like the non-phosphorylatable and putative phosphomimetic mutants were handled. Were they grown on plates containing IAA even though they were not degron tagged? If not, appropriate controls need to be included – one cannot compare, for example, atm-1(AID) grown on auxin plates to a mutant grown on normal plates and draw any conclusion (even a conclusion that the two strains appear the same is fraught).

We apologize for the confusion, and have revised the text to clarify these issues. As now stated in the Methods section and in the figure legend where it first appears (Figure 1), *control^{AID}* designates animals from the same TIR-1-expressing background strain that we injected to introduce a degron into specific genes, but the *control^{AID}* strain does not express any degron-tagged target (see Table S1 for detailed information). These animals were exposed to auxin in parallel with experimental strains expressing degron-tagged proteins. We also engineered all phosphosite and acetylation site mutations in

the same background strain (see Table S1 for detailed information). When we compared these to auxin-mediated depletion, strains expressing these site-specific mutations were also transferred in parallel to auxin-containing plates.

6. Lines 488-490: “Unless otherwise indicated, hermaphrodites at the L4 stage were transferred to seeded plates containing 1 mM IAA and incubated for 24 hours...” I may have missed it, but I did not see anywhere where other methods for IAA treatment were indicated.

We deleted this statement from the text.

7. Lines 165-166: Fig. 4A and 4B should be 4B and 4C.

We thank the reviewer for noticing this and have updated the callouts.

8. Line 195: synergistic effects are usually substantially stronger than additive/multiplicative effects. The data shown do not appear to support any synergism.

We agree and have replaced “synergistic” with “additive.”

9. Statistical significance is shown categorically (e.g. n.s., *, **, ***, ****). The journal’s editorial policies include “Give P values as exact values whenever suitable,” and this would indeed be suitable and better for the data in this manuscript.

We revised our figures to give p-values as exact values for those that are greater or equal to 0.01 according to the reviewer’s recommendation. We also revised the according Method section.

10. Image analysis: what radius was used for the rolling ball tool used for image analysis? What was done to determine that this setting was appropriate?

The standard radius used in applying the rolling ball algorithm is as large as the objects that are not considered as the background. The images we acquired were 1024x1024 pixels, with XY pixel dimensions of 0.06464 μm . Meiotic nuclei in *C. elegans* are ~3.5-4 μm (~ 50 pixels in the raw image) in diameter. We used a radius of 50 (a diameter of 100), which spans the largest object. We also tried other radii, including 25, 100 and

200, which yielded slightly different “foreground” intensities but did not alter the relationship between groups in each assay.

11. Fig. 1A and elsewhere: in some figures/legends, premeiosis and early meiosis are shown. In others, meiotic entry and midpachytene are shown. Is meiotic entry the same as early meiosis? Can midpachytene be indicated on the figure? It is nitpicky, but the Leptotene/Zygotene label seems like it is too far proximal and overlaps with where midpachytene would be found.

We apologize for the confusion. The earliest stage of meiotic prophase, which commences upon meiotic entry, is known as the transition zone or leptotene/zygotene. Early pachytene nuclei have completed synapsis but have not yet designated crossover sites. In some cases we also show nuclei at mid-pachytene. We have revised Figure 1A to be more precise.

12. Fig. 3A: it would be helpful in the legend to state which CHK2 orthologs are meiosis-specific and which are not.

We thank the referee for the suggestion and have amended the figure to show that *C. elegans* CHK-2 and *S. cerevisiae* Mek1 are meiosis-specific.

13. Fig. 3B: here and elsewhere, state what stage is being shown.

We revised the figure legends of Figures 3B, and 4B to state that they show early meiotic nuclei.

14. Fig. 4C: pS/TQ staining in *atm-1(S1853D)* is lower than in *controlAID*. Is the difference significant? This potentially lends credence to the concern that the point mutations are destabilizing the protein or otherwise impacting function in unexpected ways.

To address this issue we performed a statistical comparison between *controlAID* and *atm-1^{S1853D}*. The Wilcoxon test gave a p-value of 0.24 after Bonferroni adjustment, suggesting that the difference between the two groups may not be significant. We have revised the figure accordingly.

15. Line 413: “The conserved arginine and phospho-serine/threonine site.” It should not be called a conserved phosphosite unless phosphorylation has been demonstrated.

We have revised the text to clarify that these are “putative” phosphosites.

16. Figure 8, all IF panels: it is very difficult to see the boxes region indicating the part of the image from which the inset was taken.

We have lightened the color and changed from dashed to solid lines to make the outlined areas easier to see.

17. Figure 8: CONi and siCON should be defined in the legend, and the methods should clearly state what the controls were.

We appreciate this suggestion, which will help readers to understand and interpret the data, and have now added detailed descriptions defining CONi and siCON in the legend for Figures 8E & 8F, and in the Methods section.

18. Fig S3 is never referred to in the text.

Figure S3 and Figure S7K are called out and discussed in the Supplementary Text.

Decision Letter, first revision:

Message: Our ref: NSMB-A46048A

11th Oct 2022

Dear Dr. Yu,

Thank you for submitting your revised manuscript "ATM signaling modulates cohesin behavior in meiotic prophase and proliferating cells" (NSMB-A46048A). It has now been seen by the original referees and their comments are below. The reviewers find that the paper has improved in revision, and therefore we'll be happy in principle to publish it in Nature Structural & Molecular Biology, conditional on careful implementation of the remaining suggestions by the referees, and pending minor revisions to comply with our editorial and formatting guidelines.

We are now performing detailed checks on your paper and will send you a checklist detailing our editorial and formatting requirements in about a week. Please do not upload the final materials and make any revisions until you receive this additional information

from us.

Kind regards,
Florian

Dr Florian Ullrich
Associate Editor, Nature
Consulting Editor, Nature Structural & Molecular Biology
ORCID 0000-0002-1153-2040

Author Rebuttal, first revision:

Reviewer #1 (Remarks to the Author):

The authors have appropriately addressed the points I raised in my previous review.

We thank the referee for the positive comments and for their constructive comments, which have helped us to improve our manuscript.

Reviewer #2 (Remarks to the Author):

The revised version of the manuscript is much improved. It addressed most of my comments; only minor issues that can be addressed by text editing remain.

We appreciate the referee's positive feedback and helpful comments..

Some of the statements are still not sufficiently supported by the results and/or would benefit from editing. See list below:

1. Lines 23-25, relevant to major point 1 in the first round of review:

“We further find that cohesin-enriched domains that
23 promote DNA repair in mammalian cells also depend on WAPL inhibition by ATM. Thus, the
24 DDR and Wapl play conserved roles in cohesin regulation in meiotic prophase and
proliferating
25 cells.”

The abstract section discussing the mammalian data should be toned down. As mentioned in the first round of review, the manuscript does not show that ATM phosphorylates WAPL leading to WAPL inhibition. The data only shows that overexpression of a WAPL version that lacks putative ATM phospho-sites is able to inhibit ATM-mediated accumulation of cohesion at DNA break sites. I would suggest to rephrase the abstract along the following lines: “Further, our data suggest that cohesin-enriched domains that promote DNA repair in mammalian cells also depend on WAPL inhibition by ATM. Thus, DDR and Wapl seem to play conserved roles in cohesin regulation in meiotic prophase and proliferating cells.”

We revised the Abstract according to the reviewer’s suggestions (highlighted in the text).

2. Relevant to minor point 6. in the first round of review:

“Surprisingly, we find that 80 ATM-1 is activated at meiotic entry by the CHK-2 (Chk2) kinase, and that CHK-2 also promotes

81 the activity of PDS-5 (Pds5) and ECO-1 (Eco1), which preferentially protect cohesin complexes

82 containing REC-8 from the effects of WAPL-1.”

Whereas the data are consistent with CHK2 promoting PDS5 and ECO-1 function, the effect of CHK2 on PDS-5 (Pds5) and ECO-1 (Eco1) activity were not directly tested. The experiments only show that these three proteins synergistically promote cohesion accumulation. Therefore, lines 80-82 should be rephrased accordingly.

We agree with these comments and have revised our manuscript accordingly (highlighted in the text).

3. Reference to WAPL-1 regulation by CHK-1 mediated phosphorylation of ATM-1 is premature and repetitive in lines 163 and 179 as WAPL-1 localization experiments are described only after line 180 and conclusion from this paragraph in lines 193-195 restates earlier conclusions. The data relevant to lines 163 and 179 addresses only ATM-1 activity and not the role of ATM-1 activity in WAPL regulation.

Therefore I suggest omitting references to WAPL regulation in lines 163 and 179. For example instead of : “Together, these results indicate that CHK-2 promotes basal levels of 163 ATM-1 activity that mediate WAPL-1 downregulation even in the absence of meiotic DSBs.” I suggest”Together, these results indicate that CHK-2 promotes basal levels of ATM-1 activity in the absence of meiotic DSBs.”

Lines 173-179 may also confuse readers because the activity of phosphorylated/phosphomimetic ATM activity is called both high and basal in the absence of DSBs/CHK2:

“Additionally, phosphomimetic

174 ATM-1S1853D showed high activity even in the absence of CHK-2, supporting the conclusion that
 175 CHK-2 activates ATM-1 through this site (Fig. 4B-4D). Depletion of either CHK-2 or SPO-11 in
 176 animals expressing ATM-1S1853D resulted in similar levels of anti-pS/TQ immunofluorescence,
 177 (Fig. 4E-4F), suggesting that DSBs induce higher ATM-1 activity, but break-independent
 178 phosphorylation by CHK-2 results in a basal level of activity that is sufficient for WAPL-1
 179 downregulation.”

I suggest rephrasing along the lines:

“Additionally, in the absence of CHK-2, phosphomimetic ATM-1S1853D showed significant activity albeit at lower levels than in CHK2 proficient backgrounds (Fig. 4B-4D). Further, depletion of either CHK-2 or SPO-11 in animals expressing ATM-1S1853D resulted in similar levels of anti-pS/TQ immunofluorescence, (Fig. 4E-4F). Together these observations suggest that DSBs enhance ATM-1 activity, but break-independent phosphorylation of S1853 by CHK-2 enables basal level of ATM-1 activity.”

We thank the reviewer for pointing out this issue and we revised the manuscript to more clearly distinguish between basal and maximal (DSB-dependent) levels of ATM activity. (highlighted in the text)

4. Lines 195-197:

“Consistent with this interpretation, mutations in
 196 ATM-1 or WAPL-1 restored robust axis localization of COH-3/4 in the absence of CHK-2
 (Fig.
 197 5D and 5E).”

This sentence is vague in its current form, please, specify which ATM and WAPL mutations the sentence refers to.

We revised the statement to specify phosphomimetic mutations in ATM-1 and WAPL-1 (highlighted in the text).

5. Line 198 is an overstatement relevant to minor point 6. in the first round of review:

“CHK-2 also promotes axis assembly through cohesin acetylation independent of WAPL regulation”

Please tone down, e.g. “CHK-2 may promote...” or “CHK-2 also promotes axis assembly by synergising with cohesin acetylation independent of WAPL regulation”

We have edited or removed statements that implied that CHK-2 regulates ECO-1 or acetylation (highlighted in the text).

6. Experiments in figure 8. It is confusing that in/out ratios were calculated for RAD21 enrichment in DNA damage foci in DMSO controls. The images show no focal staining of pS/TQ or H2AX, indicating that there was no or very little DNA damage in the DMSO treated control. Hence I do not understand how ratios of RAD21 levels were measured within and outside of DNA damage foci in DMSO controls. Either these ratios should be better explained or omitted for the DMSO control. Further, in/out ratios of RAD21 in BLEO or ETO treated samples do not have to be compared with DMSO treated samples as DNA damage foci do not really exist in the DMSO control. It is sufficient to test if there is a significant difference from a hypothetical mean=1 in/out ratio to tell if there is enrichment of RAD21 in the damage foci generated by BLEO or ETO. One sample t test can be used to carry out the statistical test.

We apologize for the confusion. Although the intensities of phospho-H2A.X signals are much lower in DMSO-treated cells compared to cells treated with DNA-damaging drugs, we still observed some foci in DMSO-treated cells. We therefore used these weak H2A.X foci/basal signals as “in” regions to calculate RAD21 intensity ratios of DMSO-treated/control cells. The results of the statistical analysis are the same as to simply assume that DMSO-treated/control cells have the hypothetical mean=1 as suggested.

7. Lines 256-257 contain an overstatement relevant to major point 1 in the first round of review:

“ATM-dependent phosphorylation of WAPL mediates cohesin concentration at mammalian DNA damage foci”

The data does not show that ATM phosphorylates WAPL, hence the title for the last Result section should be changed. For example:

“ATM-mediated cohesin concentration at mammalian DNA damage foci is effectively countered by WAPL that is nonphosphorylatable on potential ATM sites.”

We thank the reviewer for pointing out this issue and revised the heading to read, “**ATM-mediated enrichment at mammalian DNA damage foci is regulated through WAPL**” (highlighted in the text)

Reviewer #3 (Remarks to the Author):

The revised manuscript by Yu, Kim, and Dernburg includes new experiments and data analyses that address many of the reviewers' concerns, as well as changes to the text and figures that clarify the experimental approaches and the data and conclusions. The authors should be commended for their clear and thoughtful responses to referees' comments, which were often followed by excerpts from revised figures showing the changes made. I found this quite helpful.

We thank the referee for the positive comments and suggestions for improvement.

My major concerns in the initial review centered on the heavy reliance on the AID degron system in this study. Many of those concerns have been adequately addressed. The addition of Table S4 demonstrates that meiotic progression and chromosome segregation are not impacted by addition of the AID and epitope tags, at least in single mutants. The clarification in the methods that all strains in a given experiment were grown in parallel on the same batch of plates is also quite helpful. However, I do still have some concerns. Most of these could be addressed with minor wording changes, but including as supplemental material some data the authors included in their rebuttal would be even better.

1. Throughout the manuscript, AID-mediated degradation is typically referred to as "depletion." This is appropriate. However, lines 104 "in the absence of CHK-2," 118 "in the absence of both PLK-1 and PLK-2," 244 "in the absence of PDS-5 and CHK-2," and elsewhere overstate what is known, and the word "absence" in this context should be avoided since absence has not been demonstrated and is likely impossible to demonstrate.

We thank the reviewer for pointing out these issues and we have revised the text and figure legends to more precisely refer to "depletion" of proteins, rather than "absence" (highlighted in the text).

2. In most of the experiments shown, auxin treatment resulted in a detectable phenotype, suggesting that at least some amount of depletion occurred even in cases where depletion is not shown by IF or western. However, negative results were obtained in a few cases. In these cases, it would strengthen the conclusions if the authors included evidence that depletion occurred. For example: Line 123 "WAPL-1 localization was also unaffected when we depleted SYP-3..." SYP-3 depletion is expected to cause asynapsis, and I believe this can be seen in Fig S1F, although this could be demonstrated more clearly in a supplemental figure. If unsynapsed chromosomes were present following auxin treatment, the authors could state, "WAPL-1 localization was also unaffected when we depleted SYP-3, an essential component of the synaptonemal complex (SC), although chromosome synapsis was disrupted demonstrating that depletion was effective." Similarly, the conclusion that ATL-1 depletion had no effect on WAPL-1 (Line 131) would be strengthened if the western blot shown in the rebuttal were included in the supplementary figures. This would also

strengthen the overall conclusion of this paper that ATM and not ATR is the key kinase, i.e. the absence of a detectable function for *atl-1* is not the consequence of a failure to knock it down.

We agree with the reviewer and revised the text in regards to SYP-3 depletion in the manuscript (highlighted in the text).

We agree with the reviewer that ATL-1 depletion is a little trickier to interpret without a Western blot analysis, such as the data we included in our rebuttal letter. We have included Western blot analysis of ATL-1 following depletion as a new extended data Figure S2G (included in the figure file & highlighted in the text). Additionally, we added a comment that depletion of ATL-1 phenocopies the effects of an *atl-1* null mutation on the size and number of germline nuclei (see Figures 2b & 2c)(Garcia-Muse et al, 2005, EMBO J, highlighted in the text). Additionally, AID-mediated depletion of ATL-1 resulted in 100% embryonic lethality, like the deletion allele *atl-1(tm853)* (data not shown).

3. I agree with the authors that simultaneous depletion of multiple targets by AID has, at least in some cases, proven to be efficient. However, this does not mean that it will be efficient in every case. One place where this is a concern is the experiment in Figure 7, which the authors address in their rebuttal. In Fig 7, simultaneous depletion of CHK-2 and PDS-5 gives a phenotype that is rescued when a third protein, WAPL-1, is also depleted. In the absence of proper controls, it is possible that addition of the third target overwhelms the TIR1-dependent ubiquitin proteasome system resulting in higher levels of CHK-2 and PDS-5 than in the double depletion. In their rebuttal, the authors show a convincing control that all three proteins are undetectable in the triple depletion strain. Since the authors have the data, why not show it in a supplement since it removes the concern? As a bonus, it would also address the restoration of COH-3/4 to wt levels in *wapl-1 chk-2* vs *chk-2*.

Alternatively, if the data are not shown, the authors could state that the return to essentially wt phenotype is unlikely to be a consequence of saturating the system, since other studies have shown that up to four proteins can be depleted simultaneously. Although this is not a strong argument, it acknowledges the possibility and counters it by informing the reader that there are precedents where this has been carefully addressed.

We have now included these two control experiments as extended data Figures S6e and S6Ff (included in the data figure files). As noted in our previous rebuttal, we have seen no indication of saturation or reduced efficiency of the AID system when multiple targets are expressed. The reviewer's concern that the system might become saturated likely stems from studies of RNA interference (RNAi), which becomes less efficient if multiple genes are targeted, possibly due to limiting concentrations of the proteins that amplify the small RNAs required or the Argonaute effectors. However, the ubiquitin-proteasome system is inherently a very high-flux system, particularly in dividing cells. Experimental evidence has shown that it typically operates far from saturation.

Proteasomes are estimated to comprise ~2% of cellular protein; mammalian cells are estimated to have $\sim 1 \times 10^6$ proteasomes, but only 200K are engaged. Adding high concentrations of small molecule degraders (PROTACs) does not saturate proteasomes (see <https://doi.org/10.1016/j.cell.2020.10.038>). Moreover, some E3 ligases have dozens of substrates but do not show evidence of saturation (<https://doi.org/10.1016/j.ceb.2009.08.004>). Our TIR1 protein is expressed under a strong germline promoter. Substrates bind only for seconds before they are degraded (<https://doi.org/10.1016/j.cell.2013.02.024>).

Other comments:

1. Line 118 "... in the absence of both PLK-1 and PLK-2 ..." Unless I missed it, the PLK-1/2 data are not shown and should either be included or not mentioned.

The data were presented in Reference #32, which is cited.

2. The title of Table S1 should not be "Strains generated in this study," since a few of the strains were generated in other studies. Perhaps, "Strains used in this study" is more accurate. The References column makes it clear which are new.

We have revised the title of Source Data Extended Table 1, as sugges

Final Decision Letter:

Message 25th Jan 2023

:

Dear Dr. Dernburg,

We are now happy to accept your revised paper "ATM signaling modulates cohesin behavior in meiotic prophase and proliferating cells" for publication as a Article in Nature Structural & Molecular Biology.

After the grant of rights is completed, you will receive a link to your electronic proof via

email with a request to make any corrections within 48 hours. If, when you receive your proof, you cannot meet this deadline, please inform us at rjsproduction@springernature.com immediately.

Due to the importance of these deadlines, we ask that you please let us know now whether you will be difficult to contact over the next month. If this is the case, we ask you provide us with the contact information (email, phone) of someone who will be able to check the proofs on your behalf, and who will be available to address any last-minute problems.

Your paper will be published online soon after we receive proof corrections and will appear in print in the next available issue. You can find out your date of online publication by contacting the production team shortly after sending your proof corrections. Content is published online weekly on Mondays and Thursdays, and the embargo is set at 16:00 London time (GMT)/11:00 am US Eastern time (EST) on the day of publication. Now is the time to inform your Public Relations or Press Office about your paper, as they might be interested in promoting its publication. This will allow them time to prepare an accurate and satisfactory press release. Include your manuscript tracking number (NSMB-A46048B) and our journal name, which they will need when they contact our press office.

About one week before your paper is published online, we shall be distributing a press release to news organizations worldwide, which may very well include details of your work. We are happy for your institution or funding agency to prepare its own press release, but it must mention the embargo date and Nature Structural & Molecular Biology. If you or your Press Office have any enquiries in the meantime, please contact press@nature.com.

If you have not already done so, we strongly recommend that you upload the step-by-step protocols used in this manuscript to the Protocol Exchange. Protocol Exchange is an open online resource that allows researchers to share their detailed experimental know-how. All uploaded protocols are made freely available, assigned DOIs for ease of citation and fully searchable through nature.com. Protocols can be linked to any publications in which they are used and will be linked to from your article. You can also establish a dedicated page to

collect all your lab Protocols. By uploading your Protocols to Protocol Exchange, you are enabling researchers to more readily reproduce or adapt the methodology you use, as well as increasing the visibility of your protocols and papers. Upload your Protocols at www.nature.com/protocolexchange/. Further information can be found at www.nature.com/protocolexchange/about.

Please note that *Nature Structural & Molecular Biology* is a Transformative Journal (TJ). Authors may publish their research with us through the traditional subscription access route or make their paper immediately open access through payment of an article-processing charge (APC). Authors will not be required to make a final decision about access to their article until it has been accepted. [Find out more about Transformative Journals](https://www.springernature.com/gp/open-research/transformative-journals)

Sincerely,

Dimitris Typas
Associate Editor
Nature Structural & Molecular Biology
ORCID: 0000-0002-8737-1319

Click here if you would like to recommend Nature Structural & Molecular Biology to your librarian:

<http://www.nature.com/subscriptions/recommend.html#forms>